# Microcephaly-associated protein WDR62 supports purine metabolism by interacting with co-chaperone BAG2

Matthew J Morris [iD][1], Yvonne Y Yeap[1,3], Jonathon R Edwards[1], Chi Chen[1], Annalisa Paolino [iD][1], Sebastian G B Furness [iD][1], S Sean Millard [iD][1], Julia K Pagan [iD][1], Laura R Fenlon[1,2] & Dominic C H Ng [iD][1✉]

## Abstract

Inherited mutations in the spindle pole-associated scaffold protein WDR62 cause autosomal recessive primary microcephaly. Previous research has characterised the roles of WDR62 in the regulation of spindle dynamics, cell division, and brain development. Here, we identify a new function of this protein in regulating purine metabolism. WDR62 interacts directly with BAG2, a co-chaperone of HSP70/90. Under stress conditions, WDR62 and BAG2 re-localise to cytoplasmic granules enriched for enzymes involved in purine synthesis (PFAS) and salvage (HPRT). In WDR62-deficient cells, purine synthesis is impaired, while purine deprivation leads to cytotoxicity and nucleoside accumulation. Furthermore, in these cells elevated BAG2 levels are linked to HPRT destabilisation, which can be reversed by BAG2 knockdown. Notably, microcephaly-associated WDR62 mutations disrupt interaction with BAG2 and fail to restore HPRT levels. *In utero* depletion of WDR62 or HPRT in the mouse neocortex causes premature delamination and migration of neural precursor cells. Interestingly, HPRT loss enhances self-renewal and proliferation of these precursors, contrasting with the reduced proliferation and precocious differentiation observed upon WDR62 loss. Our study identifies regulatory functions of WDR62 in purine metabolism that may contribute to primary microcephaly.

**Keywords** BAG2; Microcephaly; Molecular Chaperones; Purine Metabolism; WDR62
**Subject Categories** Development; Metabolism; Neuroscience

## Introduction

Microcephaly is a complex neurodevelopmental condition attributed to inherited mutations or prenatal exposure to infection, metabolic imbalances, or chemical toxicity (Woods, 2004). One commonly mutated microcephaly gene, *MCPH2*, encodes WD40 repeat-containing protein 62 (WDR62), a protein that is highly expressed in the proliferative zones of the developing neocortex and has vital non-redundant functions in embryonic brain growth (Bilgüvar et al, 2010; Yu et al, 2010; Nicholas et al, 2010).

WDR62 exhibits a dynamic subcellular distribution that is coordinated with the cell cycle (Lim et al, 2015; Bogoyevitch et al, 2012). WDR62 was initially characterised as a c-Jun N-terminal kinase (JNK)-binding protein involved in stress-activated signal transduction (Wasserman et al, 2010). During mitosis, WDR62 is associated with spindle microtubules and is required for spindle organisation, mitotic progression, and division polarity (Lim et al, 2015; Chen et al, 2014; Shohayeb et al, 2020a; Shohayeb et al, 2020c; Shohayeb et al, 2020b; Sanchez et al, 2021). The ability of WDR62 to regulate the orientation of mitotic divisions may have significant consequences on daughter cell fate, particularly during the proliferation of progenitor cell populations (Shohayeb et al, 2020b; Hu and Jasper, 2019). In addition, a proportion of WDR62 localises to centrosomes, where it contributes to centriole biogenesis and ciliogenesis (Shohayeb et al, 2020a; Zhang et al, 2019). Consequently, loss or mutation of WDR62 results in abnormal centrosome numbers that may cause genetic instability or alter primary cilia dynamics, both of which define the self-renewal capacity of progenitor cells (Shohayeb et al, 2020a; Jayaraman et al, 2016). Thus, WDR62 is a multifaceted protein crucial for cell survival, proliferation, differentiation, and the integration of environmental cues.

Despite the complex functions attributed to WDR62, it remains unclear precisely how WDR62 mutations lead to microcephaly phenotypes. For example, WDR62-related spindle orientation defects may not be sufficient to induce defects in mitotic index, cell cycle delay, and cortical neurogenesis (Megraw et al, 2011; Thornton and Woods, 2009; Noatynska et al, 2012; Homem and Knoblich, 2012; Zhang et al, 2019; Huang et al, 2021). Further, we previously reported that neuroblast-specific depletion of WDR62 in *Drosophila* did not substantially change brain volume, while the loss of WDR62 in the glial lineage reduced brain volume without observable effects on spindle orientation (Lim et al, 2017). Similarly, misoriented divisions in neural stem cells due to the

[1]School of Biomedical Sciences, Faculty of Health, Medicine and Behavioural Sciences, The University of Queensland, Brisbane, QLD 4072, Australia. [2]Queensland Brain Institute, The University of Queensland, Brisbane, QLD 4072, Australia. [3]Present address: Singapore Immunology Network, Agency for Science, Technology and Research (A*STAR), 8A Biomedical Grove, Singapore 138648, Singapore. ✉E-mail: d.ng1@uq.edu.au

loss of polarity do not consistently alter brain volume in mice (Imai et al, 2006; Konno et al, 2008). Moreover, cytoskeletal, centrosomal and cell cycle processes are interconnected. This presents challenges in delineating the direct and indirect effects of WDR62 mutations.

In addition to spindle association, a large proportion of WDR62 is cytosolic during interphase and dynamically associated with the microtubule cytoskeleton, centrioles and the Golgi apparatus (Nicholas et al, 2010; Dell'Amico et al, 2023). In response to cell stress, WDR62 is recruited to cytosolic stress granules (SGs) comprised of RNA-binding proteins (Wasserman et al, 2010). Despite well-described functions for cytoskeletal WDR62 and initial studies in SGs, the role of cytosolic WDR62 has remained undercharacterised. Using BioID to identify novel interactors, we found that WDR62 interactors included the Bcl-2-associated athanogene 2 (BAG2) molecular co-chaperone and other heat shock protein (HSP) members from the HSP70/90 family. This was consistent with WDR62 interactors previously reported, including members of the DNAJ (HSP40) family, that are also HSP70 co-chaperones (Cyr et al, 2023). This chaperone network regulates the stability, turnover and higher-order assembly of a diverse range of client proteins, including enzymes involved in purine metabolism (Pedley et al, 2018; French et al, 2013; Pedley et al, 2022). Recent revelations that HSP70 co-chaperones regulate the production of purines, which are essential for cortical development (French et al, 2013; Fumagalli et al, 2017), raise the question of whether neurometabolic dysfunction may be involved in WDR62-associated disease.

Here, we demonstrate that, under conditions of cellular stress, WDR62 was recruited to dynamic cytoplasmic granules containing molecular chaperones and enzymes involved in purine metabolism. The depletion of WDR62 led to perturbed purine synthesis and a reduction in enzymes involved in the de novo purine biosynthesis pathway. Moreover, loss of WDR62 accelerates the degradation of hypoxanthine-guanine phosphoribosyl transferase (HPRT), a key enzyme in the purine salvage pathway. We linked these effects on HPRT stability to dysregulation of the HSP70/HSP90-BAG2 chaperone machinery. Finally, we showed that depletion of HPRT led to in vivo neural progenitor defects that overlapped with WDR62 loss. Together, our findings uncover a previously unrecognised role for WDR62 in regulating purine metabolic enzymes and maintaining purine homeostasis, functions which influence embryonic brain development (Witteveen et al, 2022; Ceballos-Picot et al, 2009; Kang et al, 2013; Yeh et al, 1998).

## Results

### WDR62 interacts with molecular chaperones and their co-regulators involved in protein folding

To identify novel WDR62 binding partners, we generated WDR62 fusion proteins tagged with the BirA* promiscuous biotin ligase (BioID). This allowed for spatially restricted biotin labelling, affinity isolation, and detection of proteins and transient interactors within near proximity of WDR62 (Fig. 1A) (Roux et al, 2012). Exogenously expressed WDR62-BirA*-HA and WDR62-HA were localised to the cytosol and spindle poles in interphase and mitotic cells, respectively (Fig. 1B), resembling the spatial distribution of

endogenous WDR62 (Nicholas et al, 2010). We validated, using fluorophore-conjugated streptavidin, that biotin labelling occurred only in cells expressing BirA* fusion proteins and following biotin feeding (Fig. 1B). We also confirmed that WDR62-BirA*-HA expression led to biotinylation of established WDR62 binding partners such as JNK1/2, centrosomal protein 170 (CEP170), Aurora A Kinase (AURKA), and MAPK Binding Protein 1 (MAPKBP1) (Fig. 1C). WDR62 biotinylated AURKA in mitotic cells specifically (Fig. 1D), consistent with cell cycle stage specific interaction between these binding partners (Lim et al, 2015).

Satisfied that WDR62-BirA*-HA behaved similarly to endogenous WDR62, we performed BioID proximity-labelling screens in asynchronous or mitotically arrested (low-dose nocodazole) cells, to distinguish cell cycle-dependent proximity partners, isolated biotinylated proteins by streptavidin pulldown, and identified proteins in the pulldown fraction by LC-MS/MS. Overall, we identified 42 interactors, with 5 of these matching entries in the BioGRID database. The latter had compiled 95 interactors from a range of studies using affinity capture, yeast two-hybrid and other biochemical analyses of protein-protein interactions (Fig. 1D) (Stark et al, 2006). This identified members of the chaperonin-containing TCP1 complex (CCT2-CCT8, TCP1), the stress-responsive molecular chaperones HSP60 and HSP70, as well as several HSP40 family members (e.g., BAG2, BAG5, DNAJA2, DNAJC7) that function as HSC70/90 and HSP70/90 co-chaperones under basal and stress-activated conditions (Fig. 1D) (Mayer and Bukau, 2005; Huttlin et al, 2017). Mitotic arrest of cells led to the identification of a limited set of additional interactors including AURKA (Fig. 1E). Interestingly, WDR62-BirA* labelling of BAG2 was reduced in mitotically arrested cells compared to asynchronous populations (Fig. 1C). As BAG2 expression levels were unchanged in mitosis, unlike CEP170 and AURKA, WDR62 interactions with BAG2 appears to be reduced in mitotic cells and increased during interphase (Fig. 1C). Gene ontology (GO) analyses indicated that WDR62 proximal partners were significantly enriched for proteins localised to the cytosol and microtubule cytoskeleton and involved in the regulation of protein stability and homeostasis, with an emphasis on chaperone-mediated protein folding (Fig. 1F–H). We next used STRING to visualise a network of WDR62-associated proteins identified in our proximity labelling experiments (Fig. 1I). This reinforced a large subset of interactors with functions in protein folding in addition to centrosomal and cytoskeletal functions (Fig. 1I). Our results demonstrate that WDR62 interacts with protein chaperone complexes that regulate protein folding and stability.

### WDR62 microcephaly mutations alter binding to BAG2

Having identified BAG2 in both our BioID study and among the published interactors of WDR62, we investigated its interaction with WDR62 in more detail using AlphaFold2-Multimer (AF2) to predict heterodimeric WDR62-BAG2 structures. Considering that WDR62 contains long disordered domains, our initial approach involved evaluating the validity of predicted structures through predicted alignment error (PAE) plots. Our primary objective was to identify residues with low error scores (high confidence), particularly those involved in a predicted protein-protein interface. The predicted structural model of WDR62 showed that its WD repeat domains fold into two adjacent seven-bladed beta-propeller structures, predicted with high confidence (mean pLDDT: 76.34) (Fig. 2A,B). Conversely, most of the C-terminal half of WDR62

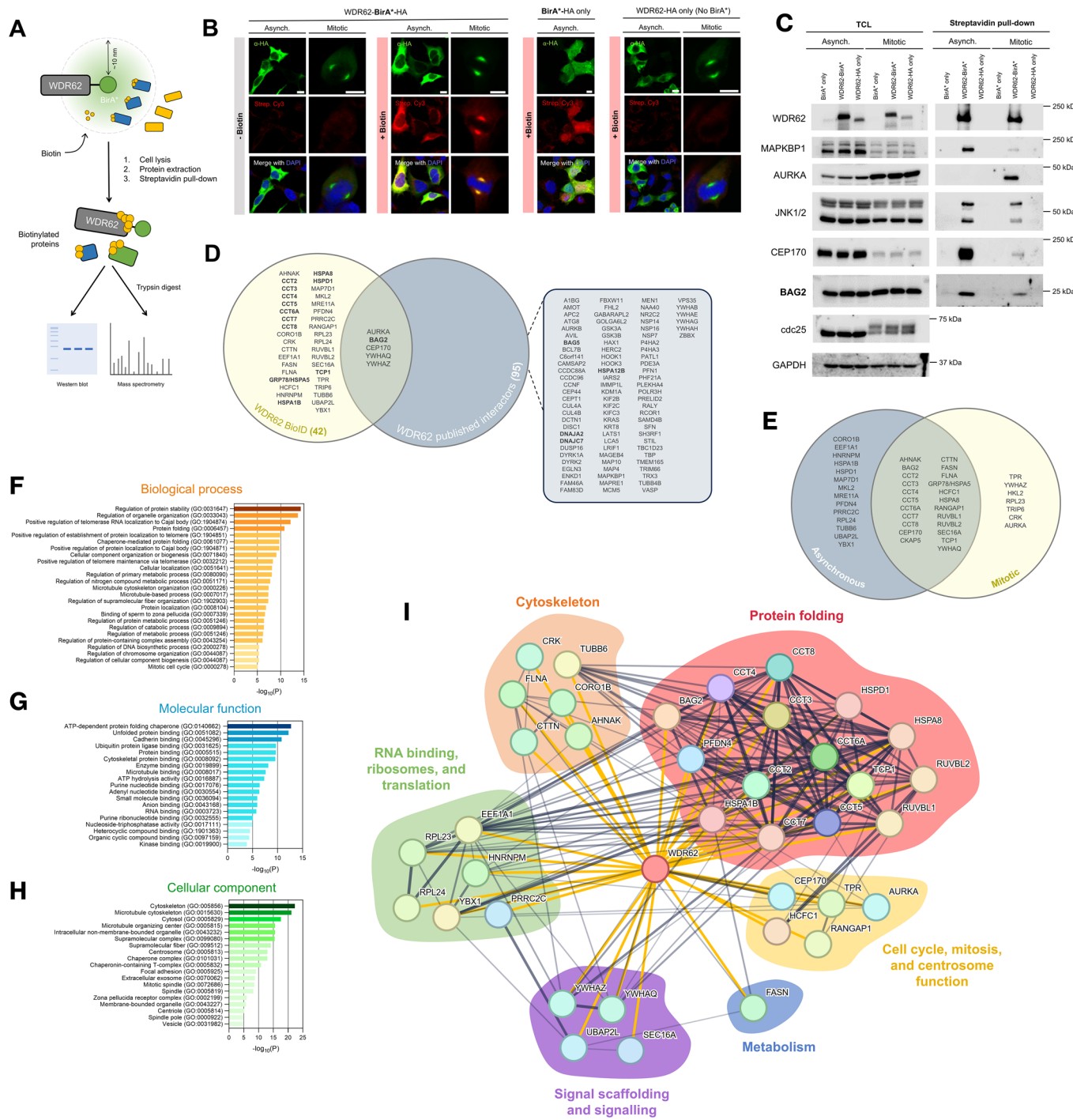

**Figure 1. BioID proximity-interaction analysis of WDR62.**

(A) Schematic of the BioID study of WDR62-interacting proteins. (B) Validation of WDR62-BirA*-HA localisation in AD293 cells by fluorescence microscopy. (C) Streptavidin-coupled beads were used to capture bioinylated proteins, and subsequent immunoblot confirms MAPKBP1, AURKA, JNK1/2, CEP170 and BAG2 as interactors of WDR62 in asynchronous and mitotically arrested cells. (D) Venn diagram of published WDR62 interactors (BioGRID database), interactors identified through BioID screening, and interactors found in both datasets. (E) BioID-identified interactors were sorted into those identified in asynchronous or mitotic cells, or both. (F–H) Gene ontology (GO) analysis of significantly enriched (F) biological processes, (G) molecular functions, and (H) cellular components mediated/localised by WDR62 interactors. GO term enrichment was performed using the PANTHER over-representation test (Fisher's exact test, FDR-corrected). The x-axis shows -log$_{10}$(FDR), labelled as −log$_{10}$(P). (I) STRING-based reconstruction of the network of proteins identified as interactors of WDR62 by our BioID study. Yellow lines indicate empirically confirmed interactions, grey lines are interactions curated in the STRING protein interaction database. Data represent n = 3 independent replicates. Scale bars represent 10 μm. Source data are available online for this figure.

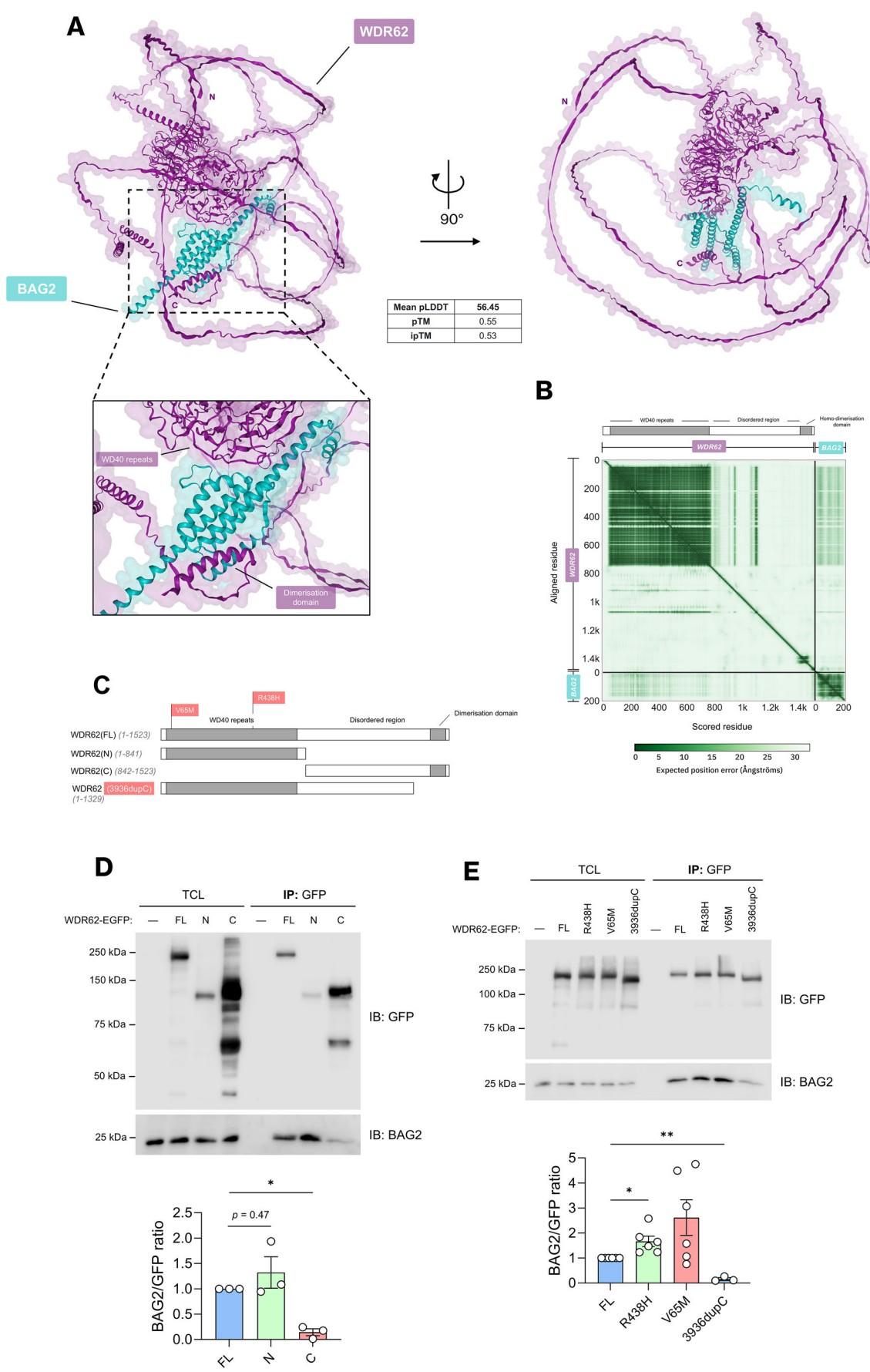

**Figure 2. Microcephaly-associated mutations in WDR62 alter binding to BAG2.**

(A) Top-ranked model predicted by AlphaFold2 between WDR62 and BAG2 with calculated mean predicted local distance difference test (pLDDT), predicted template modelling (pTM) and interphase predicted template modelling (ipTM) scores. (B) Predicted aligned error (PAE) plot of predicted WDR62-BAG2 heterodimer. X- and y-axes show indexed residues of corresponding subunits, as indicated. Aligned error in angstroms (Å) is colour coded (green = low PAE (high confidence), white = high PAE (low confidence). (C) Schematic depicting WDR62(FL) (aa 1–1523), truncated mutants WDR62(N) (1–841), WDR62(C) (842–1523), and microcephaly-associated mutants WDR62(3936dupC) (1–1329), V65M and R438H. (D) GFP-trap immunoprecipitation of WDR62(FL)-, WDR62(N)- and WDR62(C)-EGFP and immunoblot for GFP and endogenous BAG2 (*$P = 0.0372$). (E) GFP-trap immunoprecipitation of WDR62(FL)-, WDR62(R438H)-, WDR62(V65M), and WDR62(3936dupC)-EGFP and immunoblot for GFP and endogenous BAG2 (*$P = 0.0108$, **$P = 0.0089$). Densitometric BAG2 band values were normalised and expressed as BAG2/GFP ratio for (D, E) represented underneath the blots. Data represent mean ± SEM of $n > 3$ independent replicates. $P$ values calculated based on mean values using a one-way ANOVA (*$P < 0.05$, **$P < 0.005$, ***$P < 0.001$, ****$P < 0.0001$, n.s. is $P > 0.05$). Source data are available online for this figure.

exhibited low confidence scores (mean pLDDT: 35.74), indicative of disorder and the absence of a fixed three-dimensional structure (Fig. 2A,B). The predicted WDR62-BAG2 heterodimer demonstrated that BAG2 interacted with one of the two beta-propeller domains of WDR62, which contain WD repeat motifs (Fig. 2A). In addition, in most predicted models, the C-terminal tail of WDR62, which includes the α3 helix–loop–helix domain (Cohen-Katsenelson et al, 2013), appeared to interact with BAG2 on the side opposite its interface with the WD repeat domains (Fig. 2A). Indeed, heterodimeric structure predictions of BAG2 with a microcephaly-identified mutant of WDR62 lacking this helix–loop–helix domain (WDR62(3936dupC), aa 1–1329), indicated a reduction in interface area (Å²) in comparison to full-length WDR62 (Appendix Fig. S1). This observation suggests that BAG2 binds WDR62 primarily through its WD repeat domains and may rely on the C-terminal helix–loop–helix domain for additional stability.

We next sought to empirically validate these predictions and determine which WDR62 regions were required for binding BAG2. To this end, we assessed the interaction of BAG2 with EGFP-tagged full-length WDR62 (WDR62(FL)-EGFP), in addition to N-terminal (WDR62(N)-EGFP, aa 1–841) and C-terminal fragments (WDR62(C)-EGFP, aa 842–1523) corresponding to the WD40 repeat and disordered regions of WDR62, respectively (Appendix Fig. 2C). We isolated EGFP-tagged proteins with GFP-trap agarose and assessed co-precipitation of endogenous BAG2 by immunoblot. Consistent with our findings from the BioID labelling system and streptavidin pulldown, we observed that endogenous BAG2 interacts with full-length WDR62 (Fig. 2D). In addition, consistent with our AF2 prediction, BAG2 co-immunoprecipitated with WDR62(N)-EGFP, but this was significantly reduced with WDR62(C)-EGFP (Fig. 2D), suggesting an interaction with the N-terminal half of WDR62 enriched in WD repeat motifs. Given that chaperones typically bind to unfolded regions of clients (Koldewey et al, 2016), the interaction of BAG2 with the ordered WD repeat regions of WDR62 suggests that WDR62 is likely a binding partner of BAG2 and not a client or substrate.

We next determined whether patient-identified mutations in WDR62 affected BAG2 binding. These included two missense mutants (R438H and V65M) (Shohayeb et al, 2020a; Shohayeb et al, 2020b), and the frameshift mutant (3936dupC) (Nicholas et al, 2010; Yu et al, 2010; Kousar et al, 2011), mentioned above (Fig. 2C). Interestingly, we found that the R438H and V65M mutants immunoprecipitated significantly more BAG2 compared to wild-type WDR62 (Fig. 2E). Thus, the N-terminal region of WDR62 is sufficient for binding BAG2 and this is altered by microcephaly-associated missense mutations, possibly reflecting dysregulated interaction or increased chaperone

binding to help refold mutant proteins. In contrast, the 3936dupC frameshift mutant of WDR62, led to a substantial decrease in BAG2 interaction, consistent with our AF2 prediction (Appendix Fig. S1C). This indicates that, while the disordered C-terminal half of WDR62 is not sufficient to bind BAG2, the C-terminal helix–loop–helix domain —required for WDR62 dimerisation (Cohen-Katsenelson et al, 2013) —is necessary but not sufficient for BAG2 to bind full-length WDR62. These results indicate that patient-identified mutations in WDR62 substantially alter interactions with BAG2 and may impact WDR62 functions in the chaperone network.

## WDR62 assembles dynamic cytoplasmic granules distinct from conventional stress granules

BAG2 assists with the chaperone folding cycle (Qin et al, 2016; Herr et al, 2008), and it was recently revealed that the phase-separation of BAG2 into stress-induced condensates directs client proteins for ubiquitin-independent proteasomal degradation (Carrettiero et al, 2022). Similarly, WDR62 was previously reported to be recruited to cytoplasmic SGs, which are biomolecular condensates comprised of RNA and RNA-binding proteins (Guillén-Boixet et al, 2020; Wasserman et al, 2010).

In response to acute hyperosmotic stress, induced by sugars such as dextrose or sucrose, or polyols such as sorbitol (0.5 M, 1 h), WDR62 assembled distinct cytoplasmic puncta or granules (Fig. 3A). As a control, we confirmed that mCherry alone, or the WDR62 paralog MAPKBP1 do not form puncta under identical stress conditions (Appendix Figs. S2 and S3). We observed the formation of WDR62 granules at sorbitol concentrations exceeding 50 mM, with a gradual increase in granule formation as we incrementally raised the concentration over a 10 min period (Fig. 3B). WDR62 granules exhibited homogenous staining throughout their z-volume and ranged from spherical to irregular in shape with an average circularity of 0.58 ± 0.02 (Fig. 3C,D). In response to sorbitol treatment, the majority (>90%) of cells assembled WDR62 granules (Fig. 3E), with an average number of 102.8 ± 10.2 granules per cell that were on average 0.86 ± 0.15 μm in diameter (Fig. 3F). Through live-cell imaging, we observed a robust and highly rapid diffuse-to-punctate redistribution of WDR62 into cytoplasmic granules within 10 s of sorbitol exposure (Fig. 3G). Subsequently, WDR62 granules rapidly disassembled within seconds of sorbitol removal (Fig. 3G). We also observed that WDR62 granules underwent fission and fusion events (Fig. 3H). Figure 3H,I shows one such event, where granule 1a dissociates from granule 1 and fuses with granule 2 (Fig. 3I; Movie EV1). Thus, WDR62 granules were highly dynamic and underwent very rapid assembly/disassembly under permissive conditions.

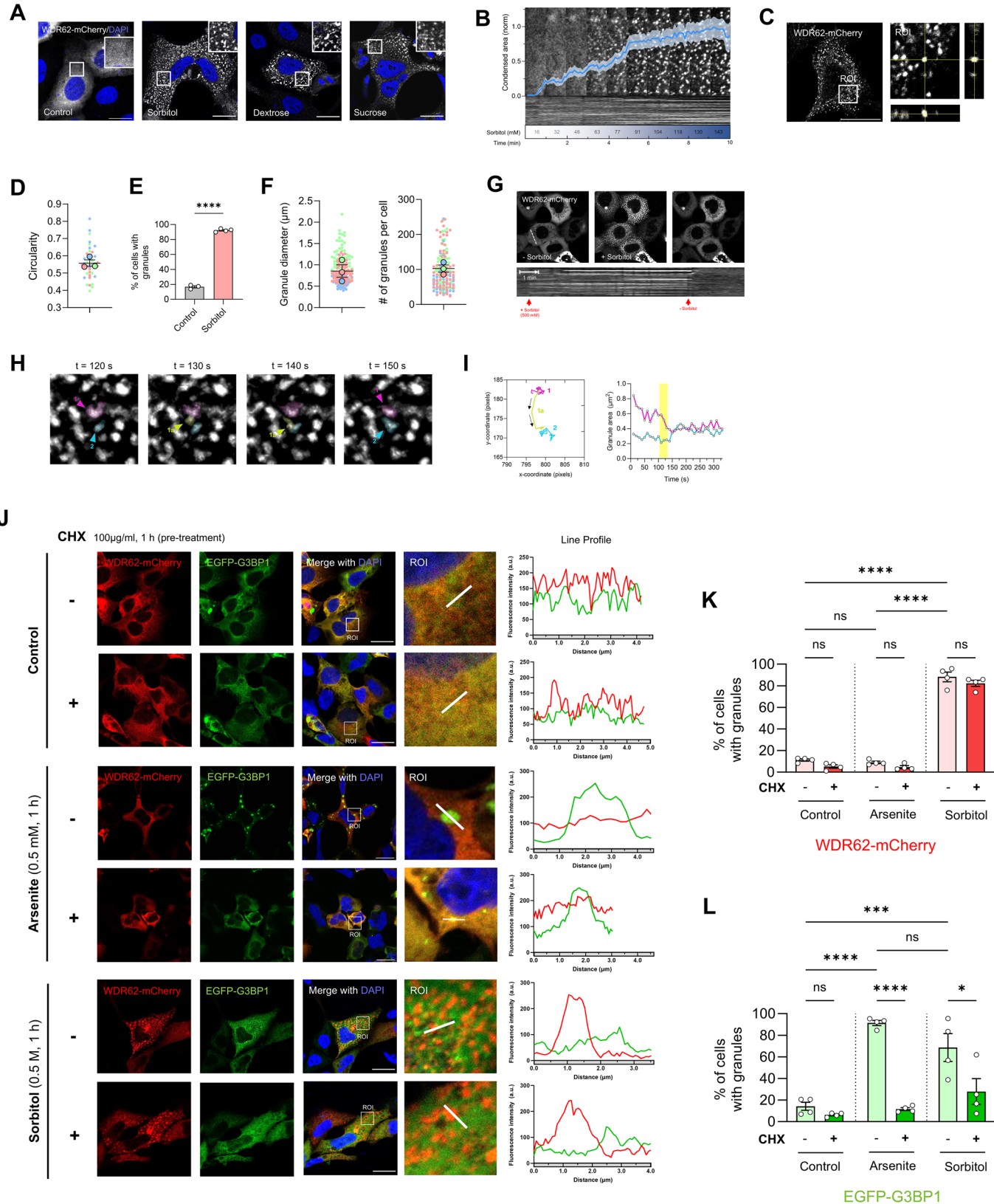

**Figure 3.  Stress-induced assembly of cytoplasmic WDR62 granules distinct to G3BP1 granules.**

(**A**) Representative images of AD293 cells expressing WDR62-mCherry, showing WDR62 de-mixes into cytoplasmic puncta following treatment with 0.5 M sorbitol, dextrose, or sucrose for 1 h. (**B**) Titration of WDR62 granule assembly. A concentrated solution of sorbitol was added dropwise to a live-cell culture dish containing AD293 cells expressing WDR62-mCherry. Sorbitol was added every minute for the duration of the experiment (10 min). Total sorbitol concentration in culture media is displayed in the figure. The blue line represents a quantification of the total WDR62 condensation area over time. Condensed area is the total sum of WDR62-mCherry expression, normalised by its highest value. Scale bar represents 5 μm. (**C**) Three-dimensional orthogonal projection of WDR62 granules. (**D**) Average circularity of WDR62 granules. A circularity of 1 is equivalent to a perfect circle. (**E**) The average proportion (%) of cells containing WDR62 granules before and after sorbitol stress. Two-tailed unpaired Student's *T* test (****$P < 0.0001$). (**F**) The average diameter (μm) and number per cell of WDR62 granules. (**G**) Live-cell imaging experiments reveal that WDR62 granules rapidly assemble (~1 min) and disassemble (~10 s) following the addition and removal of sorbitol stress. Bottom represents the kymograph corresponding to the white dashed line on the first image. (**H**) WDR62 granules undergo fission and fusion events. (**I**) The left graph shows the trajectory of both granules over the time-lapse experiment, the right graph shows the area of both granules before (0–100 s), during (100–125 s) and after merging (125–350 s). (**J**) AD293 cells transiently transfected with WDR62-mCherry and stress granule marker, EGFP-G3BP, pre-treated with 100 μg/ml cycloheximide (CHX) ( + ) or vehicle control (−) for 1 h. Cells were subsequently treated with 0.5 mM sodium arsenite or 0.5 M sorbitol. Representative confocal micrographs demonstrate minimal co-localisation between WDR62-mCherry and G3BP-EGFP, indicated by fluorescence intensity plots (*y*-axis represents fluorescence intensity (a.u.), *x*-axis (μm) represents length of white line drawn on ROI image). (**K**) Bar graph representing the proportion (%) of cells containing WDR62 granules following treatments in (**J**). (****$P < 0.0001$). (**L**) Bar graph representing the proportion (%) of cells containing G3BP1 granules following treatments in (**J**). (*$P = 0.0115$, ***$P = 0.0007$, ****$P < 0.0001$). Data represent mean ± SEM of $n = 4$ independent replicates. *P* values calculated based on mean values using a one-way ANOVA (*$P < 0.05$, **$P < 0.005$, ***$P < 0.001$, ****$P < 0.0001$, n.s. is $P > 0.05$). All scale bars represent 20 μm. Source data are available online for this figure.

To determine if sorbitol-induced WDR62 granules represented conventional SGs, which are characteristically nucleated following the release of messenger ribonucleoproteins (mRNPs) from polysomes, we assessed WDR62 granule formation following treatment with the translation elongation inhibitor, cycloheximide (CHX), which is known to trap mRNPs in polysomes and selectively inhibit SG formation (Mollet et al, 2008). In AD293 cells co-expressing WDR62-mCherry and EGFP-G3BP1, arsenite treatment induced the formation of EGFP-G3BP1 granules but had minimal effect on WDR62 (Fig. 3J–L). In addition, we observed minimal overlap between WDR62 and G3BP1 in the context of either sorbitol or arsenite treatment (Fig. 3J), consistent with previous reports (Wasserman et al, 2010). Importantly, CHX treatment did not prevent sorbitol-induced WDR62 granule assembly (Fig. 3K) but abolished G3BP1-positive SG formation induced by arsenite and sorbitol (Fig. 3L). Thus, WDR62 granules are unlike conventional SGs as they do not require the release of RNPs from stalled polysomes and do not overlap with SG markers such as G3BP1.

## WDR62 co-localises with enzymes and co-chaperones involved in purine metabolism

Given WDR62 interactions with BAG2, we next explored co-localisation of WDR62 with HSP co-chaperones by co-expressing EGFP-tagged BAG2, STIP1 or DNAJC7 with WDR62-mCherry in AD293 cells. We found BAG2 and WDR62 were co-localised, particularly in cytoplasmic granules assembled following sorbitol treatment (Fig. 4A). Similarly, STIP1 and DNAJC7 were also co-located in WDR62 granules formed in response to osmotic stress (Fig. 4B,C). The vast majority of WDR62 granules were comprised of these co-chaperones with a significant overlap of fluorescence under stress-stimulated conditions indicated by a Pearson's correlation coefficient (PCC) of 0.9, 0.79 and 0.95 for BAG2, STIP1 and DNAJC7, respectively (Fig. 4H).

HSP70/90 chaperones and co-chaperones, including BAG2, STIP1 and DNAJC7, co-cluster and are reported to be involved in the assembly of purinosomes, biomolecular condensates involved in purine metabolism (French et al, 2013). Therefore, WDR62 interaction and co-localisation with these molecular co-chaperones suggest that WDR62 granules may overlap with

purinosomes. We demonstrated that metabolic enzymes involved in de novo purine biosynthesis (DNPB) that assemble into purinosomes, such as PFAS (phosphoribosylformylglycinamidine synthase), PPAT (phosphoribosyl pyrophosphate amidotransferase), GART (glycinamide ribonucleotide transformylase) and PAICS (phosphoribosylaminoimidazole carboxylase), were all co-localised with WDR62 granules following sorbitol treatment (Fig. 4D–G), with the fluorescence of these purine enzymes tagged with EGFP in close spatial correlation with mCherry-tagged WDR62 (Fig. 4H). Moreover, we showed that WDR62 also localised to purinosomes induced by purine-depleted media, independent of sorbitol treatment (Fig. 4I). Thus, WDR62 was assembled into multicomponent structures that contain DNPB enzymes and their associated chaperone machinery. In support of this, we show that myc-tagged PFAS was co-immunoprecipitated with HA-tagged WDR62, suggesting they form a complex (Fig. 4J).

To confirm interactions between endogenous proteins, we employed proximity ligation assays (PLA) to visualise WDR62 and quantify its association with BAG2 and PFAS in situ within the cytosol. Under basal and sorbitol-treated (0.5 M, 1 h) conditions, we observed abundant PLA signals ( > 100 per cell) that indicated endogenous WDR62 interactions with BAG2 and PFAS (Fig. 4K). Our control experiments with single antibodies confirmed the specificity of detected interactions (Appendix Fig. S4). PLA signals were predominantly cytoplasmic and were nuclear-excluded. This indicated that these WDR62 interactions were localised to the cytoplasm, consistent with our fluorescence co-localisation studies. Notably, sorbitol treatment did not stimulate a significant increase in WDR62/BAG2 or WDR62/PFAS signals (Fig. 4K,L).

Our studies demonstrate that WDR62 interacts with and is co-localized with HSP co-chaperones and enzymes involved in purine metabolism. WDR62 appears to interact constitutively with these partner proteins and likely responds to stress-stimulation by redistribution to cytoplasmic granules.

## WDR62 loss sensitises cells to purine depletion and disrupts purine metabolism

WDR62 is associated with protein partners that are known to phase separate and form metabolons known as purinosomes (An et al,

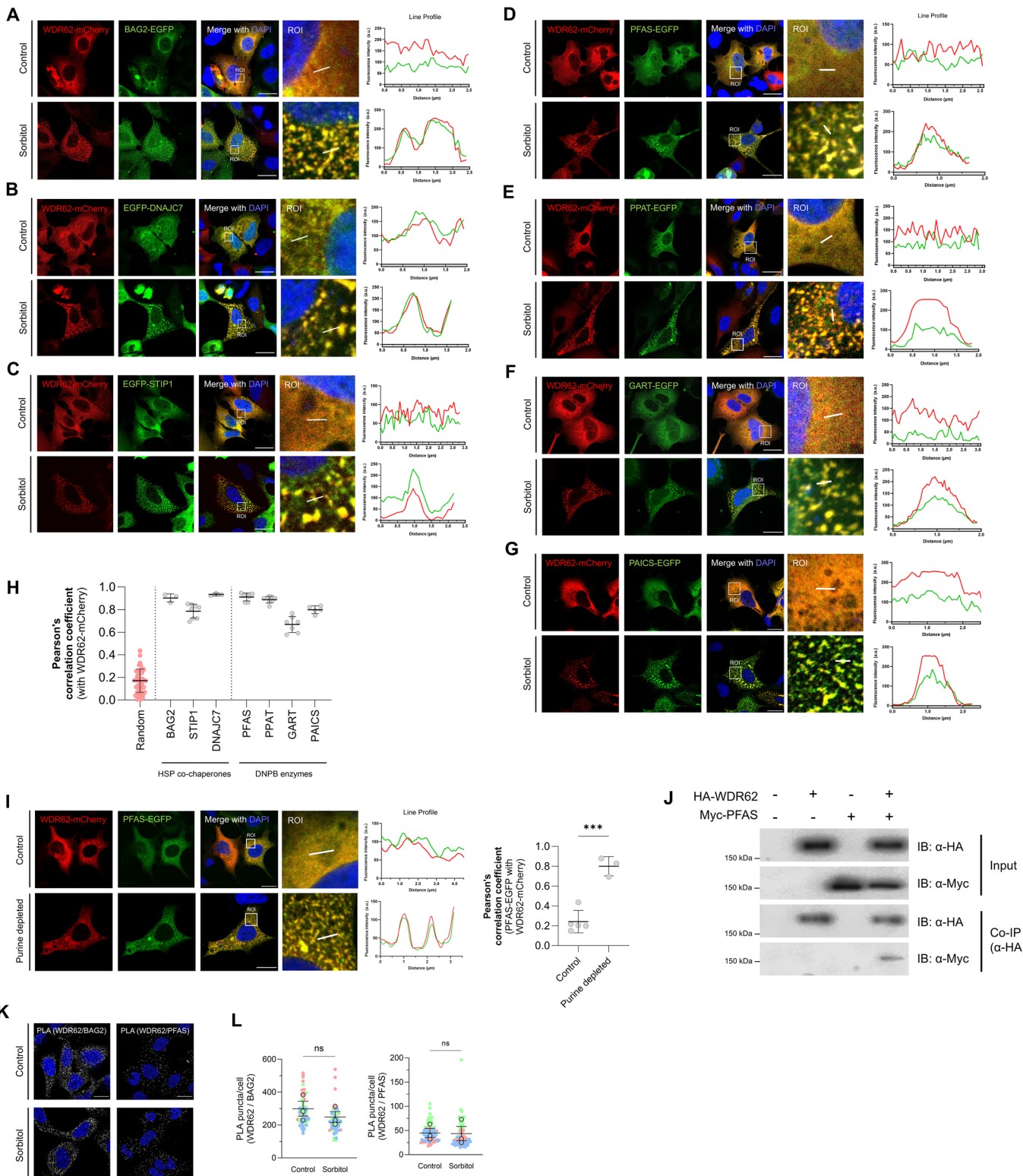

2008; French et al, 2013). Like purinosomes, we observed that WDR62 granules exhibited close spatial association with mito-chondria and microtubules (French et al, 2016) (Appendix Fig. S5). This raises the question of the role of WDR62 in the regulation of

purine metabolism. Compared to wild-type (WT) AD293 cells, the CRISPR/Cas9 deletion of WDR62 (WDR62 KO) did not alter the formation, number, or morphology of granules containing PFAS and PPAT—core enzymes required for DNPB and established

◀ **Figure 4. HSP70/HSP90 co-chaperones and de novo purine biosynthesis enzymes localise to WDR62 granules.**

Representative confocal micrographs of AD293 cells treated with 0.5 M sorbitol for 1 h. Cells transiently transfected with WDR62-mCherry and either the HSP co-chaperones (**A**) BAG2-EGFP, (**B**) EGFP-DNAJC7, (**C**) EGFP-STIP1, or de novo purine biosynthesis (DNPB) enzymes (**D**) PFAS-EGFP, (**E**) PPAT-EGFP, (**F**) GART-EGFP or (**G**) PAICS-EGFP. Co-localisation of each signal is indicated by fluorescence intensity plots to the right of each set of images (y-axis represents fluorescence intensity (a.u.), x-axis (μm) represents the length of the white line drawn on the ROI). (**H**) Bar graph representing the co-localisation between WDR62-mCherry and EGFP signal for each respective protein (mean ± SD). Each dot represents the Person's correlation coefficient for a single ROI. (**I**) Confocal micrographs of AD293 cells co-transfected with WDR62-mCherry and PFAS-EGFP, in either control (top) or purine-depleted (bottom) conditions. Bar graph on the right represents co-localisation between WDR62-mCherry and PFAS-EGFP (mean ± SD) (***$P$ = 0.0004). (**J**) Co-immunoprecipitation and immunoblot of myc-PFAS and HA-WDR62. (**K**) The association of endogenous WDR62 with BAG2 and PFAS as measured by PLA (grey spots) with DAPI-stained nuclei in blue. PLA signal is observed under basal (control) and sorbitol-treated (0.5 M, 1 h) conditions. (**L**) Quantification of the number of PLA puncta per cell for WDR62/BAG2 and WDR62/PFAS. Data represent $n = 3$ biological replicates. $P$ values calculated based on mean values using a one-way ANOVA for (**H**) and a two-tailed unpaired $T$ test for (**I, L**) (*$P < 0.05$, **$P < 0.005$, ***$P < 0.001$, ****$P < 0.0001$, n.s. is $P > 0.05$). All scale bars represent 20 μm. Source data are available online for this figure.

markers of purinosomes—induced by sorbitol treatment or by purine-depletion (Appendix Fig. S6A,B). Furthermore, WDR62 deletion did not affect the localisation of PFAS granules with microtubules or mitochondria (Appendix Fig. S7). Thus, our results indicate that WDR62 was not essential for the formation and microtubule association of purinosomes. We next moved to determine if WDR62 was functionally involved in the regulation of purine metabolism and purine-dependent cell growth/survival.

First, we examined WT cell numbers compared to WDR62 KO cells when cultured in purine-depleted media. An immunoblot analysis confirmed the loss of WDR62 expression in KO cells (Fig. 5A). In purine-rich (basal) conditions, there was no difference in the number of WT and WDR62 KO cells as they expanded over 7 days (Fig. 5B). In purine-depleted media, WT cell numbers increased in culture but not to the same extent as purine-rich media (Fig. 5B). In comparison, the number of WDR62 KO cells did not increase after 4 and 7 days in culture in purine-depleted media and were significantly reduced compared to WT cells (Fig. 5B). With purine-depletion, we observed a significantly higher percentage of rounded WDR62 KO cells compared to WT at 4 days ($31.83 \pm 3.31\%$ vs $12.82 \pm 2.04$, **$P \leq 0.01$) and 7 days ($48.99 \pm 1.64\%$ vs $23.54 \pm 2.67$, **$P \leq 0.01$) in culture indicating reduced cell viability (Fig. 5C). To further assess cell viability, we performed an LDH release assay, an indicator of cell damage or death, which revealed a ~2.4-fold increase in LDH release in WDR62 KO cells compared to WT cells under purine-depleted conditions ($41.08 \pm 7.45\%$ vs $17.89 \pm 1.87\%$, Fig. 5D).

To determine if WDR62 depletion alters purine metabolites, we cultured cells in purine-depleted media and analysed metabolite extracts from WDR62 KO compared to WT cells with untargeted liquid chromatography–tandem mass spectrometry (LC-MS/MS). In the absence of WDR62, this steady-state analysis indicated a modest reduction in nucleotides such as IMP, GMP and derivatives such as hypoxanthine and the accumulation of nucleosides such as inosine, xanthosine, and guanosine, which may reflect an imbalance in purine synthesis/salvage or increased nucleotide degradation (Appendix Fig. S8). We next investigated, by qPCR, the mRNA expression of a panel of enzymes in purine metabolic pathways. In WDR62 KO cells, we found a downregulation of enzymes involved in purine synthesis such as *PRPS1* (~48% reduction), *PFAS* (~20% reduction) and *IMPDH2* (~22% reduction), the purine salvage gene *HPRT1* (~39% reduction), and the purine degradation gene *XDH* (~64% reduction) (Fig. 5E).

Given the effect of WDR62 loss on purine enzyme mRNA expression and steady state metabolite levels, we next performed a targeted metabolomic flux analysis to evaluate whether the activity of purine synthesis and salvage pathways was changed following WDR62 loss. First, we utilised a targeted LC/MS-MS approach to measure incorporation of the $^{13}C_2$-glycine precursor into purine nucleotides through the de novo synthesis pathway (Fig. 5F). In purine-rich media, WDR62 KO cells showed significantly reduced labelling of multiple purine nucleotides (AMP, ADP, ATP, GDP, GTP) (Fig. 5F). Interestingly, this effect was not observed under purine-depleted conditions where both parental and WDR62 KO cells upregulated de novo synthesis to a similar extent (Appendix Fig. S9A). Thus, while WDR62 exerts influence on purine synthesis, it is not required for compensatory upregulation of DNPB when challenged with purine depletion. Consistent with this, WDR62 loss did not affect the assembly and distribution of purinosomes (Appendix Figs. S6 and S7), which are involved in enhancing DNPB during purine depletion (An et al, 2008). Next, we examined the purine salvage pathway by measuring the incorporation of $^{13}C_5$-hypoxanthine in purine metabolites, finding no overt differences between WT and KO cells, suggesting that incorporation via the salvage pathway was also not grossly altered under these conditions (Fig. 5G; Appendix Fig. S9B). In all, WDR62 loss impaired purine synthesis under purine-rich conditions and sensitised cells to purine depletion. These results suggest that WDR62 plays a role in maintaining purine metabolic capacity and may participate in feedback regulation of purine pathway activity.

## WDR62 regulates the expression and stability of HPRT

While the flux of metabolites through the purine salvage pathway did not require WDR62, we observed that the expression of HPRT, a purine recycling enzyme, was substantially reduced (~49%) in WDR62 KO cells (Fig. 6A) and this was rescued by expression of full-length WDR62-EGFP (Fig. 6B). While the mRNA levels of *HPRT1* and several enzymes involved in purine synthesis, such as *PFAS*, were downregulated in WDR62 KO cells (Fig. 5E), the degree of protein loss led us to investigate whether post-translational mechanisms and proximity partners of WDR62 involved in protein folding or stability (Fig. 1), such as BAG2, might also regulate HPRT. To determine if post-translational mechanisms contributed to WDR62-dependent loss of HPRT, we performed a cyclohexamide (CHX) chase experiment to evaluate HPRT half-life (Fig. 6C). Following inhibition of protein synthesis with CHX, we found more rapid degradation of HPRT in WDR62 KO cells compared to WT (Fig. 6C). However, the degradation of the purine synthesis enzyme PFAS was not appreciably increased in WDR62 KO cells (Fig. 6C). Thus, WDR62 is required to maintain the stability of HPRT.

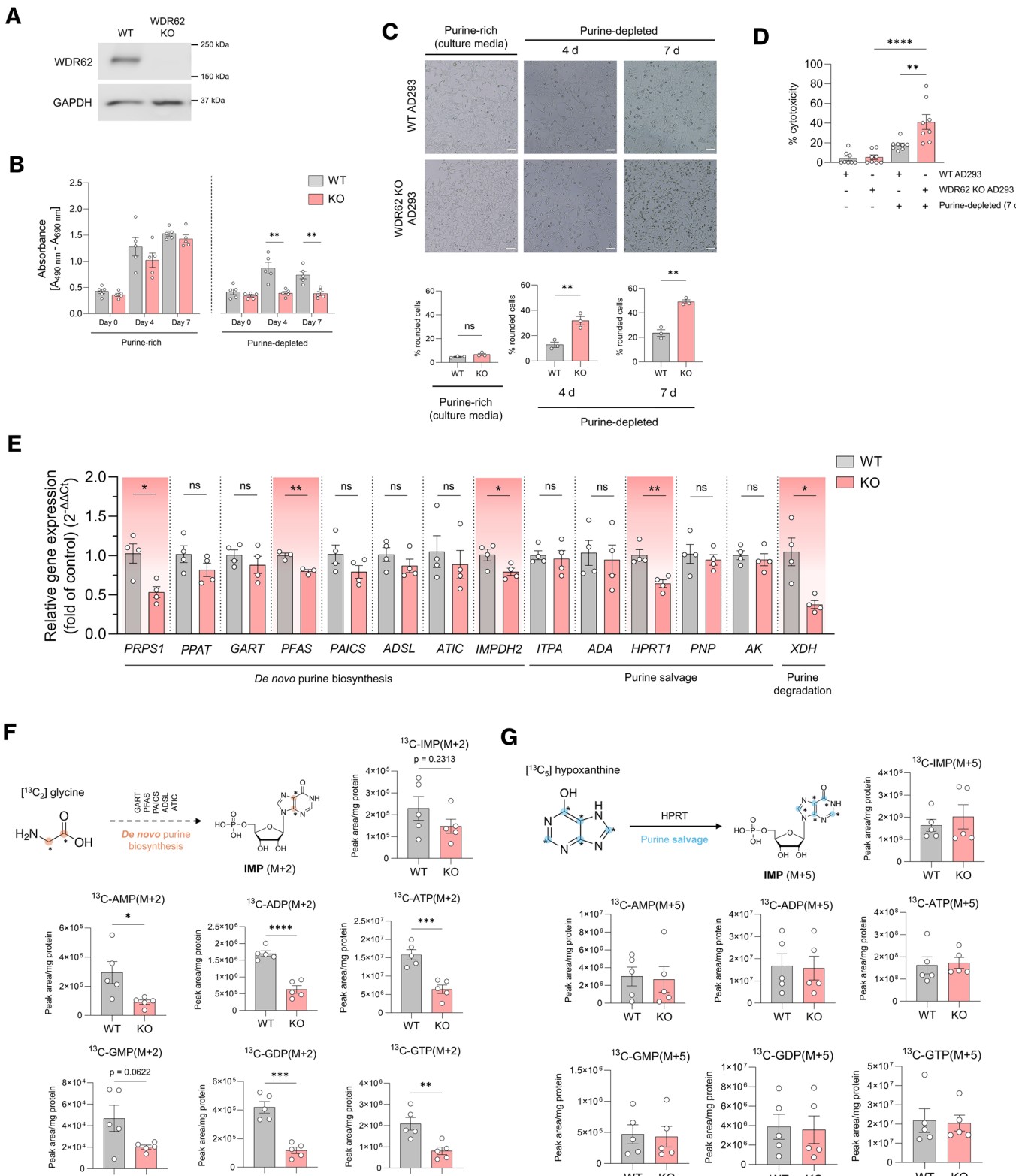

In the absence of WDR62, we also observed altered subcellular localisation of endogenous HPRT (Fig. 6D). Notably, HPRT expression in the cytoplasm of WDR62 KO cells was less diffuse and exhibited a more aggregated/fibrillar appearance (Fig. 6D,E). In a line scan analysis, HPRT fluorescence exhibited greater signal

standard deviation (normalised to line length) in WDR62 KO cells compared to WT cells, which suggests increased propensity for HPRT to aggregate when WDR62 is lost (Fig. 6E). Since HSP70 and HSP90 are known to regulate the stability and folded state of many purine metabolic enzymes (Pedley et al, 2018; Pedley et al, 2022)

**Figure 5.  WDR62 regulates cellular purine metabolism.**

(**A**) Western blot confirming the loss of WDR62 expression in WDR62 KO cells. (**B**) XTT assay quantification of cell proliferation ($A_{490nm}$-$A_{690nm}$) of WT and KO cells cultured for 0, 4 and 7 days in purine-rich or purine-depleted media. Day 4 (**$P = 0.0028$), Day 7 (**$P = 0.0024$). (**C**) Representative images of WT and WDR62 KO cells cultured in purine-rich or purine-depleted media. Bar graphs depict the proportion (%) of rounded cells in each category. Day 4 (**$P = 0.0081$), Day 7 (**$P = 0.0013$). (**D**) LDH assay of percentage cytotoxicity (% cytotoxicity) of WT and KO cells cultured for 7 days in purine-rich or purine-depleted media (**$P = 0.0027$, ****$P < 0.0001$). (**E**) qPCR of WDR62, de novo purine biosynthesis (*PRPS1, PPAT, GART, PFAS, PAICS, ADSL, ATIC*) purine salvage (*ITPA, ADA, HPRT1, IMPDH2, AK*) and purine degradation (*XDH*) gene expression in WT and WDR62 KO AD293 cells. PRPS1 (*$P = 0.0133$), PFAS **($P = 0.0066$), IMPDH2 (*$P = 0.049$), HPRT1 (**$P = 0.005$), XDH (*$P = 0.0109$). (**F**) Schematic of incorporation of $^{13}$C from $^{13}$C-glycine into the purine ring. Bar graphs of normalised peak areas of $^{13}$C[M + 2] labelled purine metabolites, as measured by LC-MS/MS in WT and KO AD293 cells when cultured in purine-rich conditions. $^{13}$C-AMP[M + 2] (*$P = 0.0296$), $^{13}$C-ADP[M + 2] (****$P < 0.0001$), $^{13}$C-ATP[M + 2] (***$P = 0.0008$), $^{13}$C-GDP[M + 2] (***$P = 0.0002$), $^{13}$C-GTP[M + 2] (**$P = 0.0062$). (**G**) Schematic of incorporation of $^{13}$C from $^{13}$C-hypoxanthine into the purine ring. Bar graphs of normalised peak areas of $^{13}$C[M + 5] labelled purine metabolites, as measured by LC-MS/MS in WT and KO AD293 cells when cultured in purine-rich conditions. Asterisks denote statistical significance (two-tailed unpaired *T* test) in comparison of WT and KO groups for each target gene. Data represent $n = 8$ independent replicates for (**B–D**), $n = 4$ for (**E**), and $n = 5$ for (**F, G**). *P* values calculated based on mean values using a one-way ANOVA for (**B–D**) or two-tailed unpaired *T* tests for (**E–G**). (*$P < 0.05$, **$P < 0.005$, ***$P < 0.001$, ****$P < 0.0001$, n.s. is $P > 0.05$). All data represent mean ± SEM. Scale bars represent 100 μm. Source data are available online for this figure.

and given WDR62 binding of chaperone machinery, we postulated that HPRT aggregation may be linked to a dysregulation of HSP70/90 chaperone activity. To investigate this, we treated cells with 17-AAG and NVP-AUY922, small-molecule inhibitors of HSP70 and HSP90, respectively. Interestingly, this reversed the fibrillar morphology of HPRT in WDR62 KO cells (Fig. 6F,G). We confirmed this effect with a second HSP70 inhibitor, MKT-077 (Appendix Fig. S10A,B). This indicates that the stability and cytoplasmic distribution of HPRT is regulated by HSP70/90, and the cytoplasmic aggregation of HPRT, associated with WDR62 loss, is due to an imbalance or excessive HSP70/90 activity.

## Cellular HPRT levels are regulated by WDR62-BAG2

BAG2 is a co-factor for molecular chaperones such as HSP70 but is also associated with HSP90 (Gano and Simon, 2010). In certain pathological contexts, BAG2 overexpression has been implicated in the misfolding and aggregation of cytoplasmic proteins in a HSP90-dependent manner (Huang et al, 2023). In addition to interactions with BAG2 (Fig. 2), we showed that exogenously expressed WDR62 co-immunoprecipitated with HSP70 but not HSP90 (Appendix Fig. S11A–C). We found that endogenous WDR62 interacted with HSP70 and HSP90 in PLAs (Appendix Fig. S11D,E). These interactions were not altered with HSP70 or HSP90 activity inhibition (Appendix Fig. S11B,D,E). Thus, WDR62 forms a complex with heat shock protein chaperone,s and this does not appear to be due to chaperone-client protein interaction.

Despite WDR62 interaction with BAG2, we find that BAG2 knockdown did not prevent and was not required for stress-induced assembly of WDR62 or PFAS/PPAT granules (Appendix Fig. S6C–E). We next sought to determine if WDR62 regulated HPRT in a BAG2-dependent manner. In cells lacking WDR62, we observed a ~2.6-fold increase in BAG2 expression, accompanied by reduced HPRT levels (Fig. 6A). Similarly, transient depletion of WDR62 with siRNA led to increased BAG2 levels (Appendix Fig. S12A). BAG2 overexpression was also reversed by transfecting full-length WDR62 in WDR62 KO cells (Fig. 6B). Interestingly, the expression of the R438H or 3936dupC mutants of WDR62, which displayed altered binding to BAG2 (Fig. 2E), did not rescue the HPRT levels in WDR62 KO cells (Fig. 6B). We also assessed the expression of other co-chaperones, including STIP1 and DNAJC7, both of which co-localised in WDR62 granules (Fig. 5A–C), as well

as HSP70 and HSP90. The expression of these chaperones remained unchanged in WDR62 KO cells (Appendix Fig. S12B).

To investigate the mechanistic relationship between WDR62, HPRT and chaperones, we also examined whether HPRT co-localised with WDR62 granules induced by osmotic stress. Following sorbitol treatment, endogenous HPRT adopted a punctate distribution, with puncta that juxtaposed and co-localised with WDR62 granules, with a modestly increased interaction compared to untreated cells (Fig. 6H). To further examine their relationship, we performed proximity ligation assays (PLA) to assess direct protein interactions. Indeed, we confirmed an interaction between endogenous WDR62 and HPRT, which increased in response to sorbitol treatment (Fig. 6I). WDR62-HPRT interactions were not altered by HSP70/90 inhibition, but HSP70 binding to HPRT trended higher with HSP70 activity inhibition but did not reach statistical significance (Appendix Fig. S12C,D). Thus, WDR62-HPRT complex formation does not require protein folding activity, but HPRT may represent a client of HSP70. We next sought to determine the role of BAG2 in regulating WDR62/HPRT binding. We found that depletion of BAG2 with siRNA impaired WDR62 interaction with HPRT (Fig. 6J). Similarly, WDR62 KO trended towards reduced HPRT/BAG2 interaction, with an ~32% reduction in PLA signal per cell, although this did not reach statistical significance (Fig. 6K). This suggests that WDR62, BAG2 and HPRT form an interaction network, which depends on WDR62 and BAG2 expression, and the loss of WDR62 or BAG2 may underlie the altered stability of HPRT in this context.

To investigate this, we utilised siRNA to knockdown the elevated levels of BAG2 in WDR62 KO cells to determine if BAG2 contributed to the diminished half-life of HPRT in the absence of WDR62. Compared to WT levels, HPRT levels were significantly reduced in WDR62 KO cells transfected with non-targeting siRNA, while siRNA depletion of BAG2 in WDR62 KO cells restored HPRT to WT levels (Fig. 6L). In contrast, depletion of BAG2 did not rescue the aggregated appearance of HPRT in WDR62 KO cells (Appendix Fig. S10C). Thus, while BAG2 depletion rescued HPRT levels, it did not reverse the aggregated appearance of HPRT. However, we note that BAG2 siRNA reduced BAG2 levels to substantially below WT cells (Fig. 6I). It remains possible that an imbalance of BAG2 relative to HSP70/90 underlies disrupted folding and cytosolic distribution of HPRT.

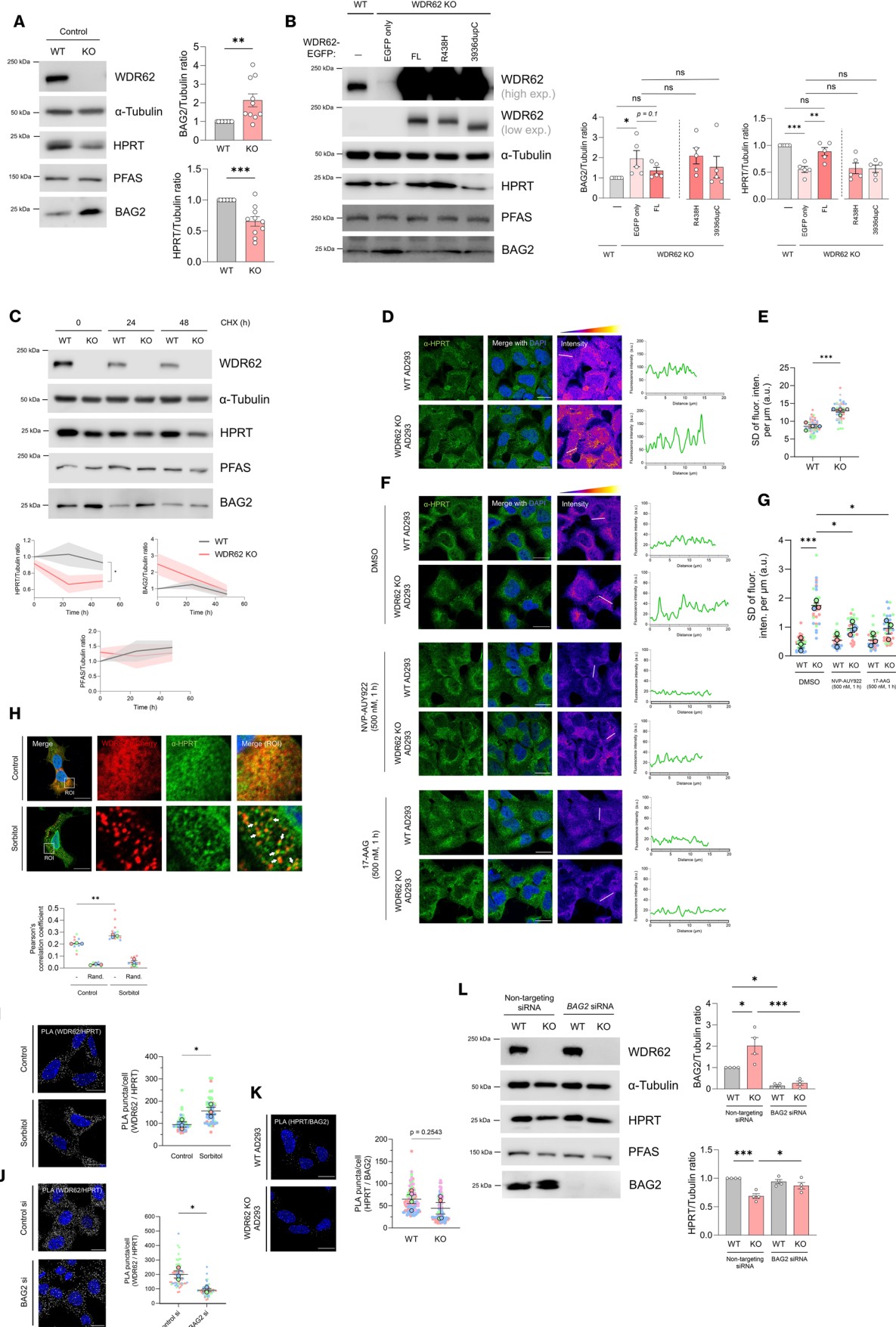

◀

**Figure 6.   WDR62 depletion causes BAG2-dependent loss of purine metabolic enzymes.**

(A) Immunoblot of purine metabolic enzymes PFAS and HPRT.and co-chaperone BAG2 in WT and WDR62 KO AD293 cells. BAG2 (**$P = 0.0046$), HPRT (***$P = 0.0003$).
(B) Immunoblot of WDR62 KO AD293 cells rescued with EGFP only, WDR62(FL)-EGFP, or mutants WDR62(R438H)-EGFP and WDR62(3936dupC)-EGFP (*$P = 0.0464$, **$P = 0.0026$, ***$P = 0.0002$). (C) Cycloheximide chase assay examining the half-life of HPRT, PFAS and BAG2 in WT and WDR62 KO cells. WT and WDR62 KO cells treated with 100 μM CHX for 0, 24, or 48 h (*$P = 0.0196$). (D, F) Immunofluorescence and confocal micrographs of endogenous HPRT in WT and WDR62 KO cells treated with vehicle (DMSO), or the HSP90 inhibitors NVP-AUY922 or 17-AAG (500 nM, 1 h). Far right image represents pseudo-coloured "fire" LUT to better visualise pixel intensity. Fluorescence intensity plots to the right of each set of images (*y*-axis represents fluorescence intensity (a.u.), *x*-axis (μm) represents the length of the white line drawn on pseudo-coloured intensity image. (E, G) SuperPlots of averaged normalised (per μm) standard deviation (SD) of fluorescence intensity (a.u.) along random lines plotted in the cytoplasm of random cells. Higher SD values indicate greater heterogeneity in fluorescence intensity, characteristic of a less diffuse and more punctate distribution. WT vs. KO untreated (***$P = 0.0004$), WT vs. KO DMSO (***$P = 0.0002$), KO DMSO vs. KO NVP (*$P = 0.0151$), KO DMSO vs. KO AAG (*$P = 0.0157$). (H) Immunofluorescence and confocal micrographs of endogenous HPRT with WDR62-mCherry in control and sorbitol-treated (0.5 mM, 1 h) cells (**$P = 0.001$). (I–K) The association of endogenous WDR62 with HPRT, or endogenous HPRT with BAG2 as measured by PLA (grey spots) with DAPI-stained nuclei in blue. PLA signal is observed under control or sorbitol-treated conditions for (I) (*$P = 0.0385$), or control siRNA or BAG2 siRNA conditions for (J) (*$P = 0.0147$), and WT and WDR62 KO conditions for (K). Quantification of the number of PLA puncta per cell represented in SuperPlots on the right of the images. (L) Western blot analyses of WDR62, BAG2, HPRT and PFAS expression in WT and WDR62 KO cells treated with non-targeting or *BAG2* siRNA. IB: BAG2, non-targeting siRNA, WT vs. KO (*$P = 0.017$), IB: BAG2, WT cells, non-targeting vs. BAG2 siRNA (*$P = 0.0469$), IB: BAG2, KO cells, non-targeting vs. BAG2 siRNA (***$P = 0.0002$), IB: HPRT, non-targeting siRNA, WT vs. KO (***$P = 0.0006$), IB: HPRT, KO cells, non-targeting vs. BAG2 siRNA (*$P = 0.0331$). Data represent $n = 10$ independent replicates for (A), $n = 5$ for (B), $n = 9$ for (C), $n = 4$ for (D, E, K, L), and $n = 3$ for (F, G, H, I, J). *P* values calculated based on mean values using a one-way ANOVA for (B, C, G, H, L) or a two-tailed unpaired *T* test for (A, E, I, J, K) (*$P < 0.05$, **$P < 0.005$, ***$P < 0.001$, ****$P < 0.0001$, n.s. is $P > 0.05$). All scale bars represent 20 μm. All data represent mean ± SEM. Source data are available online for this figure.

## WDR62 and HPRT regulate neural progenitor cells during neocortex development

Given that HPRT deficiency is reported to cause neurodevelopmental disorders, our finding that WDR62 regulates HPRT levels suggests a mechanism by which WDR62 may influence brain growth (Guibinga et al, 2013; Bilgüvar et al, 2010; Yu et al, 2010; Nicholas et al, 2010). To investigate the role of WDR62 and HPRT in cortical development, we electroporated experimentally validated siRNAs targeting *Wdr62, Hprt*, or a non-targeting control (Fig. 7A) into the embryonic day 14.5 (E14.5) mouse neocortex. Forty-eight hours post-electroporation (E16.5), brains were harvested, and immunohistochemistry was performed to detect proliferative and mitotic markers, Ki67 and PH3, respectively, as well as apical and intermediate progenitor (IP) markers Pax6 and Tbr2. We first assessed the spatial distribution of GFP-positive (GFP⁺) cells in the neocortex and noted that the overall thickness of ventricular and subventricular zones (VZ/SVZ) and cortical plate (CP) was not substantially different between groups (Appendix Fig. S13A). However, compared to control animals, we revealed in both WDR62- and HPRT-depleted brains, a marked reduction of GFP+ cells in the lower VZ/SVZ and enrichment in the upper VZ/SVZ and intermediate zone (IZ) (Fig. 7B). While there was a degree of variability, between animals within the same siRNA group, in the extent of movement of GFP+ cells from the ventricular surface (Appendix Fig. S13B), our findings indicate precocious delamination and differentiation of progenitors following WDR62 or HPRT depletion compared to control animals (Fig. 7B). Notably, WDR62 knockdown significantly reduced Ki67 expression, consistent with premature cell cycle exit and impaired self-renewal, in line with previous studies (Bogoyevitch et al, 2012; Xu et al, 2014) (Fig. 7C,D). In contrast, HPRT depletion led to increased Ki67 levels in GFP+ cells (Fig. 7C,D). Specifically, there was a significant increase in Ki67+ cells in the IZ and the upper region of VZ/SVZ of HPRT-depleted cortices (Fig. 7C,D). We also observed a modest but non-significant reduction in PH3 levels in WDR62-depleted progenitors and an apparent increase in PH3 with HPRT loss within VZ/SVZ regions (Fig. 7C,E). Furthermore, HPRT depletion significantly increased Tbr2+ cells in the IZ and reduced Tbr2-

cells in VZ/SVZ with HPRT loss (Fig. 7F). Given previous reports that GFP-labelled cells that are Tbr2- in the VZ/SVZ are an accurate indicator of apical progenitors (Englund et al, 2005; Di Bella et al, 2021), our findings indicate depletion of HPRT enhanced transition of apical to basal progenitors and differentiation into IPs without a loss in proliferative capacity. Together, these findings suggest that while both WDR62 and HPRT regulate progenitor differentiation, their mechanisms in cortical development diverge such that HPRT loss favours the differentiation of apical progenitors into proliferative IPs, whereas WDR62 loss impairs the self-renewal of neural progenitors, likely via delamination and precise differentiation into neurons. This shows that the loss of WDR62 and HPRT displays overlapping but also divergent functions in regulating the development of cortical progenitors in vivo. These mechanisms highlight a previously unrecognised WDR62 link between proteostasis and purine metabolism, providing insight into how dysregulation of these processes may contribute to congenital microcephaly.

## Discussion

We present evidence of a novel role for WDR62 regulating the expression of purine metabolic enzymes and, through interaction with the BAG2 co-chaperone, the stability and turnover of HPRT, which is involved in purine salvage. We also find that WDR62 forms dynamic granules that co-localise with co-chaperones and purine metabolic enzymes that are known to form purinosomes.

Our studies indicate that BAG2 interacts with the WD-repeat region and the C-terminal α-helical dimerisation domain of WDR62. Furthermore, we show that microcephaly-associated mutations alter WDR62-BAG2 interactions, suggesting that defects in this novel regulatory pathway of purine metabolism may contribute to abnormal neurodevelopment. However, it is unclear why microcephaly-linked missense mutations in WDR62 enhanced BAG2 interactions despite not being contained within the predicted WDR62-BAG2 interface (Fig. 2). It is possible that these specific residues might participate in uncharacterised protein-protein interactions that normally sterically impede on the nearby BAG2

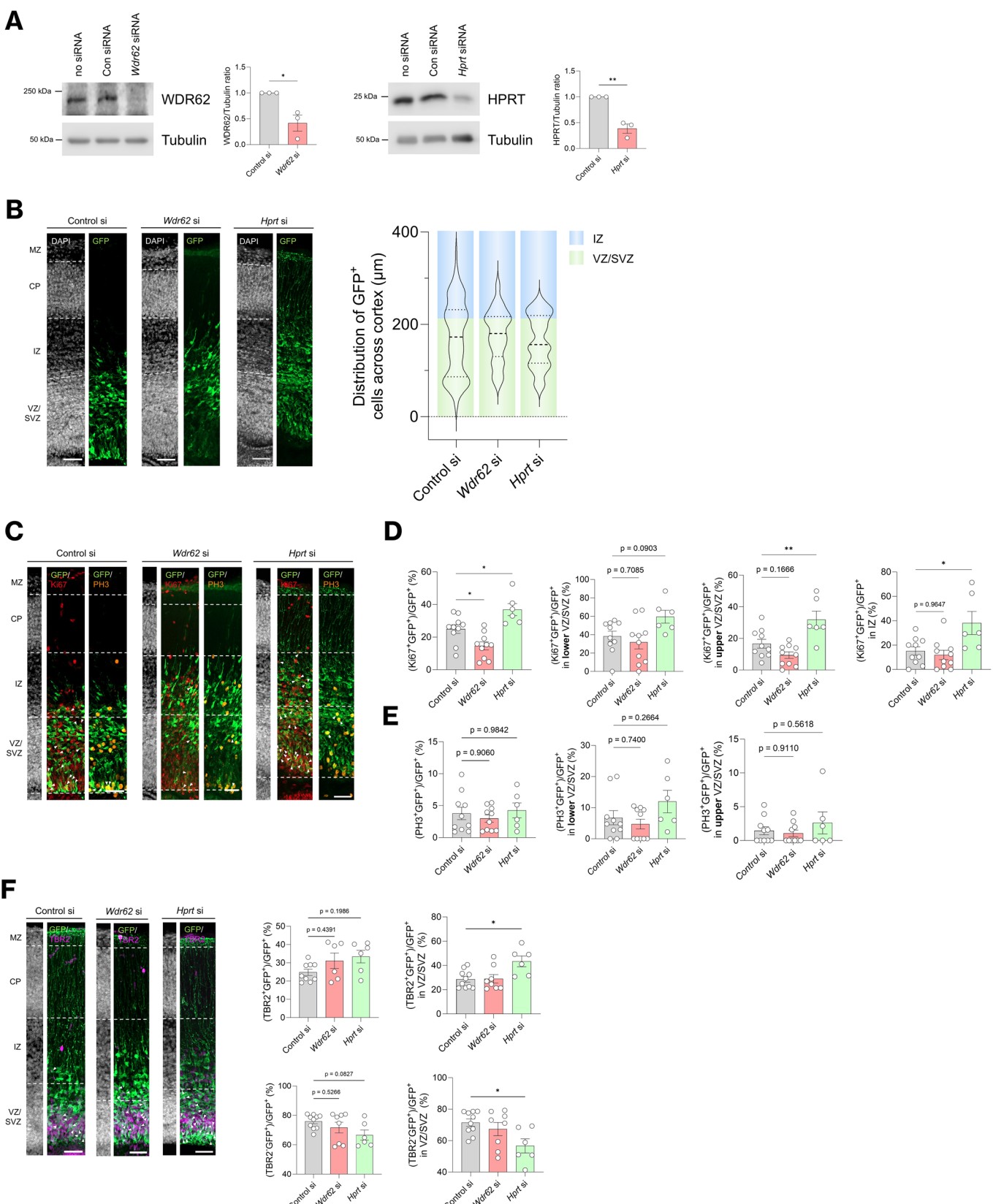

**Figure 7.   HPRT depletion alters cell proliferation, migration, and differentiation in embryonic mouse cortex.**

(**A**) Immunoblots of (left to right) WDR62, HPRT, and BAG2 in Neuro2a cells transfected with no siRNA, non-targeting siRNA, or siRNA against *Wdr62*, *Hprt*, or *Bag2*, respectively. Bar graphs to the right of each immunoblot represent densitometric quantification of signal over $n = 3$ biological replicates (*$P = 0.0199$, **$P = 0.0025$). (**B–D**) In utero electroporation of control siRNA or siRNA targeting *Wdr62*, *Hprt*, or *Bag2* into E14.5 cortices, followed by analysis at E16.5. (**B**) Distribution of electroporated GFP-positive (GFP$^+$) cells in the ventricular zone/subventricular zone (VZ), intermediate zone (IZ), cortical plate (CP), or marginal zone (MZ). (**C**) Immunostaining electroporated brains sections for Ki67 and PH3 as markers of cell proliferation and mitosis respectively in electroporated brain sections. (**D**) Analysis of cell proliferation (Ki67$^+$/GFP$^+$) and (**E**) mitotic (PH3$^+$/GFP$^+$) cells in cortex, IZ and the VZ/SVZ, the latter delineated in half between the region closest to the apical surface of the lateral ventricles (lower) and away from the apical region (upper). (Ki67$^+$/GFP$^+$)/GFP$^+$, control vs. *Wdr62* si (*$P = 0.0275$), (Ki67$^+$/GFP$^+$)/GFP$^+$, control vs. *Hprt* si (*$P = 0.0281$), (Ki67$^+$/GFP$^+$)/GFP$^+$ in upper VZ/SVZ, control vs. *Hprt* si (**$P = 0.0081$), (Ki67$^+$/GFP$^+$)/GFP$^+$ in IZ, control vs. *Hprt* si (*$P = 0.0338$). (**F**) Analysis of cell differentiation into intermediate progenitors (TBR2$^+$/GFP$^+$) and TBR2-negative progenitors (TBR2$^-$/GFP$^+$) in electroporated brain sections. (TBR2$^+$/GFP$^+$)/GFP$^+$ in VZ/SVZ (*$P = 0.0213$), (TBR2$^-$/GFP$^+$)/GFP$^+$ in VZ/SVZ (*$P = 0.0236$). All scale bars represent 50 μm. All data represent mean ± SEM. Source data are available online for this figure.

binding site, thereby potentially enhancing WDR62-BAG2 interactions when mutated. Conversely, we found that frameshift mutations such as 3936dupC, which truncate the C-terminal dimerisation domain while preserving the full N-terminus, disrupt binding to BAG2 (Fig. 2E). This suggests that although the N-terminus can engage with BAG2, the dimerisation domain contained within the disordered C-terminus may stabilise the interaction. Supporting this idea, we observed a trend towards increased WDR62(N) interaction with BAG2 compared to full-length WDR62 (Fig. 2D).

The fact that microcephaly-associated WDR62 mutations can lead to increased or decreased binding of BAG2, depending on the specific mutation, suggests that either excessive or deficient interactions between WDR62 and BAG2 may be detrimental. Consistent with this idea, WDR62 and BAG2 expression levels were feedback-regulated, where the loss of WDR62 leads to an increase in BAG2 (Fig. 6A). We speculate that V65M and R438H missense mutations and increased WDR62-BAG2 interactions may sequester BAG2, disrupting its normal function. In contrast, frameshift mutations may reflect a knockout effect with decreased BAG2 binding and regulation. Hence, an abnormal increase or decrease in WDR62-BAG2 interactions may similarly result in disruptions in purine enzyme turnover and purine homeostasis.

WDR62 forms cytoplasmic granules that co-localise with co-chaperones and purine metabolic enzymes. These granules resembled purinosomes, and we described their assembly in response to hyperosmotic stress, which may trigger protein phase separation through mechanisms such as cell volume exclusion and macromolecular crowding (Gao et al, 2022; Ding et al, 2021). Notably, we found that WDR62 granules assemble rapidly even at low sorbitol concentrations (~50 mM) (Fig. 3B), suggesting that even milder osmotic stress, particularly under prolonged exposure, could drive WDR62 condensation in physiological conditions. WDR62 condensation may serve as a link between osmotic stress responses and protein homeostasis or purine metabolism, though this requires further exploration.

Purinosomes have not previously been associated with osmotic stress and it is unclear if WDR62 granules contain all necessary components of the de novo purine biosynthesis (DNPB) pathway and/or represent functional metabolons that enhance metabolic flux under conditions of purine depletion, hypoxia, or casein kinase 2 (CK2) inhibition (Doigneaux et al, 2020; An et al, 2010; Chan et al, 2015). Moreover, there is evidence of metabolically inactive purine enzyme assemblies that sequester pathway components, hindering bona fide purinosome assembly and metabolic flux (Pedley et al, 2022). Nonetheless, our study confirms that WDR62

granules co-localise with at least four of six DNPB enzymes (Fig. 5D–G), and we observed the presence of WDR62 within granules formed under purine-depletion conditions which may represent bona fide purinosomes (Fig. 5I). However, while HSP70/90, their co-chaperones, and other WD repeat-containing proteins are involved in purinosome assembly (Yamada et al, 2020), we show WDR62 depletion did not prevent the formation of these PFAS/PPAT granules, which catalyse the first committed steps in purine biosynthesis (Appendix Fig. S6). Consistent with this, WDR62 loss reduced purine synthesis under basal conditions but not under purine-depleted conditions, which induce the formation of purinosomes (Fig. 5). This suggests that WDR62 association with PFAS or PPAT was not required for their assembly into purinosome-related granules.

We also find that WDR62 is involved in maintaining purine homeostasis. WDR62 KO cells contained lower levels of enzymes involved in de novo purine biosynthesis, with a reduced rate of purine biosynthesis when cultured in the presence of purines (Fig. 5E,F). Under purine-depleted conditions, we observed an increased rate of purine synthesis overall, and this was not perturbed by WDR62 loss. Thus, WDR62 does not appear to be involved in meeting cellular demands of purines under metabolic load, or alternatively, other mechanisms may increase to compensate for WDR62 loss in response to purine starvation. Since purine synthesis is usually governed by feedback regulation by purine intermediates and end-products, these findings raise the possibility that WDR62 is involved in feedback control of purine biosynthesis enzyme expression under basal conditions and coordinating changes in purine synthesis in response to purine availability.

Under purine-depleted conditions, we observed in WDR62 KO cells an accumulation of nucleosides, such as inosine, guanosine and xanthosine, under steady state (Appendix Fig. S8B). Moreover, metabolic flux analysis indicated that, in the absence of purines, the rates of purine salvage were not substantially altered in the absence of WDR62. This suggests an increased degradation of nucleotides such as IMP, GMP and XMP into their nucleoside analogues with WDR62 loss that is not compensated for, for example, by an increase in purine salvage. It remains to be determined whether this contributes to the increase in WDR62 KO cell death observed under purine-depleted conditions (Fig. 5). Inosine and guanosine nucleosides have a broad spectrum of cellular effects and can be cytotoxic in certain contexts (Batiuk et al, 2001). Moreover, an imbalance of specific nucleotides may also perturb cell cycle progression and attenuate proliferation (Diehl et al, 2022). Our findings raise the possibility that disrupted purine metabolism may contribute to cell death and cell cycle defects caused by WDR62

mutation during human cortical development (Nicholas et al, 2010; Yu et al, 2010; Bilgüvar et al, 2010).

We demonstrated that the WDR62-BAG2 axis was involved in regulating the stability of the purine salvage enzyme, HPRT. The mechanism behind BAG2-mediated HPRT degradation after WDR62 loss is of particular interest. Indeed, BAG2 exhibits dual roles in proteostasis, depending on its relative stoichiometry. At low levels, BAG2 stimulates ATP turnover and enhances chaperone cycling of HSP70-DNAJ pairs, promoting protein folding (Rauch and Gestwicki, 2014). In contrast, high stoichiometric concentrations of BAG2 inhibit ATP turnover, stabilise unbound HSP70, and suppress folding activity, potentially shifting its function toward protein degradation (Rauch and Gestwicki, 2014). We identified endogenous interactions between WDR62, HPRT, BAG2, and associated chaperones. WDR62 is not a client protein as the inhibition of protein folding did not disrupt WDR62 interactions with BAG2/HSP70. Rather, the interaction between HPRT and BAG2 appears to be reduced in WDR62 KO cells (Fig. 6I), suggesting that WDR62 may act as a scaffold to stabilise this interaction. Such a scaffolding role is consistent with established roles for WDR62 in MAPK signal scaffolding and transduction (Cohen-Katsenelson et al, 2011). Loss of WDR62 may disrupt a protective association between HPRT and BAG2, impairing HPRT stability. This disruption could lead to HPRT misfolding, elevated BAG2 levels, and a shift in BAG2 activity toward pro-degradative roles, ultimately driving HPRT destabilisation. The interaction between HSP70-HPRT trended higher with inhibition of HSP70 activity but did not reach statistical significance (Appendix Fig. S12D). Thus, our data does not definitively establish HPRT as a client of HSP70, but this remains a possibility. The precise mechanisms behind how a multi-unit complex comprising WDR62, BAG2 and HSP70 regulates HPRT turnover will require further testing.

Further, increased BAG2 expression following WDR62 loss could alternatively represent a compensatory response to degrade misfolded clients. BAG2 has been shown to facilitate the degradation of large protein aggregates, such as phosphorylated Tau, via ubiquitin-independent proteasomal pathways (Carrettiero et al, 2009; Carrettiero et al, 2022), and to activate autophagy in the context of protein aggregation (Lima et al, 2022; Liang et al, 2020). Supporting this, we observed that HPRT expression is less uniformly distributed in WDR62 KO cells, consistent with a loss of solubility (Fig. 6D). In this study, HSP70/90 inhibition restored the normal distribution of HPRT, also indicating a role for HSP70/90 in proper HPRT folding and stability (Fig. 6F). Indeed, elevated BAG2 expression has also been implicated in the aggregation of mutant p53 and can be rescued by BAG2 knockdown or the inhibition of HSP90 (Huang et al, 2023). Furthermore, HSP70/90 also plays a role in regulating the solubility of other purine metabolic enzymes, including PFAS (Pedley et al, 2022).

The proteolytic mechanism driving the enhanced degradation of HPRT in WDR62 KO cells remains unknown. Many purine metabolic enzymes are known clients of proteasomal degradation pathways (Pedley et al, 2018). Interestingly, another WD repeat protein linked to microcephaly, WDR47, regulates proteostasis by controlling autophagic flux (Kannan et al, 2017). However, the role of the WDR62-BAG2 axis in maintaining purine enzyme stability may extend beyond post-translational regulation to include transcriptional mechanisms. This is supported by the observed

decrease in *HPRT1* mRNA levels in WDR62 KO cells (Fig. 5E). BAG family proteins have been reported to exhibit transcription factor activity, influencing the transcription of diverse clients, including nuclear hormone receptor-related genes (Zeiner and Gehring, 1995; Takayama and Reed, 2001). Additionally, purine intermediates such as AICAR and SAICAR are known to modulate dimeric interactions between transcription factors, regulating purine homeostasis at the transcriptional level (Pinson et al, 2009). Thus, the downregulation of purine pathway genes in WDR62 KO cells may also originate from feedback mechanisms triggered by the accumulation of purine intermediates. Such accumulation could stem from the initial destabilisation of purine enzymes due to impaired BAG2-mediated proteostasis.

Although HPRT expression is reduced in WDR62 KO cells, incorporation of $^{13}C_5$-hypoxanthine into purine nucleotides remained relatively stable in WDR62 KO cells, suggesting that residual salvage activity by HPRT was sufficient to maintain salvage flux. We find that HPRT depletion in vivo disrupts cortical development in a manner that partially overlaps with WDR62 loss but also causes distinct phenotypes. While the loss of *Wdr62* or *Hprt* both resulted in precocious differentiation of apical progenitors from the ventricular surface, their effects on progenitor fate diverged such that the loss of *Hprt* resulted in the generation of IPs with enhanced self-renewal, whereas *Wdr62* loss impaired self-renewal. Patients with Hprt mutations and *Hprt*-deficient mice do not present with severe structural brain abnormalities typically observed with WDR62 mutation or loss (Witteveen et al, 2022; Bilgüvar et al, 2010). While the delamination of apical precursors is an early step towards differentiation, the increased proliferation observed within in utero *Hprt* knockdown is consistent with prior studies of *Hprt* mutant mice (Witteveen et al, 2022), and likely compensates for early delamination to sustain brain growth. Thus, while WDR62-HPRT may regulate apical radial glia, WDR62 has additional functions, including complex functions in purine metabolism independent of HPRT, that are indispensable for normal brain development.

It is also unresolved whether alterations to cortical development in HPRT-depleted progenitors are a direct consequence of disrupted purine salvage. Purines are essential for the rapid proliferation, differentiation, and survival of neural progenitors in cortical development, supporting these processes through nucleotide synthesis, energy metabolism, and purinergic signalling (Mizukoshi et al, 2023; Yamaoka et al, 1997; Liu et al, 2008; Lin et al, 2007; Mastrangelo et al, 2012). Beyond purine metabolism, HPRT deficiency has also been implicated in dysregulated Wnt/β-catenin signalling, which is critical for neural progenitor growth and differentiation in neurodevelopment (Buchman et al, 2011). Notably, HPRT-deficiency results in reduced β-catenin expression and phosphorylation (Mastrangelo et al, 2012; Kang et al, 2011; Witteveen et al, 2022), potentially shifting its function from Wnt signalling to cell adhesion (Bienz, 2005). This suggests that HPRT might influence progenitor positioning, adhesion, or migration through mechanisms extending beyond its canonical role in purine salvage. Deficiencies in purine metabolic enzymes, other than HPRT, disrupt neurogenesis and neural progenitor proliferation and fate decisions. Intriguingly, NACHT and WD repeat domain containing 1 (NWD1), another microcephaly-associated WD repeat protein, interacts with the purine enzyme PAICS to promote purinosome assembly and facilitate proper neuronal differentiation and migration (Yamada et al, 2020). It remains to be determined whether WDR62 regulation of other

purine enzymes contributes to the development of microcephaly associated with WDR62 mutations.

Overall, our study reveals a role for the WDR62-BAG2 axis in maintaining HPRT stability. We find a novel function for WDR62 in regulating purine metabolism that may influence neural progenitor migration, fate and self-renewal. These findings establish a mechanistic link between proteostasis, purine metabolism, and cortical development, all of which are pathways that may contribute to the aetiology of microcephaly in *MCPH2* patients.

# Methods

### Reagents and tools table

| Reagent/resource | Reference or source | Identifier or catalogue number |
|---|---|---|
| **Experimental models** | | |
| CD-1 (*Mus musculus*) | Queensland Brain Institute, University of Queensland | |
| **Recombinant DNA** | | |
| pcDNA3.1-WDR62-BirA-HA | Generated by restriction enzyme cloning. | Backbone was Addgene #36047. |
| pcDNA3.1-BirA-HA | Purchased from Addgene. | #36047 |
| pXJ41-HA-WDR62 | Generated by restriction enzyme cloning. | Backbone was pXJ41-HA empty vector. |
| pEGFP-WDR62(FL) | Generated by restriction enzyme cloning. | Backbone was pEGFP-C3 (Clontech). |
| pEGFP-WDR62(N) | Generated by restriction enzyme cloning. | Backbone was pEGFP-C3 (Clontech). |
| pEGFP-WDR62(C) | Generated by restriction enzyme cloning. | Backbone was pEGFP-C3 (Clontech). |
| pEGFP-WDR62(R438H) | Generated by site-directed mutagenesis of pEGFP-WDR62(FL). | |
| pEGFP-WDR62(V65M) | Generated by site-directed mutagenesis of pEGFP-WDR62(FL). | |
| pEGFP-WDR62(3966dupC) (1–1329) | Generated by site-directed mutagenesis of pEGFP-WDR62(FL). | |
| pWDR62-mCherry | Generated by restriction enzyme cloning. | Backbone was pmCherry-N1 (Clontech). |
| pEGFP-G3BP1 | Generated by restriction enzyme cloning. | Backbone was pEGFP-C3 (Clontech). |
| pXJ40-myc-WDR62(842-1329) | Generated by restriction enzyme cloning. | Backbone was pXJ40-myc. |

| Reagent/resource | Reference or source | Identifier or catalogue number |
|---|---|---|
| pXJ40-myc-WDR62(1290-1523) | Generated by restriction enzyme cloning. | Backbone was pXJ40-myc. |
| pEGFP-WDR62(D511N) | Generated by site-directed mutagenesis of pEGFP-WDR62(FL). | |
| pEGFP-WDR62(A1078T) | Generated by site-directed mutagenesis of pEGFP-WDR62(FL). | |
| pEGFP-BAG2 | Generated by restriction enzyme cloning. | Backbone was pEGFP-C3 (Clontech). |
| pEGFP-DNAJC7 | Generated by restriction enzyme cloning. | Backbone was pEGFP-C3 (Clontech). |
| pEGFP-STIP1 | Generated by restriction enzyme cloning. | Backbone was pEGFP-C3 (Clontech). |
| pFGAMS-EGFP | Purchased from Addgene. | #99107. |
| pPPAT-EGFP | Purchased from Addgene. | #99105. |
| pGART-GFP | Purchased from Addgene. | #99106. |
| pPAICS-EGFP | Purchased from Addgene. | #99108. |
| pMAPKBP1-mCherry | Generated by restriction enzyme cloning. | Backbone was pmCherry-N1 (Clontech). |
| pXJ40-myc-FGAMS | Generated by restriction enzyme cloning. | Backbone was pXJ40-myc empty vector. |
| pFGAMS-mCherry | Generated by restriction enzyme cloning. | Backbone was pmCherry-N1 (Clontech). |
| pCAG-YFP | Tetsuichiro Saito, Chiba University, Japan | |
| **Antibodies** | | |
| WDR62 (rabbit polyclonal) | Bethyl Laboratories | A301-560A |
| GAPDH (rabbit polyclonal) | Abcam | ab9485 |
| α-Tubulin (mouse monoclonal) | Sigma-Aldrich | T5168 |
| HA (rabbit polyclonal) | Sigma-Aldrich | H6908 |
| HA (mouse polyclonal) | Roche | ROAHA |
| Myc (mouse monoclonal) | Santa Cruz | sc-40 |
| Cytochrome C (mouse monoclonal) | BD Biosciences | 556432 |
| XDH (rabbit monoclonal) | Abcam | ab109235 |

| Reagent/resource | Reference or source | Identifier or catalogue number |
|---|---|---|
| HPRT (rabbit polyclonal) | Abcam | ab10479 |
| PFAS (rabbit monoclonal) | Cell Signaling Technology | 76957 |
| BAG2 (goat polyclonal) | Abcam | ab752 |
| DNAJC7 (rabbit polyclonal) | Proteintech | 11090-1-AP |
| STIP1 (rabbit monoclonal) | Abcam | ab126724 |
| HSP70 (mouse monoclonal) | Abcam | ab2787 |
| HSP90 (rabbit polyclonal) | Abcam | ab13495 |
| PH3 (rabbit monoclonal) | Cell Signaling Technology | 53348 |
| TBR2 (sheep polyclonal) | Invitrogen | PA5-47818 |
| GFP (chicken polyclonal) | Abcam | ab13970 |
| Ki67 (mouse monoclonal) | BD Pharmingen | 550609 |
| BAG2 (rabbit polyclonal) | Invitrogen | PA5-96794 |
| WDR62 (rabbit polyclonal) | Novus Biologicals | NB100-77302 |
| WDR62 (mouse monoclonal) | Sigma-Aldrich | W3269 |
| **Oligonucleotides and other sequence-based reagents** | | |
| Human *WDR62F* | GGGATAGTCCGCATCTTCCAG | Bioneer. |
| Human *WDR62R* | CTGTGGAAGAGGAAGCTGGGC | Bioneer. |
| Human *GAPDH F* | AATGGGCAGCCGTTAGGAAA | Bioneer. |
| Human *GAPDH R* | GCCCAATACGACCAAATCAGAG | Bioneer. |
| Human *ACTB F* | ACAGAGCCTCGCCTTTGCC | Bioneer. |
| Human *ACTB R* | GATATCATCATCCATGGTGAGCTGG | Bioneer. |
| Human *RNA18S F* | CTTAGAGGGACAAGTGGCG | Bioneer. |
| Human *RNA18S R* | ACGCTGAGCCAGTCAGTGTA | Bioneer. |
| Human *PRPS1 F* | CCAGGAGACCTGAGTGACCT | Bioneer. |
| Human *PRPS1 R* | CAGCTGCATGGCAGATTGTG | Bioneer. |
| Human *PFAS F* | CCCCATCCCATCCAGAGTTTC | Bioneer. |
| Human *PFAS R* | GTCTAAAGCCAGACCAAGCTCC | Bioneer. |
| Human *PPAT F* | GCTTACGCAGGAAAGTGTGG | Bioneer. |
| Human *PPAT R* | TGGCTGAATGAAGGTTCTCCC | Bioneer. |
| Human *GART F* | CCACTTCCAGATCAGGCTGTAA | Bioneer. |
| Human *GART R* | TTCCAACGCCATCTGTTCCA | Bioneer. |
| Human *PAICS F* | GCTGAGTATGAAGGGGATGG | Bioneer. |
| Human *PAICS R* | ACATCACTGGTCCCAAACCA | Bioneer. |
| Human *ADSL F* | AAATGCACTTGACCTGCTTTTGC | Bioneer. |
| Human *ADSL R* | TCAGCTGTGCAGGCTGGAAAT | Bioneer. |

| Reagent/resource | Reference or source | Identifier or catalogue number |
|---|---|---|
| Human *ATIC F* | CTGGAACCATTGGCGAGGAT | Bioneer. |
| Human *ATIC R* | GTACGCCACACCACTCCTTT | Bioneer. |
| Human *ITPA F* | TGTAACGGGGAACGCCAAGA | Bioneer. |
| Human *ITPA R* | TCTCCAGAAACCACTTTATGTAGGG | Bioneer. |
| Human *ADA F* | CGAAGTAGTAAAAGAGGCTGTGG | Bioneer. |
| Human *ADA R* | AATGACTGCATGCTCCGTGT | Bioneer. |
| Human *HPRT1 F* | CCTGGCGTCGTGATTAGTGA | Bioneer. |
| Human *HPRT1 R* | CGAGCAAGACGTTCAGTCCT | Bioneer. |
| Human *IMPDH1/2 F* | ACGGCCTCACCTACAATGAC | Bioneer. |
| Human *IMPDH1/2 R* | GGATGAAGCCAATACCGCCT | Bioneer. |
| Human *AK F* | CTCGGCAGGATGGAAGAGAAG | Bioneer. |
| Human *AK R* | AGAGGTGGGTGTAGCCATACT | Bioneer. |
| Human *XDH F* | ATGGGCCAAGGCCTTCATAC | Bioneer. |
| Human *XDH R* | TAGACGGCCTGTCCATTGAG | Bioneer. |
| **Chemicals, enzymes and other reagents** | | |
| Sodium arsenite | Sigma-Aldrich | S7400 |
| Sucrose | Chem-Supply | SA030 |
| Dextrose | Sigma-Aldrich | D9439 |
| Sorbitol | Sigma-Aldrich | S3889 |
| Cycloheximide | Sigma-Aldrich | 01810 |
| NVP-AUY922 | Cayman Chemical | 10012698 |
| 17-AAG | Cayman Chemical | 11039 |
| MKT-077 | Cayman Chemical | 14395 |
| Sodium chloride | Sigma-Aldrich | J21618 |
| **Software** | | |
| Leica LAS X | Leica | |
| Zeiss ZEN Blue | Zeiss | |
| Fiji (ImageJ) | https://www.nature.com/articles/nmeth.2019 | |
| SCIEX ProteinPilot™ (ver. 4.5) | SCIEX | |
| SCIEX Analyst | SCIEX | |
| MS-DIAL (ver. 5.1.221218) | https://www.nature.com/articles/nmeth.3393 | |
| GraphPad Prism 9 (ver. 9.2) | Dotmatics | |
| LI-COR Image Studio™ | LI-COR | |
| **Other** | | |

## Antibodies, reagents, and plasmids

Primary antibodies used in this study are outlined in detail in Appendix Table S1. Secondary antibodies used in this study were: anti-rabbit IgG (H + L) Alexa Fluor 488 (Invitrogen, A11008, 1:800 (IF)); anti-mouse IgG (H + L) Alexa Fluor 555 (Invitrogen, A21422, 1:800 (IF)); goat anti-rabbit IgG (H + L) HRP (Invitrogen,

31460, 1:10,000 (WB)); goat anti-mouse IgG (H + L) HRP (Invitrogen, 62-6520, 1:10,000 (WB)); donkey anti-goat IgG (H + L) HRP (Invitrogen, A15999, 1:5000 (WB)).

Origins of all plasmids used in this study are outlined in Appendix Table S2. Traditional restriction enzyme cloning (double-restriction digestion and ligation) techniques were used for all cloning procedures. The empty pmCherry-N1 and pmCherry-C1 vectors were purchased from Clontech. Empty pXJ40-myc, -HA and -FLAG vectors are from (Xiao et al, 1991). Recombinant plasmids formed with the above vectors were completed using standard restriction-enzyme cloning techniques with PCR-amplified inserts generated with specific primers. Plasmids were ligated using the Rapid DNA Ligation Kit (Thermo Scientific, K1422) as per the manufacturer's instructions. All constructs validated by full sequence analysis. Primers outlined in Appendix Table S3.

## AlphaFold2 protein structure prediction

Amino acid sequences for human WDR62 and BAG2 were retrieved from UniProt. The ColabFold v1.5.5 implementation of AlphaFold2 was used (Mirdita et al, 2022) on a cloud 8x Tesla V100 16GB GPU. PAE Viewer (http://www.subtiwiki.uni-goettingen.de/v4/paeViewerDemo) was used to view predicted structure and predicted alignment error (PAE). Proteins, Interphases, Structures and Assemblies (PDBePISA) (https://www.ebi.ac.uk/pdbe/pisa/) was used to calculate interphase area (Å) for predicted complexes.

## Cell culture, transfections, and chemicals

WT AD293 cells (WT) (Agilent Technologies, Inc.) and AD293 cells deleted of WDR62 by CRISPR/Cas9-sgRNA (WDR62 KO) were maintained by standard protocols and assessed free of mycoplasma (Lim et al, 2016). In brief, cells were maintained in culture media containing Dulbecco's Modified Eagle Medium (DMEM) (Gibco, 21969035) supplemented with 10% (v/v) foetal bovine serum (FBS) (Gibco, 10099141) and 100 µg/ml penicillin-streptomycin (Pen-Strep) (Gibco, 15140122). Cells were cultured at 37 °C and 5% $CO_2$ within a humidified incubator. DNA transfections were performed with Lipofectamine™ 2000 (Invitrogen, 11668019) according to the manufacturer's instructions. Where indicated, cells were treated with sodium arsenite (arsenite; 0.5 mM; Sigma-Aldrich, S7400), D-sorbitol (sorbitol; 0.5 M; Sigma-Aldrich, S3889); sucrose (0.5 M; Chem-Supply, SA030), dextrose (0.5 M; Sigma-Aldrich, D9439); cycloheximide (CHX; 100 µg/ml; Sigma-Aldrich, 01810), NVP-AUY922 (NVP; 500 nM; Cayman Chemical, 10012698), sodium chloride (NaCl; 0.5 M; Sigma-Aldrich, J21618); betaine (20 mM; Sigma-Aldrich, 61962). For purine-depletion studies, cells were cultured in DMEM supplemented with 10% dialysed FBS (Gibco, A3382001) for 7 days. Dialysed FBS, commonly prepared by filtration with a 10 kDa cutoff semipermeable membrane, effectively eliminates small molecules such as nucleotides while retaining serum proteins and therefore is commonly used to generate purine-depleted media (Baresova et al, 2018; An et al, 2008; Wu et al, 2024).

## RNAi

RNA transfections were performed using Lipofectamine™ RNAi-MAX (Invitrogen, 13778150) according to the manufacturer's

instructions. ON-TARGETplus non-targeting control pool (D-001810-10) was used as the siRNA control. ON-TARGETplus human BAG2 pool (L-011961-00), and human WDR62 pool (L-031771-01) were obtained from Dharmacon™ (Horizon Discovery).

## Immunofluorescence

Cells were grown on square 22-mm$^2$ uncoated glass coverslips in six-well plates. All steps are performed at room temperature unless otherwise stated. Cells were rinsed three times quickly with ice-cold PBS, then fixed with 4% (w/v) paraformaldehyde (PFA) in PBS for 20 min. Cells rinsed three times in PBS for 5 min each, then permeabilised with 0.2% (v/v) Triton X-100 in PBS (PBS-T) for 10 min. Cells rinsed again three times in PBS for 5 min each, then blocked with 1% (w/v) bovine serum albumin (BSA) (Sigma-Aldrich, A9418) in PBS-T for 30 min. Cells subsequently stained with appropriate primary antibodies (Appendix Table S1), and then with appropriate fluorophore-conjugated secondary antibodies for 1 h in darkness. Unless otherwise stated, antibodies were diluted in 1% BSA in PBS-T. Cells rinsed three times in PBS for 5 min each, counterstained with DAPI for 5 min, and mounted in ProLong™ Diamond Antifade Mountant (Invitrogen, P36970) on glass slides. Cells overexpressing fluorescently tagged protein constructs were simply fixed, washed and counterstained with DAPI (Invitrogen, D1306).

## Protein extractions, immunoblots and co-immunoprecipitation

After treatments, cells were lysed with RIPA buffer [150 mM NaCl, 100 mM $Na_3VO_4$, 50 mM Tris-HCl pH 7.3, 0.1 mM EDTA, 1% v/v Triton X-100, 1% w/v sodium deoxycholate and 0.2% w/v NaF] supplemented with protease inhibitors (cOmplete™, Mini, EDTA-free Protease Inhibitor Cocktail, Roche, 11836170001). Protein lysates were cleared by centrifugation (16,000×g, 15 min) and protein concentrations determined by protein assay with Bradford dye reagent (Bio-Rad, 5000006). Proteins were separated by SDS-PAGE without boiling due to the heat sensitivity of WDR62 (Wasserman et al, 2010), then transferred onto polyvinylidene fluoride (PVDF) membranes and immunoblotted. Bands were detected by enhanced chemiluminescence and quantified by densitometry on a Licor Odyssey Fc imager using Image Studio 5.2. Unless stated otherwise, band densities are normalised to untreated WT cells as a reference condition to allow comparison of relative fold change in protein levels between replicate experiments.

For co-immunoprecipitation experiments, cleared cell lysates were incubated with rabbit anti-HA primary antibodies, followed by affinity isolation with Protein A agarose beads (Roche, PROTAA-RO) overnight at 4 °C on an end-over-end rotator. Beads were pelleted and washed thoroughly in RIPA buffer with fresh protease inhibitors. Immunoprecipitated proteins were eluted from beads with Laemmli buffer and separated by SDS-PAGE for immunoblotting.

For GFP-Trap immunoprecipitation experiments, cell lysates were incubated with GFP-Trap® agarose (ChromoTek, gta) for 1 h at 4 °C on an end-over-end rotator. Beads were subsequently pelleted and washed thoroughly in RIPA buffer. Immunoprecipitated proteins were eluted with Laemmli buffer and separated by SDS-PAGE for downstream immunoblot.

## Micrograph acquisition and processing

Immunofluorescence and fluorescence images were acquired with a Leica SP8 DMi8 inverted point scanning confocal microscope (Leica Microsystems, UK) equipped with a Leica HC PL APO 63x/ 1.30 GLYC CORR CS2 objective, or a Zeiss LSM900 inverted laser-scanning confocal microscope with Airyscan 2 equipped with a 63× C-Plan Apo NA 1.4 oil-immersion objective. Image exposure times, gain and other settings were standardised between samples. Resolution was set at $1024 \times 1024$ or $2048 \times 2048$ with bi-directional scanning. Pinhole size set at 1. Image panels were assembled with ImageJ and Microsoft PowerPoint. Images collected using either Leica LAS X or Zeiss ZEN software and processed with Fiji (ImageJ) (Schindelin et al, 2012), where brightness and contrast were minimally adjusted across all panels indiscriminately. These adjustments merely enhanced viewing quality and did not obscure original information present in raw files.

For quantification of granule size and number (per cell), images were analysed with Fiji (ImageJ). Micrographs were converted to 8-bit grayscale images, and granules were masked by intensity thresholding (Image → Adjust → Threshold). Thresholding was performed manually and was carefully adjusted to capture granules accurately. Granules were analysed with the *Analyse Particles* tool (Analyse → Analyse Particles), which selected granules with a size between 0.1 and $10 \, \mu m^2$, hence excluding background noise and large aggresomes/ aggregates. Granules were approximated as a sphere, and area ($\mu m^2$) was converted to average diameter ($\mu m$) accordingly. In total, 25–30 cells were analysed per biological replicate. Biological replicates are those in which experiments were completed on different days and different passage numbers. SuperPlots were used to convey both cell-level and experimental variability where appropriate. In this case, cell-level values were pooled for each biological replicate, and the mean was calculated for each pool. These means were used to calculate the average and SEM bars. Only cells with at least 10 granules were subjected to quantification.

Co-localisation analyses were performed using the JaCoP plugin in ImageJ (Fiji). To quantify spatial correlation, $10 \, \mu m^2$ regions of interest (ROIs) were randomly selected within the cytoplasm and subjected to Pearson's correlation coefficient (PCC) analysis to generate $R^2$ values. To assess the association of WDR62 granules/ purinosomes with mitochondria or microtubules, $10 \, \mu m^2$ ROIs were randomly cropped from larger images. WDR62 granules were segmented by intensity thresholding, while mitochondria (CytC) and microtubules ($\alpha$-tubulin) were manually delineated using the freehand selection tool, then thresholded. The DiAna (distance analysis) plugin was used to quantify object counts and determine the proportion of granules overlapping with mitochondria or microtubules. Granules exhibiting any signal overlap with CytC or $\alpha$-tubulin channels were classified as mitochondria-associated or microtubule-associated, respectively. To validate co-localisation results, randomised images were generated whereby one channel was rotated 90° relative to the other before analysis. To prevent objects of interest from being rotated outside the cell boundary, the largest continuous square region within the cytosol was selected for each analysis. Within these defined regions, Pearson's correlation coefficient was compared to the coefficient calculated after the rotation of one channel (randomised).

For the analysis of fluorescence signal variation in Fig. 6, random linear ROIs were placed within the cytoplasm of randomly selected cells. A Gaussian blur (radius = 2) was applied prior to analysis to suppress smaller background fluctuations in fluorescence signal while preserving larger intensity variations. The standard deviation (SD) of raw fluorescence intensity values was subsequently normalised to line length.

## Live-cell imaging

Cells were seeded onto eight-well glass-bottom chamber slides (Ibidi). Media was changed to DMEM, no phenol red, supplemented with 10% FBS and 100 $\mu$g/ml Pen-Strep 24 h prior to imaging. Imaging was routinely conducted on a Zeiss LSM900 inverted laser-scanning confocal microscope, equipped with a 63x C-Plan Apo NA 1.4 oil-immersion objective and retrofitted with an incubation chamber to maintain physiological conditions (37 °C and 5% $CO_2$). Tracking of granules in time-lapse images conducted with the TrackMate plugin of ImageJ (Fiji).

## Cytotoxicity detection (LDH) assay

WT AD293 and WDR62 KO AD293 cells were seeded at $4 \times 10^4$ cells/well in a 96-well tissue culture plate and cultured in purine-rich or purine-depleted conditions for 7 days. The next day, cells were assayed for lactate dehydrogenase (LDH) activity with a Cytotoxicity Detection Kit[PLUS] (LDH) (Roche, 11644793001) according to the manufacturer's instructions. Absorbance (OD) was measured on a FLUOstar Omega Microplate Reader (BMG Labtech) at 490 nm. Two controls were included: background control (cell-free, assay medium only) and high control (maximum LDH release, achieved by cell lysis in Triton X-100). To determine experimental absorbance values, average absorbance values of the sextuplicate samples and controls were calculated and absorbance values of the background control were subtracted. Percent cytotoxicity was determined by the following equation: Cytotoxicity (%) = (exp. value/high control) × 100. Any negative values after subtracting the average background absorbances were considered as zero.

## BioID pulldown and protein mass spectrometry

WT AD293 cells were seeded in 10 cm dishes, and transfected with WDR62-BirA*-HA, WDR62-HA or BirA*-HA the following day with Lipofectamine 2000, as per the manufacturer's instructions. The following day, cells were treated with 50 $\mu$M biotin for 24 h. Cells arrested in mitosis by low-dose nocodazole treatment (200 ng/ ml) represent *mitotic* populations. *Asynchronous* populations are untreated. Cells were washed in PBS and lysed in RIPA lysis buffer. Biotinylated proteins were captured with Pierce™ Streptavidin Agarose (Thermo Fisher Scientific, 20353) on a rotator for 4 h at 4 °C. Agarose beads were washed three times in RIPA buffer and were collected by centrifugation for 1 min between wash steps. For immunofluorescence, cells were probed with Streptavidin-Cy3™ (Sigma-Aldrich, S6402). For western blot analysis, streptavidin beads were resuspended in Laemmli buffer with 1 mM biotin following pulldown. Pulldown fractions and total cell lysates were resolved by SDS-PAGE, transferred to PVDF membranes, and probed with primary antibodies for known WDR62 interactors. To identify novel proteins labelled by the WDR62-BirA pulldown, biotinylated proteins were cleaved from streptavidin beads by

tryptic digestion and were prepared for LC-MS/MS analysis. Briefly, streptavidin agarose beads were subjected to an overnight on-bead tryptic digestion at 37 °C using 0.5 µg of trypsin in the presence of 0.25 M urea and 5 mM TCEP, in 50 mM TEAB buffer (pH 8.0). Following digestion, samples were centrifuged, and the supernatants were desalted and concentrated using ZipTip C18 columns. Peptides were analysed by LC-MS/MS on a TripleTOF 5600 platform. Protein identification was performed using ProteinPilot™ software (version 4.5), and quantitation was carried out using a label-free data-independent acquisition (DIA) approach. Protein fractions from three independent WDR62-BirA pulldown replicates were analysed, and candidate interactors were considered as those identified in at least two separate experiments.

We used the BioGRID database (ver. 4.4.232) to retrieve a list of experimentally validated WDR62-interacting proteins curated from a range of affinity capture, yeast two-hybrid, and other biochemical analyses for detecting protein-protein interactions (Stark et al, 2006). The database was queried using the WDR62 protein name and manually filtered for interactions supported by physical association evidence. The list was subsequently compared to our interactors identified through our BioID proximity labelling experiments.

Gene ontology (GO) term enrichment was conducted using the PANTHER over-representation test (https://geneontology.org/). FDR-adjusted $P$ values from the output were plotted directly.

## XTT assay

WT AD293 and WDR62 KO AD293 cells were seeded at $2 \times 10^4$ cells/well in a 96-well tissue culture plate and were incubated at 37 °C and 5% $CO_2$ in purine-rich or purine-depleted conditions. Cell numbers were determined by XTT at day 1, day 4, and day 7 after seeding. The XTT assay enables the quantification of cellular redox potential, providing a colorimetric readout of proliferation as a function of cell number in culture by measuring cellular redox potential. Absorbance (OD) was obtained using a FLUOstar Omega Microplate Reader (BMG Labtech) at two wavelengths: 490 and 690 nm. The primary measurement wavelength was 490 nm, corresponding to the peak absorbance of the formazan product. To account for non-specific background absorbance caused by culture media or other factors, absorbance at 690 nm was measured and subtracted from absorbance at 490 nm. Cell numbers were then expressed as arbitrary units (a.u.) based on the subtracted value ($A_{490nm} - A_{690nm}$).

## Untargeted metabolomics

For untargeted analyses of intracellular metabolites, cells were seeded into T175 flasks and grown in purine-depleted media for 1 week. Cells were washed once in ice-cold PBS, harvested by scraping, and thoroughly resuspended in 10 ml ice-cold PBS. A small sample (~30 µl) was removed for cell counting on a hemocytometer. Cells were pelleted ($200 \times g$, 5 min, 4 °C) and the cell pellet was immediately quenched and resuspended in 800 µl of 100% methanol (pre-cooled at $-80$ °C for ≥1 h). Quenched cell pellets were vortexed vigorously, thawed on ice for 10 min, then snap frozen in $LN_2$ for 10 min. This freeze-thaw cycle was repeated two more times. Lysates centrifuged ($7000 \times g$, 15 min, 4 °C) and supernatant collected. Pellet resuspended in 200 µl of 80% (v/v) methanol in water (also pre-cooled at -80 °C) and extracted again

by freeze-thaw as above. Combined supernatants from the two extractions were centrifuged ($16,000 \times g$, 30 min, 4 °C) to remove all insoluble material, dried with a Genevac™ miVac Quattro vacuum concentrator (7 mbar, ~3 h, 35 °C), and reconstituted in 30 µl of 80% (v/v) methanol in water with 0.1% (v/v) formic acid. Samples were then analysed by LC-MS/MS at the SCMB Mass Spectrometry Facility, The University of Queensland.

The LC platform consisted of an Exion UPLC system. Samples were separated on a Phenomenex Kinetex C18 column (100 mm × 2.1 mm, 1.7 µm) using a water-methanol gradient. The UPLC column was maintained at 35 °C and samples were stored in the autosampler at 4 °C. Flow rate was 250 µl/min and injection volume was 10 µl. Solvent A was 0.1% formic acid in MilliQ water; solvent B was 100% methanol. The MS platform consisted of a SCIEX X500B QTOF mass spectrometer controlled by SCIEX Analyst software. The MS was operated in positive and negative ion mode and scanned from 6 to 1200 $m/z$ in SWATH DIA (data-independent acquisition) mode. Data were analysed using MS-DIAL software (ver 5.1.221218). MS-DIAL was used to match experimental mass spectra, corresponding to [M-H]- ions, against the MassBank mass spectral library. Compound identification was achieved by weighted similarity score after considering retention time, mass and MS/MS spectra. Ion identification was completed in MSFINDER. Relative abundance (a.u.) of metabolites was calculated by dividing the peak area (AUC) of each metabolite by the peak areas of all detected metabolites in the same sample (mTIC). mTIC represents the sum of all peak areas for 'genuine' or positively identified metabolites belonging to the same MS/MS scan, to avoid normalising against non-biological artefacts.

## Targeted metabolite flux analysis

To examine de novo purine synthesis and salvage flux in cells, $^{13}C_2$-glycine and $^{13}C_5$-hypoxanthine isotope tracing strategies were used. WT and WDR62 KO cells were seeded in 10 cm dishes (~60% confluent) and cultured in DMEM (Gibco, 11965092) containing 10% FBS (Gibco, 10099141) for 24 h. The next day, cells were washed once with PBS and cultured either in DMEM containing 10% FBS or 10% dialysed FBS (Gibco, A3382001) (purine-depleted) for 24 h. The next day, cells were again washed once with PBS and cultured either in MEM (Gibco, 11095080) containing 10% FBS or 10% dialysed FBS, supplemented with 1 mM $^{13}C_2$-glycine (Sigma-Aldrich, 283827) or 60 µM $^{13}C_5$-hypoxanthine (Cambridge Isotope Laboratories Inc., CLM-8042-PK) for 4 h.

For the extraction of intracellular metabolites, cells were washed once with ice-cold PBS and scraped into 1 ml of extraction buffer containing 40% methanol, 40% acetonitrile, 19.9% water and 0.1% formic acid. Cell solution vortexed and sonicated with a probe sonicator with 3 ×5 s pulses, followed by 2 x freeze-thaw cycles in $LN_2$ to facilitate cell lysis. Insoluble material pelleted by centrifugation at $16,000 \times g$ for 5 min at 4 °C. Supernatant containing intracellular metabolites was pipetted into a fresh tube. The pellet was dissolved in 8 M urea for protein quantitation. Supernatant was dried to a pellet with a SpeedVac (Eppendorf), before being freeze-dried overnight, and reconstituted in 2% ACN. LC-MS/MS analysis was performed using a Shimadzu UHPLC system (Nexera X2) coupled to a Shimadzu 8060 triple quadrupole mass spectrometer (Shimadzu, Kyoto, Japan). Chromatographic separation was performed on an Acquity HSS T3 1.8 µm, 2.1 × 100 mm column (Waters Corporation, 186003539) with

Acquity HSS T3 1.8 μm 2.1 × 5 mm guard columns (Waters Corporation, 186003976). The mobile phase consisted of A: 7.5 mM tributylamine with acetic acid (pH 4.95), and B: 100% acetonitrile. Acquisition was performed in negative ionisation mode with a 250 μl/min flow rate, 5 μl injection volume, and column temperature of 45 °C.

## RNA extraction, cDNA synthesis and qPCR analysis

Total RNA was extracted from WT and WDR62 KO AD293 cells using the PureLink™ RNA Mini Kit (Invitrogen, #12183020). 1 μg total RNA was used to perform reverse transcription using SuperScript™ III Reverse Transcriptase (Invitrogen, #18080044), as per the manufacturer's instructions. On-column DNase treatment was performed using the PureLink™ DNase Set (Invitrogen, #12185010). Real-time PCR was performed on a QuantStudio™ 7 Flex Real-Time PCR System (Applied Biosystems) after mixing cDNA, QuantiNova SYBR Green Master Mix (Qiagen, #208052) and gene-specific primers as listed in Appendix Table S4. Gene-specific primers were designed using the NCBI Primer-BLAST tool (https://www.ncbi.nlm.nih.gov/tools/primer-blast/), where all primer sets produced an amplicon 80-200 bp in size, had at least two mismatches within the last 5 bp at the 3' end, and at least one primer spanned an exon-exon junction. Relative gene expression was calculated with the $2^{-\Delta\Delta Ct}$ method using GAPDH, β-actin and 18S rRNA as internal controls. Data were presented as fold-change relative to control (WT) cells.

## Proximity ligation assay (PLA)

PLA was performed using Duolink® Proximity Ligation Assay Kit (Sigma-Aldrich) according to the manufacturer's instructions. Cells were seeded on round 1 cm² coverslips in a 24-well plate and fixed the next day with 4% PFA for 20 min. Cells permeabilised with 0.2% Triton X-100 in PBS for 10 min, then blocked with Duolink® Blocking Solution at 37 °C for 1 h, then incubated with primary antibodies at 4 °C overnight. The next day, cells were washed twice in Wash Buffer A for 5 min, then incubated with PLA probes at 37 °C for 1 h. Cells were again washed twice in Wash Buffer A for 5 min, then incubated with ligation solution at 37 °C for 30 min. Cells were again washed twice with Wash Buffer A for 5 min, then incubated with amplification solution at 37 °C for 100 min. Cells were then washed twice in Wash Buffer B for 10 min, then in 0.01x Wash Buffer B for 1 min. Coverslips mounted onto slides with Duolink® In Situ Mounting Medium with DAPI and examined by confocal microscopy within 4 days.

## In utero electroporation and brain slice analysis

All animal procedures were approved by the University of Queensland's Health Sciences Animal Ethics Committee (2024/AE000026) and conducted in compliance with the Australian code for the care and use of animals for scientific purposes. Animals were housed under standard conditions with controlled environments, including light–dark cycles, and ad libitum access to food and water. In utero electroporation was performed on time-mated CD1 mice at embryonic day 14.5 (E14.5) as previously described (Paolino et al, 2023). Briefly, embryos were electroporated with siRNA targeting *Wdr62*, *Hprt* or a non-targeting control (10 μM

final concentration), co-delivered with a CAG-eYFP expression plasmid (1 μg/μl). After 48 h, embryonic brains were collected and drop-fixed in 4% paraformaldehyde for at least twenty-four hours before dissection and sectioning.

Brains were embedded in 3.4% agarose and sectioned into 50 μM coronal slices using a vibratome (VT1000S, Leica Biosystems). Immunostaining was performed as previously described (Paolino et al, 2023). Antibodies described in Appendix Table S1. Investigators were blinded to sample identity prior to image acquisition. Imaging was performed on a Zeiss LSM900 laser-scanning confocal microscope with Airyscan 2, employing tile scanning and Z-stack acquisition to capture a region of interest encompassing the eYFP-positive cortical patch. Cortical layers including the ventricular zone/subventricular zone (VZ/SVZ), intermediate zone (IZ), cortical plate (CP) and marginal zone (MZ) were delineated based upon nuclei density in the DAPI channel. Cell counting was performed using Fiji (ImageJ) following maximum intensity Z-projection.

## Analyses and statistical comparisons

All statistical analyses and the generation of graphs as conducted in GraphPad Prism 9 software (version 9.2.0). Densitometric analysis of Western Blot bands was performed in LI-COR Image Studio™ software. All bands were normalised against a loading control (GAPDH or α-tubulin). The D'Agostino-Pearson test for normality was used to determine whether or not the data assumed a Gaussian (normal) distribution. In groups that assumed Gaussian distributions, parametric tests were employed (e.g., two-tailed Student's $T$ test, one-way ANOVA with Tukey's multiple comparisons, and two-way ANOVA with Sidak's multiple comparisons). In groups that did not assume a Gaussian distribution, non-parametric tests were employed (Mann-Whitney rank-sum test, Kruskal-Wallis test with Dunn's multiple comparisons, and two-way ANOVA with Sidak's multiple comparisons). The threshold of statistical significance was set at $P \leq 0.05$. All error bars represent mean ± standard error of mean (SEM). All figures were generated using GraphPad Prism 9, Microsoft PowerPoint, or Adobe Illustrator. With the exception of in utero electroporation studies, confocal images were analysed without prior blinding, and investigators were aware of sample identities during imaging. Each experiment was conducted at least three times for at least three independent biological replicates ($n = 3$).

# Data availability

The mass spectrometry data from this publication have been deposited in the Figshare database (https://figshare.com/) and assigned the identifier https://doi.org/10.6084/m9.figshare.30959294. The BioID dataset for Fig. 1 has also been deposited to the ProteomeXchange Consortium via the PRIDE partner repository with the dataset identifier PXD074147.

The source data of this paper are collected in the following database record: biostudies:S-SCDT-10_1038-S44318-026-00724-0.

## Peer review information

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

## Acknowledgements

We acknowledge the use of the SBMS Imaging Facility at the University of Queensland (UQ) in providing the necessary imaging capabilities to complete this project, and the SCMB Mass Spectrometry Facility for untargeted metabolomics. We wish to specially thank Dara Daygon and Robin Palfreyman from Queensland Metabolomics and Proteomics (QMAP) for their help and assistance with targeted metabolomics and data analysis, including corrections for natural isotope abundance. We also acknowledge and thank Milot Mirdita, Sergey Ovhinnokov, Martin Steinegger and the ColabFold team for making their AlphaFold2 modelling pipeline available for public use. MJM is supported by an Australian Government Research Training Program (RTP) scholarship and a Tour de Cure PhD Support Scholarship (RSP-079-2024).

## Author contributions

**Matthew J Morris**: Conceptualisation; Investigation; Writing—original draft; Writing—review and editing. **Yvonne Y Yeap**: Investigation. **Jonathon R Edwards**: Investigation. **Chi Chen**: Investigation. **Annalisa Paolino**: Investigation. **Sebastian G B Furness**: Writing—review and editing. **S Sean Millard**: Resources; Writing—review and editing. **Julia K Pagan**: Resources; Writing—review and editing. **Laura R Fenlon**: Investigation; Writing—review and editing. **Dominic C H Ng**: Conceptualisation; Resources; Supervision; Funding acquisition; Writing—review and editing.

Source data underlying figure panels in this paper may have individual authorship assigned. Where available, figure panel/source data authorship is listed in the following database record: biostudies:S-SCDT-10_1038-S44318-026-00724-0.

## Disclosure and competing interests statement

The authors declare no competing interests.

