## [Peer Review File · The EMBO Journal]

Microcephaly-associated protein WDR62 supports purine metabolism by interacting with co-chaperone BAG2

Matthew Morris, Yvonne Yeap, Jonathon Edwards, Chi Chen, Annalisa Paolino, Sebastian Furness, S Sean Millard, Julia Pagan, Laura Fenlon, and Dominic Ng

Corresponding author(s): Dominic Ng (d.ng1@uq.edu.au)

Review Timeline:

Transfer Date:	30th Jun 25
Editorial Decision:	4th Aug 25
Revision Received:	30th Nov 25
Editorial Decision:	17th Dec 25
Revision Received:	7th Jan 26
Accepted:	5th Feb 26

Editor: Ioannis Papaioannou

**Transaction Report: This manuscript was transferred to
The EMBO Journal following peer review at Review Commons.**

**Review
COMMONS**

Review #1

1. Evidence, reproducibility and clarity:

Evidence, reproducibility and clarity (Required)

In Morris, M.J. et al., the authors strive to better understand the roles for the microcephaly protein WDR62 in brain growth and function. To accomplish this, an in situ biotinylation assay was performed and identified 42 proteins proximal to WDR62 including the Hsp70 co-chaperone BAG2. Through a series of co-immunoprecipitation assays, the BAG2-WDR62 interaction was shown to be mediated through the structured N-terminal region of WDR62, and it was proposed that common WDR62 mutations disrupt this interaction. In AD293 cells, loss of WRD62 expression resulted in an increase in the expression of BAG2 expression while reducing HPRT expression. Subsequent loss of BAG2 expression by siRNA treatment restored the expression of HPRT suggesting that there is an association between the stability of HPRT and BAG2, likely mediated through its proposed association with Hsp70/90 molecular chaperones. Finally, the authors investigate the subcellular localization and ability of WRD62 to phase separate. WRD62 was shown to form discrete condensates induced by sorbitol-mediated hyperosmotic stress. The formation of WRD62 granules are hypothesized to be driven by cell volume exclusion and macromolecular crowding. These granules appear similar, both in physical appearance and characteristics, to other reported biomolecular condensates such as those reported in metabolism (e.g. purinosomes). WRD62-containing condensates were shown to colocalize with enzymes in de novo purine biosynthesis; however, this association is not required for purinosome formation and/or its stability under both purine-depleted and sorbitol-driven growth conditions. Overall, the manuscript provided a previously unrealized and exciting association between WDR62 and purine metabolism.

EVIDENCE, REPRODUCIBILITY AND CLARITY

****Summary:**** The current manuscript reads as multiple manuscripts with findings that are at times weakly connected (in my opinion). For example, I had a hard time understanding how the BioID results relate to the discovery of WRD62 phase-separation and its colocalization with purinosomes. I would strongly encourage the authors to consider dividing the results into separate manuscripts to strengthen their claims and create a more focused and cohesive manuscript (or series of manuscripts). I believe then several of my reservations associated with the current manuscript will be addressed, and in my opinion, the hard work from the authors will be better received across the scientific community.

I would like to commend the authors for all the work that went into the current version of the manuscript. Being part of a biochemistry and cell biology research group, I completely understand how much time and effort must have went into generating these data. That being said, I felt that there were several instances where clarification and additional information is warranted to arrive at the conclusions made by the authors. These points are outlined below.

****Major Comments:****

1. There appears to be a discrepancy between the data presented in Figure 1 and what is stated in the main text. Clarification is necessary to better understand the results:

- The following statement (and derivatives of it) are repeated throughout the manuscript:

"...we found that the WDR62 interactome comprised molecular chaperones such as HSP70, HSP90, and their co-regulators, BAG2, STIP1, and DNAJC7" (lines 91-93, 316-318, 422-425). STIP1 and DNAJC7 were not identified in the list of 42 proximal proteins to WDR62 (Figure 1D). DNAJC7 was included because of a previous report curated in the BioGRID database, and there is no mention of HSP90 in the data produced in Figure 1. Please revise the main text to reflect the data that was generated.

- Based on the data presented in the Venn Diagrams in Figure 1D, the author's numbers do not seem to be consistent with the sentence on lines 126-128. I count 37 proteins unique to their BioID study, 90 unique to the BioGRID database, and 5 proteins that overlap between the two data sets. This sentence needs to be revised.

- What data were used to generate the interaction map in Figure 1I? Enzymes tied to purine metabolism were not identified from the data presented in Figure 1D but have now appeared. A discussion of this in the main text is warranted.

2. This reviewer has several reservations on how the various key players in the manuscript are related to substantiate the conclusions made in the manuscript. For instance, how is HPRT, purinosomes, and WDR62 related? How about HSP90, WRD62, and HPRT? Pairwise connections were made throughout the manuscript; however, trying to tie all three together is difficult with the data presented.

- The authors tried to connect HPRT, purinosomes, and WDR62 with BAG2; however, this study could greatly improve if we understood how a knockdown of BAG2 impacts purinosome formation and/or WDR62 colocalization with purinosome enzymes.

- Is HPRT a client of HSP90? And how are WRD62 and HSP90 related since they do not associated (based on your BioID data)? These connections would again strengthen the arguments made in the manuscript and help to explain the HSP90 inhibition data presented in Figures 7F and 7G.

3. Caution is warranted when making conclusions about WDR62 (and its granules) and

purinosomes.

- The authors describe the association between WDR62 and purinosomes differently throughout the text. I would recommend that the authors come to some conclusion about this and be consistent.

A. (Lines 339-340) "WDR62 granules represent or overlap substantially with the phase-separated metabolons known as purinosomes". Based on the data presented, it appears that these might still be different entities but either overlap or have similar components. Purinosome localization with mitochondria (approx 60-80%) and microtubules (approx 90-95%) were significantly higher than those reported for WDR62 granules (approx 40%). This comparison would suggest that not all WDR62 granules behave similarly to purinosomes. And from the dot plot in Figure 3G, about half of the characterized WDR62 granules do not align with the previously reported characteristics of purinosomes.

B. In the abstract and introduction, the authors state that WDR62 is being recruited to the purinosome and leave out the other possibility. I would recommend that the authors soften this claim in these sections because of the above possibility but also the lack of characterization of the sorbitol-induced "purinosomes". There is little discussion or evidence for how sorbitol induces purinosome formation. Is de novo purine biosynthesis activated upon sorbitol treatment? Are multiple de novo purine biosynthetic enzymes present in the sorbitol-induced "purinosomes"? Further, I agree that there is a tendency for WDR62 to associate with condensates that bear an enzyme within de novo purine biosynthesis; however many of these proteins are known to self-aggregate upon cell stress. Therefore, the entities that are being observed and called purinosomes might not be bone fide purinosomes. Additional care is necessary to make these statements. In my opinion, the current data only suggests that this might be a possibility.

- (Lines 325-329) The authors reference a previous manuscript demonstrating that co-chaperones co-cluster with purinosomes. Based on this fact, they infer that WDR62 granules might represent purinosomes since WDR62 interacts with these same set of co-chaperones. These co-chaperones interact with a large number of different proteins (in fact, most kinases), so it is uncertain how the authors decided to go down this path to link purine metabolism with WDR62. Discussion of how this connection was made would help elevate the story. What additional insights did they have that lead them down these investigations?

- If WDR62 is not required for purinosome formation, why would it localize with the purinosome? Is there any hypothesis that could be readily tested to better help understand this observation? Providing a better understanding of this would greatly elevate the work.

A. (OPTIONAL) Please validate that the associations between WDR62 and the purine biosynthetic enzymes occur on the endogenous level (void of transient transfection). Many methods such as immunofluorescence and proximity ligation assays have been used by others to demonstrate protein-purinosome interactions. This result would reduce any concern that the association is a result of overexpression (artifact).

B. Figures 6F and 6G conclude that nucleosides from purine-depleted growth conditions accumulate while the corresponding monophosphates do not change between WDR62 knock-out and wildtype cells. Given that purine-depleted growth conditions activate de novo purine biosynthesis (uncertain if this has been demonstrated in AD293 cells), could this result simply demonstrate that purine salvage is no longer used and the nucleosides have accumulated and are awaiting degradation (or exportation) rather than a loss of HPRT expression as inferred from the stated conclusions? The conclusions could be better substantiated with the use of a stable isotope incorporation assay.

Is there a difference in the contribution of de novo purine biosynthesis and purine salvage to the generation of the monophosphates (AMP, GMP) between WDR62 knockout and wildtype AD293 cells? Use of a stable isotope (such as ^{15}N -glutamine) could help to come to the appropriate conclusion.

(Lines 483-485) If there is a change in de novo purine biosynthesis, are there any detectable changes in AICAR levels that might influence purine metabolism at the transcriptional level?

Are the data and the methods presented in such a way that they can be reproduced? Are the experiments adequately replicated and statistical analysis adequate?

1. For purine-depleted studies (metabolite analyses, microscopy), how long were the cells grown in purine-depleted medium before the analysis? And how was the purine-depleted medium generated? Please reference any source that might have been used.

2. Details describing the BioID experiment are minimal. How many replicates were performed, was label-free or TMT quantitation used for the protein identification. Further the data analysis and mining of the proteins from the BioID study are missing - What database was used to identify the proteins from the peptides? Please include this information in the Materials and Methods section as well as a link to a repository where the LC-MS/MS data generated can be found. Additionally, it would be very helpful to have a

- spreadsheet or table that lists the biotinylated proteins and expectant or p values for each.
3. Please include information about the streptavidin pulldown presented in Figure 1C.
 4. Many of the figure legends could benefit from a statistical description.
 5. There seems to be only two data points for Figure S3A. While there is no significant difference observed, it would be ideal to have additional replicates.
 6. (Figure 5I) Please provide statistical analysis to demonstrate the colocalization between FGAMS and WDR62 is robust in purine-depleted AD293 cells.

****Minor Comments:****

Do you have suggestions that would help the authors improve the presentation of their ideas and conclusions?

1. The use of HSP90 inhibitors is a little confusing given the connections being made with BAG2 and other HSP70 co-chaperones in Figure 1.
 - Does the same conclusions hold true with an HSP70 inhibitor or siRNA?
 - (OPTIONAL) There are a lot of discrepancies between Hsp90 inhibitors and effective treatment concentrations. For example, NVP-AUY922 caused purinosomes to disassemble whereas STA9090 cause purinosomes to change morphology and adopt a more aggregated state. Do other Hsp90 inhibitors share the same phenotypic response as NVP-AUY922 in this study?
 - The treatment time (24 h) with NVP-AUY922 is very long. Given that Hsp90 interacts with hundreds of proteins, it is hard to understand whether the effect of Hsp90 inhibition is direct or indirect. Shorter times (1 h or less) would be more insightful.
2. (OPTIONAL) Does the 2.6-fold increase in BAG2 increase its association with WDR62?
3. Is the degradation of HPRT occurring through BAG2-mediated proteasomal degradation? Showing HPRT recovery by treating the cells with MG132 along with CHX would provide meaningful clues as to how BAG2 and HPRT might be related - Is BAG2 concentration increasing to facilitate the enhanced degradation of HPRT?
4. Does HPRT colocalize with WDR62 in cells?
5. (OPTIONAL) It would be nice to see validation experiments of some of the hits in Figure 1D or 1E in a co-immunoprecipitation experiment conducted similar to Figure 1C.
6. The authors presented the findings that suggest that BAG2 interacts differently with commonly observed WDR62 mutations in MCPH2? How do these mutations affect WDR62 condensation, colocalization with purinosomes, or alter HPRT activity? Tying back the observations to something clinical would help elevate the overall significance of the

findings.

Are the text and figures clear and accurate?

1. There are many times throughout the manuscript that the wrong figure is being referenced. These mistakes caused significant confusion at many times while reviewing the manuscript. Please double check all in-text references to figures. For example, I believe that you meant to use Figure S1C instead of Figure 2E with the statement on lines 183-185. Again, I believe that correct figure reference on line 501 is Figure 7G not Figure 7E.
2. The figure legend on Figure S4 does not match the figure and the main text references. Please verify that the text in the figure legends correspond correctly to the figure.
3. Please provide this data for the sentence on lines 399-400 in the supplemental file.
4. I believe that the authors use the phrase "cell proliferation" to describe cell viability in the main text. In the Materials and Methods section, the authors state "The XTT cell proliferation assay enables quantification of cellular redox potential, providing a colorimetric readout of cell viability." Cell proliferation, viability, and cytotoxicity are different measurements, so please revise to reflect the correct experiment that was performed.

Other Minor Comments:

7. Move the sentence "In contrast, despite reduced mRNA..." (lines 387-388) to the last section when a reduction in PFAS expression was first mentioned.
8. Please reference the following in the manuscript:
 - BioGRID database in the main text and Materials and Methods section
 - The reported study showing the DNAJC7-WDR62 interaction (as curated from BioGRID)
 - Fiji in the Materials and Methods section
9. (Line 461-463) The authors state the following: "the loss of WDR62 leads to an increase in BAG2 and vice-versa (Fig. 7A) (Fig. S9B). I am not sure that the vice-versa (i.e. loss of BAG2 increases WDR62) is true. From the data presented in Figure 7H, I do not see a significant change in WDR62 expression upon BAG2 siRNA treatment.
10. For your BioID study, do you know how many or the proportion of cells that were mitotically arrested with the low dose of nocodazole (200 ng/mL)? Given the small number of unique proteins that were in the mitotic only population, it is curious to know how enriched the cells were and whether WDR62 localization is important in the context of this study.

11. Just to clarify, the WDR62-HA lane (third in each set) in Figure 1C is not WDR62-BirA*-HA and that it is only being used as a control.

12. In the Discussion (lines 439-441) "We also show that WDR62 forms dynamic, phase-separated granules that co-localise with chaperones and purine metabolic enzymes, resembling purinosomes." I believe that the authors meant to say co-chaperones instead of chaperones given no microscopy data was presented showing the colocalization of HSP70/90 with WDR62 granules. Please revise.

****Referees cross-commenting****

I agree with the comments and recommendations by the other reviewers. Many of our shared comments are those that need to be addressed to substantiate the claims made by the authors throughout the manuscript. The proposed experiments across the reviewer comments appear feasible given that similar experiments have already been presented in this version of the manuscript. I strongly encourage the authors to consider these comments when revising their manuscript to help strengthen their claims and boost its overall significance and impact.

2. Significance:

Significance (Required)

Describe the nature and significance of the advance (e.g. conceptual, technical, clinical) for the field. Place the work in the context of the existing literature (provide references, where appropriate).

The work presented explains a previously unknown role for WDR62 in the regulation of purine metabolism. Despite all the hard work that was performed to reach their conclusions, the use of the AD293 cell line and the lack of correlating the common WDR62 disease-promoting mutations to the observed findings throughout the entire manuscript slightly reduced my enthusiasm for this work. The presented study leverages a lot of existing literature to establish connections between WR62, co-chaperones, and purine metabolic enzymes, with an emphasis on purinosome metabolon, a condensate comprised of the enzymes in de novo purine biosynthesis.

State what audience might be interested in and influenced by the reported findings.

The audience that might be interested in the reported findings would likely be those tied to biomolecular condensates in cellular metabolism and their connection to disease. I also

feel that researchers that study microcephaly might be interested in this work. In my opinion, I believe that a broader readership could happen if additional studies were performed to make stronger connections between studies presented.

Define your field of expertise with a few keywords to help the authors contextualize your point of view. Indicate if there are any parts of the paper that you do not have sufficient expertise to evaluate.

My field of expertise is tied to understanding the regulation of cellular metabolism through the use of biochemical and biophysical techniques. I am not as familiar with the in depth details of proteomic analysis such as those required for accurate reporting of data tied to protein proximity labeling (BioID) methods.

3. How much time do you estimate the authors will need to complete the suggested revisions:

Estimated time to Complete Revisions (Required)

(Decision Recommendation)

Cannot tell / Not applicable

Yes

Review #2

1. Evidence, reproducibility and clarity:

Evidence, reproducibility and clarity (Required)

****Summary:**** The authors provide evidence to reveal the novel functions of WDR62 protein in maintaining the stability and activity of purine metabolic enzymes and overall

purine homeostasis. WDR62 interacted with BAG2, and they are recruited to purinosome. WDR62 loss caused accelerated degradation of purine salvage enzyme HPRT, and led to the accumulation of purine nucleotide intermediates.

While this study is compelling and significant for the field of neurodevelopmental disorders including microcephaly and purine metabolism, there are several concerns that should be addressed before publication.

****Major comments:****

- Although all experiments are conducted using non-neuronal cultured cells, does this phenomenon also occur in neuronal cells?
- What is the interaction between endogenous WDR62 and Bag2? This is because in overexpression systems, multiple chaperones may interact with the target protein during protein folding.
- Is endogenous WDR62 also present in the purinosome in purine depleted or sorbitol condition?
- Regarding Fig6 and Fig7, when HPRT decreases and inosine accumulates in WDR62-KO condition, did the levels of hypoxanthine, xanthine, and uric acid change?
- Does HPRT and the three microcephaly-associated WDR62 mutants also recruited in the purinosome in purine depleted or sorbitol condition?
- In Fig7C, HPRT/tubulin ratio appears to decrease in WT from 0hr to 24h, but the graph does not show this decrease. Additionally, quantification of PFAS(FGAMS) and BAG2/tubulin should be performed.
- Fig7D is problematic. HPRT in WDR62-KO cells seems to localize in the nucleus, possibly due to stronger exposure in KO conditions compared to WT. Also, the line scan is drawn in areas with low signal in WT. The comparison should be performed in areas with high perinuclear signal.
- The localization of HPRT should be compared in WT and WDR62-KO with BAG2 siRNA. It is also possible to confirm whether the phenotypes observed in KO, such as cell proliferation and xanthosine/inosine levels, are rescued.
- It should be considered that the induction of Bag2 in WDR62-KO might allow purinosome formation to proceed normally due to compensation. The co-localization of WDR62 and purine enzymes during purinosome formation should be compared when BAG2 expression is suppressed. Similarly, any changes in BAG2 localization in WDR62-KO should be examined. Furthermore, the purinosome formation ability should be compared in WDR62KO + Bag2 siRNA condition.
- The reduction of HPRT in WDR62-KO should be examined for potential effects of

enhanced degradation via the ubiquitin-proteasome system or the autophagy-lysosome system.

- Although it is known that HPRT-KO mice do not exhibit any effects on normal brain development except in some dopaminergic neurons, what are your thoughts on this?

****Minor comments:****

- Please write the full name before the abbreviation of the gene.

- There is no measurement data for Fig7C, and a measurement line is drawn only in one panel of the ROI.

- The line 488 "Fig11" looks like a typo.

- The table could not be found.

- It is strange that all measurement values for WT or control in Fig2, Fig7, and FigS9 are exactly 1.0 without any variation. Please check the measurement method again.

- Please write the method for purine depleted medium.

****Referees cross-commenting****

I concur with the accurate point observations by the other reviewers. The authors should address the most of the comments provided, as many of the suggested experiments are feasible. If the paper aims to elucidate the one of the causes of microcephaly, specifically, the issues related to cell type and endogenous proteins experiments need to be resolved, and addressing these issues would substantially enhance its quality and impact.

2. Significance:

Significance (Required)

Most of the roles of purinosomes in the central nervous system remain unknown. The discovery that the WDR62/MCPH2 gene, responsible for microcephaly, is related to purinosomes will have a major impact on this field. Additionally, the ability to easily induce purinosomes through sorbitol phase separation is a significant technical advance in terms of cost and simplicity. Furthermore, many genes related to microcephaly, such as MCPH, are factors directly involved in cell division by regulating the mitotic spindle and centrosomes. This study has revealed a new role for WDR62, uncovering part of a novel molecular mechanism for microcephaly.

3. How much time do you estimate the authors will need to complete the suggested revisions:

Estimated time to Complete Revisions (Required)

(Decision Recommendation)

Between 1 and 3 months

4. Review Commons values the work of reviewers and encourages them to get credit for their work. Select 'Yes' below to register your reviewing activity at Web of Science Reviewer Recognition Service (formerly Publons); note that the content of your review will not be visible on Web of Science.

Yes

Review #3

1. Evidence, reproducibility and clarity:

Evidence, reproducibility and clarity (Required)

****Summary:****

In the present work, authors describe a novel role of the microcephaly associated protein WDR62 in purine metabolism under cell stress conditions. In the proposed cellular model (AD293 WDR62 overexpression system), the WDR62 proximity binding partners are firstly identified and categorized according to their functional role in the cell (protein folding, purine metabolism, and stress granules). Among them, authors focus on BAG2 - a HSP70/90 co-chaperone involved in cellular stress responses. After the characterization of the WDR62-BAG2 physical interaction sites, suggested to be disrupted by WDR62 pathogenic mutations, their functional interaction in cellular stress responses is investigated. WDR62-associated granules are extensively characterized for their physical and dynamic properties under different conditions (i.e., hyper-osmotic stress). Further, through the evaluation of N- and C-terminally truncated form of WDR62 authors characterize the protein regions responsible for WDR62-containing granule condensation - suggesting a potential mechanism disruption in the event of pathological WDR62 mutations. Lastly, authors provide evidence that WDR62 condensation does not occur in canonical stress granules but in the so called-purinosomes, where it participates in the regulation of purine metabolic pathway stabilizing HPRT (purine salvage enzyme) via

WDR62-BAG2-HSP70/90 axis.

****Major comments:****

Overexpression system and the employed cell line are a major limitation of the study. There is no experimental data on human neural cells and on endogenous WDR62, underestimating the potential difference in cell type-specific metabolism. In light of this consideration, the provided introduction and conclusions on neural development and microcephaly have to be re-formulated. I suggest providing a more general introduction/conclusions on WDR62 role (and alterations) in cell division and cell metabolism (neurodevelopment and cancer share common patterns) since purine homeostasis is not exclusive of neural progenitor cells.

This reviewer thinks that the structure of the work is a bit convoluted (too many results in main figures that are not substantial). I suggest to re-organize and to prioritize the most relevant results. Further, it would be clinically relevant to add WDR62 mutant constructs in the functional evaluation of purine metabolism to better dissect the physiological role of WDR62 and the impact of the mutations on cellular physiology.

Fig. 1: Overexpressed WDR62 fluorescence signal might be artifactual and may hide more detailed localization pattern during interphase. Authors should also provide endogenous WDR62 immunofluorescence panel in AD293 cells. Additionally, the "cytosolic" localization of WDR62 during interphase (indicated in the introduction, lines 88-89) has been re-defined in recent works pointing out that the protein is dynamically associated with the interphasic centrosome, the Golgi apparatus, and spindle poles during mitosis.

Fig. 1C lacks quantification of BAG2/CEP170/AURKA signal. Further, how can authors exclude that is not nocodazole effect on microtubules disruption which impairs WDR62 spindle pole localization and therefore protein-protein interactions? A panel showing that "low dose" nocodazole do not impinge endogenous and exogenous WDR62 localization in mitotic cells is needed.

Fig. 3 H-J: The fluorescence signals are saturated (also evident in the intensity profile plot) and thus not applicable for any analysis. Further, how these linear ROIs are chosen? The signal pattern is not homogeneously distributed in those images. Please provide a more consistent fluorescence analysis.

****Minor comments:****

Abstract, line 49: How can these WDR62 mutations can result in a complete loss of the protein ("In cells lacking WDR62") if authors report co-IP experiments (Fig. 2) with clear mutant WDR62 bands? Rephrase accordingly.

Result referred to Fig. 2D reports that "BAG2 co-immunoprecipitated with WDR62(N)-EGFP but not WDR62(C)-EGFP". The blot and the relative quantification in figure 2D instead show BAG2 signal in the WDR62(C)-EGFP - even if significantly lower. Please rephrase accordingly.

Lines 186-187: authors declare that the C-terminal tail comprising the helix-loop-helix domain is required for BAG2 to bind full-length WDR62. There are no such data in support of this. The C-terminal fragment includes both the disordered region and the dimerization domain. How can authors conclude that the dimerization domain alone is sufficient to bind BAG2?

Lines 189-190: results in AD293 cell line are not directly applicable in demonstrating that poor WDR62-BAG2 interaction can lead to alterations in brain development. Please rephrase.

Line 196: Indicate here, as the first mention, stress granules as "SGs" and use the abbreviation consistently throughout the manuscript.

Line 255: are human neural progenitor cells enough sensitive to sorbitol? If not, the proposed experimental design is a bit artifactual and the results/conclusions cannot be related to neural development alterations. I suggest applying more "physiological" stressors and frame the results in meaningful neurodevelopmental/tumorigenic environment. Please add this point to the discussion.

Line 240: WDR62 granules association with microtubules and especially mitochondria is not convincing (Fig. S5). This data seems to be a bit qualitative, please provide more detailed quantification of this parameter.

Fig. 4 is convoluted. I suggest moving some data to supplementary to improve the clarity of the figure.

Line 273: "Liquid-like protein condensates also exchange their contents with the bulk cytosol [52]". Reference 52 reviews the existing literature referred to biomolecular

condensates that exert nuclear function (e.g., genome organization, gene expression, and DNA repair). No mention on events involving cytoplasm. Please add a more relevant reference.

Lines 290-291: have authors considered the effect of sorbitol on microtubules dynamic that might reflect in granules dynamic changes?

Line 295: I suggest moving the prediction of the disordered region of WDR62 when first mentioned (e.g., Supplement referred to Fig. 2)

Fig. S6C-E, I: Unclear which is the criterion by which a cell is marked as "with" or "without" granules.

Fig. S8: Unclear, also from the micrograph showed in the figure, how authors have counted/considered the microtubules/mitochondria associated purinosomes. Seems very qualitative and observer dependent. Please provide a more reliable analysis.

Fig. 6A: The same blot of WDR62 KO is shown in Fig. S7. Please remove one.

Fig. 6C, D: Method for cell proliferation measure is indirect and "rounded cells" as indicator of cell death is sub-optimal. Analysis with specific markers would be preferable in both cases.

Fig.7B: Why the transfection control vector "EGFP only" significantly increases/decreases the BAG2/HPRT expression with respect to the negative control?

Paragraph from line 410 to 434: very confusing, the reported results are not well conveyed and therefore not convincing. To be reformulated.

Lines 524-526: the author's conclusion that: "...the loss of purine metabolic enzymes, including HPRT, disrupts neurogenesis, resulting in microcephaly, cell cycle defects, ciliopathies, and abnormalities in proliferation and neural progenitor fate decisions, mirroring the loss of WDR62." is not supported by the cited literature [29] and by the results presented in this work. Please provide additional references or remove the statement.

Lines 527-529: if authors state that "...other WD repeat-containing and microcephaly-associated proteins interact with purine enzymes..." have to provide additional references in addition to the NWD1 one. Otherwise, these lines should be rephrased as "another WD

repeat containing and microcephaly-associated protein...".

Reference 62 is not well indexed in the Reference section. Please adjust.

****Referees cross-commenting****

This reviewer thinks that the points raised by reviewer #1 and #2 are very accurate and significant. Some of them are also shared between our three review reports and in general are referred to: clarity of the manuscript improvement, little consistency between the results displayed in the figures and the text/conclusions in some points, concerns about the reliability of some measurements/result and the employed cellular model, and the lack of endogenous protein data.

2. Significance:

Significance (Required)

General assessment:

The here described new role of WDR62 in purine metabolism and the proposed pathway are novel and relevant to shed light on pathophysiological cellular and molecular mechanisms that potentially underlie neurodevelopmental defects and carcinogenesis - processes in which WDR62 is implicated. The experimental design is extended and generally well-conceived even though quite dispersive in some points.

The strength of the work resides in its versatility - making these findings potentially applicable to different cell types and different contexts (e.g., from neural development to malignancies) - and in the protein-protein interactions characterization under several conditions.

Similarly, the major weakness is the generalist trait of the findings that describes WDR62 cellular behavior mostly in an over-expression system in an immortalized cell line, underestimating the intrinsic metabolic and protein expression-level differences among cell types.

Advance:

WDR62 is a scaffold protein with pleiotropic functions and a plethora of molecular interactors. Literature reports many molecular pathways involving WDR62 mainly in cell

cycle progression, primary cilia biogenesis and centrosomal functions in a neurodevelopmental context. In the present work, authors describe mechanistic insights of a never reported WDR62-BAG2-HSP70/90 molecular pathway shedding new light on the role of this protein in cellular metabolism thus providing a new perspective on WDR62 pathophysiological functions.

Audience:

Basic research audience will be interested in this research work. The described molecular pathway involving WDR62 in purine metabolism might be relevant to other research on how WDR62 cellular and molecular dynamics are impactful on neural development and malignancies.

Expertise:

Human neural development and alterations, iPSCs, neural stem cells, CRISPR-Cas9

3. How much time do you estimate the authors will need to complete the suggested revisions:

Estimated time to Complete Revisions (Required)

(Decision Recommendation)

Between 1 and 3 months

Yes

Full Revision

Manuscript number: RC-2024-02582

Corresponding author(s): Dominic Ng

[Please use this template only if the submitted manuscript should be considered by the affiliate journal as a full revision in response to the points raised by the reviewers.]

*If you wish to submit a preliminary revision with a revision plan, please use our "Revision Plan" template. **It is important to use the appropriate template to clearly inform the editors of your intentions.**]*

1. General Statements [optional]

We thank the reviewers for their thoughtful and constructive feedback on our manuscript submission. In response, we have undertaken substantial revisions to both the experimental work and manuscript text and have made concerted attempts to address each comment in detail. As suggested by all reviewers, we have worked to refine and focus the narrative of our study, which reveals a novel role for the microcephaly-associated protein WDR62 in the regulation of purine metabolism. This revised manuscript now presents a more cohesive and integrated story, emphasizing the mechanistic link between WDR62 and purine regulation via its interaction with BAG2 co-chaperone and HPRT purine salvage enzyme.

To strengthen our conclusions, we performed numerous additional experiments as requested by reviewers, including proximity ligation assays to validate endogenous protein interactions, metabolomic flux analyses to assess purine synthesis and salvage pathways, and *in vivo* studies to compare the neurodevelopmental impact of HPRT and WDR62 depletion. We also clarified and corrected inconsistencies in data presentation, figure references, and statistical analyses, and provided expanded methodological details to ensure reproducibility.

We have revised the manuscript to better delineate the relationships between WDR62, BAG2, HPRT, and purine biosynthetic enzymes, and softened speculative claims where appropriate. Moreover, we removed several figures in the main text and supplementary material relating to the phase condensation of WDR62 granules and reorganized figures to improve clarity and focus, and incorporated reviewer suggestions to enhance the significance and impact of our findings.

We believe these revisions have substantially improved the manuscript and hope that the updated version meets the expectations of the reviewers and the broader scientific community.

This section is mandatory. Please insert a point-by-point reply describing the revisions that were already carried out and included in the transferred manuscript.

Reviewer #1 (Evidence, reproducibility and clarity (Required)):

In Morris, M.J. et al., the authors strive to better understand the roles for the microcephaly protein WDR62 in brain growth and function. To accomplish this, an in situ biotinylation assay was performed and identified 42 proteins proximal to WDR62 including the Hsp70 co-chaperone BAG2. Through a series of co-immunoprecipitation assays, the BAG2-WDR62 interaction was shown to be mediated through the structured N-terminal region of WDR62, and it was proposed that common WDR62 mutations disrupt this interaction. In AD293 cells, loss of WRD62 expression resulted in an increase in the expression of BAG2 expression while reducing HPRT expression. Subsequent loss of BAG2 expression by siRNA treatment restored the expression of HPRT suggesting that there is an association between the stability of HPRT and BAG2, likely mediated through its proposed association with Hsp70/90 molecular chaperones. Finally, the authors investigate the subcellular localization and ability of WRD62 to phase separate. WRD62 was shown to form discrete condensates induced by sorbitol-mediated hyperosmotic stress. The formation of WRD62 granules are hypothesized to be driven by cell volume exclusion and macromolecular crowding. These granules appear similar, both in physical appearance and characteristics, to other reported biomolecular condensates such as those reported in metabolism (e.g. purinosomes). WRD62-containing condensates were shown to colocalize with enzymes in de novo purine biosynthesis; however, this association is not required for purinosome formation and/or its stability under both purine-depleted and sorbitol-driven growth conditions. Overall, the manuscript provided a previously unrealized and exciting association between WDR62 and purine metabolism.

1. EVIDENCE, REPRODUCIBILITY AND CLARITY

Summary: The current manuscript reads as multiple manuscripts with findings that are at times weakly connected (in my opinion). For example, I had a hard time understanding how the BioID results relate to the discovery of WRD62 phase-separation and its colocalization with purinosomes. I would strongly encourage the authors to consider dividing the results into separate manuscripts to strengthen their claims and create a more focused and cohesive manuscript (or series of manuscripts). I believe then several of my reservations associated with the current manuscript will be addressed, and in my opinion, the hard work from the authors will be better received across the scientific community.

Response: We thank Reviewer #1 for acknowledging the novelty of our work and appreciate the constructive feedback regarding the lack of integration among individual findings. In response, we have removed content related to condensate formation and conducted additional experiments to more thoroughly characterize the mechanisms of WDR62 interaction. These new data, along with revisions to the manuscript text, have strengthened the coherence of our findings. We believe the revised manuscript now presents a more unified narrative, highlighting the complex roles of WDR62 in regulating purine metabolism.

I would like to commend the authors for all the work that went into the current version of the manuscript. Being part of a biochemistry and cell biology research group, I completely understand how much time and effort must have went into generating these data. That being said, I felt that there were several instances where clarification and additional information is warranted to arrive at the conclusions made by the authors. These points are outlined below.

Major Comments:

1. There appears to be a discrepancy between the data presented in Figure 1 and what is stated in the main text. Clarification is necessary to better understand the results:

- The following statement (and derivatives of it) are repeated throughout the manuscript: "...we found that the WDR62 interactome comprised molecular chaperones such as HSP70, HSP90, and their co-regulators, BAG2, STIP1, and DNAJC7" (lines 91-93, 316-318, 422-425). STIP1 and DNAJC7 were not identified in the list of 42 proximal proteins to WDR62 (Figure 1D). DNAJC7 was included because of a previous report curated in the BioGRID database, and there is no mention of HSP90 in the data produced in Figure 1. Please revise the main text to reflect the data that was generated.

Response: We thank the reviewer for this valid point and highlighting the instances where our description of results did not accurately reflect the data generated. We have reworded the relevant sections (e.g. lines 105-107,) in our revised manuscript to better delineate interactors identified in BioID studies (BAG2) as opposed to those previously reported on protein interaction databases such as BioGRID (DNAJC7).

- Based on the data presented in the Venn Diagrams in Figure 1D, the author's numbers do not seem to be consistent with the sentence on lines 126-128. I count 37 proteins unique to their BioID study, 90 unique to the BioGRID database, and 5 proteins that overlap between the two data sets. This sentence needs to be revised.

Response: We thank the reviewer for pointing out this inconsistency. There were 95 protein interactors of WDR62 from BioGRID while we identified 42 proteins in our BioID study with 5 proteins overlapping. We have revised the main text (lines 144-146) and Fig. 1D to accurately reflect the protein numbers identified.

- What data were used to generate the interaction map in Figure 1I? Enzymes tied to purine

metabolism were not identified from the data presented in Figure 1D but have now appeared. A discussion of this in the main text is warranted.

Response: We generated the interaction map in Fig. 1I using STRING to visualise WDR62 protein-protein interactions derived from both the BioGRID database and our BioID analysis. As the reviewer correctly points out, purine metabolic enzymes were not direct interactors of WDR62. Purine enzymes are linked to the molecular chaperones which, in turn, associated with WDR62 from our BioID analysis. The links between purine enzymes and chaperones were obtained from the BioGRID database. In response to this feedback, we have revised our manuscript to include a more detailed description of how the interaction map in Fig. 1I was generated, both in the main text (lines 148-157) and the legend for Figure 1. The BioGRID interactions between heat shock proteins and purine enzymes were introduced in the manuscript text at lines 264-266.

2. This reviewer has several reservations on how the various key players in the manuscript are related to substantiate the conclusions made in the manuscript. For instance, how is HPRT, purinosomes, and WDR62 related? How about HSP90, WRD62, and HPRT? Pairwise connections were made throughout the manuscript; however, trying to tie all three together is difficult with the data presented.

- The authors tried to connect HPRT, purinosomes, and WDR62 with BAG2; however, this study could greatly improve if we understood how a knockdown of BAG2 impacts purinosome formation and/or WDR62 colocalization with purinosome enzymes.

Response: We have incorporated additional experiments in our revised manuscript to better connect HPRT, WDR62 and BAG2. Using proximity ligation assays (PLA) we demonstrated endogenous interactions between WDR62 and BAG2 (Fig. 4K), as well as between WDR62/HPRT and BAG2/HPRT (Fig. 6I-J). The interaction between BAG2 and HPRT was decreased in WDR62 KO cells (Fig. 6J), and recent experiments revealed that BAG2 depletion similarly disrupted the WDR62/HPRT interaction. These findings suggest that WDR62 expression, and presumably its interaction with BAG2, is necessary for BAG2-mediated regulation of HPRT.

Further, we found that the loss of HPRT expression in WDR62 KO cells was reversed by siRNA depletion of BAG2 (Fig. 6K), supporting our model in which elevated BAG2 levels in the absence of WDR62 promote aberrant HPRT degradation. Collectively, our results suggest that proper BAG2 regulation of HPRT requires WDR62.

To address the reviewer's suggestion, we also examined WDR62 cytoplasmic localisation following BAG2 depletion and found that BAG2 was not required for WDR62 to form granules in response to osmotic stress. We also show that BAG2 is not responsible for purinosome assembly or for the subcellular distribution/localisation of HPRT.

- Is HPRT a client of HSP90? And how are WRD62 and HSP90 related since they do not associated (based on your BioID data)? These connections would again strengthen the

arguments made in the manuscript and help to explain the HSP90 inhibition data presented in Figures 7F and 7G.

Response: Although our BioID data did not explicitly identify an association between WDR62 and HSP90, we initially focused on HSP90 due to the established role of BAG2 in protein misfolding and degradation through its interaction with HSP90 (doi: 10.7150/thno.78492). We hypothesised that while WDR62 may not directly interact with HSP90, its interaction with BAG2 could provide an indirect link. To strengthen our conclusions and address the limitations of our HSP90 inhibition data (NVP-AUY922), we performed additional experiments using a second HSP90 inhibitor (17-AAG) and an HSP70 inhibitor (MKT-077) across both short (1 h) and long (24 h) treatment durations (Fig 6 and Fig S10). Further details are provided in our response below to minor comment #1.

3. Caution is warranted when making conclusions about WDR62 (and its granules) and purinosomes.

Response: We acknowledge the reviewer's feedback and have revised our manuscript to focus on the functional characterisation of WDR62 interaction and co-localisation with BAG2 and related HSP co-chaperones. As part of this revision, we removed the FRAP studies and sections discussing WDR62 phase separation and purinosome assembly (further details below). Additionally, we have softened our description of cytoplasmic WDR62 granules as purinosomes. Instead, we describe WDR62 as forming dynamic puncta containing purine enzymes and discuss the possibility that these granules may represent or overlap with *bona fide* purinosomes.

• The authors describe the association between WDR62 and purinosomes differently throughout the text. I would recommend that the authors come to some conclusion about this and be consistent.

Response: We thank reviewer #1 for pointing out inconsistencies in our conclusions regarding WDR62 and purinosomes between sections of our manuscript. We have revised our manuscript to ensure our description of these findings are consistent throughout. Specifically, our findings show that WDR62 responds to osmotic and metabolic stress by forming dynamic cytoplasmic granules that share many protein components with purinosomes (Fig. 5). This suggests that WDR62 may be a novel component of bona fide purinosomes or that WDR62 granules substantially overlap with purinosomes both spatially and compositionally. However, the formation of granules by purine enzymes was not perturbed by WDR62 KO (Fig. S6). Thus, we conclude that while WDR62 colocalized with purine enzyme containing granules consistent with purinosomes in response to cell stress, WDR62 was not required for granule formation by purine enzymes such as PFAS and PPAT.

A. (Lines 339-340) "WDR62 granules represent or overlap substantially with the phase-separated metabolons known as purinosomes". Based on the data presented, it appears that these might still be different entities but either overlap or have similar components. Purinosome localization with mitochondria (approx 60-80%) and microtubules (approx 90-95%) were significantly higher than those reported for WDR62 granules (approx 40%). This comparison would suggest that not all WDR62 granules behave similarly to purinosomes. And from the dot

plot in Figure 3G, about half of the characterized WDR62 granules do not align with the previously reported characteristics of purinosomes.

Response: In Fig. 3G, we measured the diameter and distribution of WDR62 granules and found their size and number per cell closely matched those reported for BAG2 condensates (doi: 10.1038/s41467-022-30751-4). This aligns with our findings that WDR62 interacts with BAG2 and is recruited to similar subcellular compartments. The reviewer correctly notes that WDR62 granules only partially align with previously reported characteristics of purinosomes, suggesting that they may be distinct entities. Our revised manuscript acknowledges this possibility while also emphasizing that WDR62 granules share features and colocalise with many purinosome components. To enhance the focus and clarity of the manuscript, we have removed Fig. 3G as the diameter and number of WDR62 granules are already reported in Fig. 3F.

B. In the abstract and introduction, the authors state that WDR62 is being recruited to the purinosome and leave out the other possibility. I would recommend that the authors soften this claim in these sections because of the above possibility but also the lack of characterization of the sorbitol-induced "purinosomes". There is little discussion or evidence for how sorbitol induces purinosome formation. Is de novo purine biosynthesis activated upon sorbitol treatment? Are multiple de novo purine biosynthetic enzymes present in the sorbitol-induced "purinosomes"? Further, I agree that there is a tendency for WDR62 to associate with condensates that bear an enzyme within de novo purine biosynthesis; however many of these proteins are known to self-aggregate upon cell stress. Therefore, the entities that are being observing and called purinosomes might not be bone fide purinosomes. Additional care is necessary to make these statements. In my opinion, the current data only suggests that this might be a possibility.

Response: As indicated, we have softened our claim that stress-induced WDR62 granules represented bona fide purinosomes. Fig. 3 of our revision more precisely describes the characteristics of WDR62 granules while Fig. 4 now reports on the co-localisation of WDR62 granules with protein chaperones and de novo purine synthesis enzymes typically associated with purinosomes. We now conclude that WDR62 may be associated with purinosomes but may also represent distinct entities with shared components and characteristics. Notably, proteins such as BAG2 and PFAS may undergo phase separation in response to stress independently of purinosome assembly.

In additional work conducted for our revised manuscript, we find that WDR62 loss reduced rates of purine synthesis in cells cultured in the presence of purines (Fig. 5) but was not involved in de novo purine biosynthesis under purine-depleted conditions (Fig. S9). This was consistent with the finding that WDR62 loss did not prevent stress-induced formation of PFAS or PPAT granules (Fig. S6) which are likely to represent purinosomes. We concede that additional investigation is required to determine the functional significance of WDR62 granules in response to stress stimuli and purine depletion.

- (Lines 325-329) The authors reference a previous manuscript demonstrating that co-

chaperones co-cluster with purinosomes. Based on this fact, they infer that WDR62 granules might represent purinosomes since WDR62 interacts with these same set of co-chaperones. These co-chaperones interact with a large number of different proteins (in fact, most kinases), so it is uncertain how the authors decided to go down this path to link purine metabolism with WDR62. Discussion of how this connection was made would help elevate the story. What additional insights did they have that lead them down these investigations?

Response: BAG2 functions as a co-chaperone that regulates the activity of HSP70/90. While the reviewer correctly points out that co-chaperones such as BAG2 have a broad number of clients, numerous studies have established the role of HSP70/90 in purine metabolism (e.g. doi: 10.1016/j.isci.2020.101058, 10.1073/pnas.1300173110) and in neurodevelopment (10.3389/fnins.2018.00821). Moreover, purines are critical for normal brain development and dysregulation is well known to lead to congenital defects including microcephaly. As such, when we identified a role for WDR62 in the chaperone network through interaction with BAG2, it was not a leap to hypothesise that neurometabolic defects stemming from dysregulated purine production or salvage might be involved in WDR62-associated microcephaly.

Indeed, we show that WDR62 are localised with purine enzymes in response to purine-depletion and that WDR62 depletion leads to metabolic dysregulation. WDR62 has several binding partners with multiple cellular functions, and we do not exclude alternative mechanisms involved in cortical development. However, the mechanistic link with heat shock proteins and purine metabolism is a novel one that would be of broad interest in molecular neurodevelopmental biology. On this feedback, we have revised main text (lines 214-218, lines 260-263, lines 292-295, lines 378-383) to better explain the rationale underlying our experiments and overall study focus.

If WDR62 is not required for purinosome formation, why would it localize with the purinosome? Is there any hypothesis that could be readily tested to better help understand this observation? Providing a better understanding of this would greatly elevate the work.

Response: Given the role of HSP70/90 in purinosome assembly and the interaction of WDR62 with BAG2, and purine enzymes PFAS and PPAT, we were initially surprised that WDR62 depletion did not affect stress-induced PFAS and PPAT granule formation (Fig. S6). At the time of writing the original manuscript, we interpreted these granules as purinosomes. However, it remains possible that WDR62 might have a function in purine synthesis or in purinosome assembly that remains unidentified. Indeed, we have not yet tested different cell types or additional conditions that induce purinosome formation or determined the localisation or activity of other purine synthesis enzymes. Thus, we concede our conclusions on WDR62 and purinosome formation were premature.

As our revised manuscript is now focused on the WDR62-BAG2-HPRT interaction and given the reviewer's prior comment that PFAS and PPAT colocalization in granules may not represent purinosomes in all contexts, we acknowledge that potential WDR62 functions in purinosomes warrants further investigation beyond this study. In the revised discussion (lines 473-497) we address these limitations and propose alternative interpretations of our findings.

A. (OPTIONAL) Please validate that the associations between WDR62 and the purine biosynthetic enzymes occur on the endogenous level (void of transient transfection). Many methods such as immunofluorescence and proximity ligation assays have been used by others to demonstrate protein-purinosome interactions. This result would reduce any concern that the association is a result of overexpression (artifact).

Response: As suggested, we conducted proximity ligation assays (PLA) to validate endogenous interactions between WDR62 and BAG2, HPRT, and PFAS (Fig. 4K, Fig. 6I-K). Notably, sorbitol treatment increased the interaction between WDR62 and HPRT (Fig. 6H, I), supporting the role of WDR62 in regulating HPRT under cellular stress. Additionally, WDR62 deletion appear to reduce the interaction between BAG2 and HPRT (Fig. 6K), while BAG2 depletion similarly reduces the interaction between WDR62 and HPRT (Fig. 6J). These findings support a model in which WDR62 and BAG2 cooperatively regulate HPRT stability.

B. Figures 6F and 6G conclude that nucleosides from purine-depleted growth conditions accumulate while the corresponding monophosphates do not change between WRD62 knock-out and wildtype cells. Given that purine-depleted growth conditions activate *de novo* purine biosynthesis (uncertain if this has been demonstrated in AD293 cells), could this result simply demonstrate that purine salvage is no longer used and the nucleosides have accumulated and are awaiting degradation (or exportation) rather than a loss of HPRT expression as inferred from the stated conclusions? The conclusions could be better substantiated with the use of a stable isotope incorporation assay.

Is there a difference in the contribution of *de novo* purine biosynthesis and purine salvage to the generation of the monophosphates (AMP, GMP) between WDR62 knockout and wildtype AD293 cells? Use of a stable isotope (such as ^{15}N -glutamine) could help to come to the appropriate conclusion.

Response: We thank the reviewer for this helpful suggestion to better characterize WDR62-dependent purine defects in more detail. In our revised manuscript we performed targeted metabolomics experiments and tracked the incorporation of $^{13}\text{C}_2$ -glycine and $^{13}\text{C}_5$ -hypoxanthine into purine nucleosides to assess purine synthesis and purine salvage flux between WT and WDR62 KO cells (n=5). Indeed, purine nucleotides in KO cells showed a significant loss of incorporation of $^{13}\text{C}_2$ from $^{13}\text{C}_2$ -glycine, consistent with impaired *de novo* synthesis in cells cultured in presence of purines. In contrast, labelling from $^{13}\text{C}_5$ -hypoxanthine showed no overt differences between WT and KO cells, suggesting that incorporation via the salvage pathway is not grossly altered under these conditions. We have subsequently added a section to the discussion (lines 498-521) to discuss these results which suggest that the reduced HPRT levels in KO cells may be sufficient to sustain rates of purine salvage which are not altered with WDR62 loss. Thus, the accumulation of nucleosides is most likely due to increase conversion from monophosphates or reduced degradation to uric acid. Nonetheless, we show that WDR62 is required for purine synthesis under basal conditions and has a complex role in regulating purine metabolism.

(Lines 483-485) If there is a change in de novo purine biosynthesis, are there any detectable changes in AICAR levels that might influence purine metabolism at the transcriptional level?

Response: This remains a possibility. However, we did not detect the AICAR intermediate in our untargeted LC-MS/MS metabolomics analysis perhaps due to low relative abundance and/or low stability. As a result, we were unable to comment on AICAR levels but this would be an interesting research direction to pursue in subsequent follow up studies.

Are the data and the methods presented in such a way that they can be reproduced? Are the experiments adequately replicated and statistical analysis adequate?

1. For purine-depleted studies (metabolite analyses, microscopy), how long were the cells grown in purine-depleted medium before the analysis? And how was the purine-depleted medium generated? Please reference any source that might have been used.

Response: We removed purines from the cell culture environment by incubating cells for 7 days with DMEM supplemented with FBS dialyzed to remove small molecules such as nucleosides and nucleobases. This important methodological detail was omitted in error in our original submission. Our revised manuscript includes description of how we depleted cells of purines in the Materials & Methods at Lines 636-640 with reference to source materials and prior studies.

2. Details describing the BioID experiment are minimal. How many replicates were performed, was label-free or TMT quantitation used for the protein identification. Further the data analysis and mining of the proteins from the BioID study are missing - What database was used to identify the proteins from the peptides? Please include this information in the Materials and Methods section as well as a link to a repository where the LC-MS/MS data generated can be found. Additionally, it would be very helpful to have a spreadsheet or table that lists the biotinylated proteins and expectant or p values for each.

Response: We performed three independent biological replicates ($n = 3$) for the BioID experiment. We apologise for the omission and have now included this information in the Fig. 1 legend. Label-free quantitation was used for protein identification, and peptides were identified using the ProteinPilot™ Software (v. 4.5) database. As part of our revision, we have updated the Materials and Methods section to include these details and will also provide a spreadsheet listing all biotinylated proteins across replicates, including their p-values. Furthermore, we have submitted our LC-MS/MS data as supplementary files associated with this manuscript.

3. Please include information about the streptavidin pulldown presented in Figure 1C.

Response: Streptavidin pulldown followed by immunoblot for known WDR62 interacting proteins is described in our Materials & Methods section at line 753-759. Proteins bound to Streptavidin agarose beads were eluted with Laemmli buffer following washing. Pulldown fractions and total lysates were then resolved on SDS-PAGE, transferred to PVDF and blotted with primary antibodies to detect WDR62 interacting proteins such as CEP170, JNK and AURKA. We also used this method to confirm biotin-labelling and affinity isolation of BAG2 in Fig. 1C.

4. Many of the figure legends could benefit from a statistical description.

Response: As requested, we have updated the legends for all relevant figures and supplementary figures to include statistical descriptions, specifying analyses used and replicate (n) numbers. These additions complement the detailed description of our statistical methods provided in the Materials & Methods section (line 884).

5. There seems to be only two data points for Figure S3A. While there is no significant difference observed, it would be ideal to have additional replicates.

Response: We have completed an additional replicate and updated Fig. S3A for our revised submission. This study now includes $n = 3$ independent biological replicates. While we observed a slight increase in the proportion of cells with MAPKBP1 granules in response to sorbitol stress, this change was not statistically significant. In contrast, WDR62 formed granules in a much larger proportion of cells (~90%) in response to stress (Fig. 3E).

6. (Figure 5I) Please provide statistical analysis to demonstrate the colocalization between FGAMS and WDR62 is robust in purine-depleted AD293 cells.

Response: Our revised manuscript now includes three independent replicates assessing WDR62 co-localisation with PFAS in purine-depleted AD293 cells (Fig. 4I in revision). We consistently observed a high degree of co-localisation, as quantified by Pearson's correlation coefficient (mean = 0.8), which was significantly different from control conditions.

Minor Comments:

Do you have suggestions that would help the authors improve the presentation of their ideas and conclusions?

1. The use of HSP90 inhibitors is a little confusing given the connections being made with BAG2 and other HSP70 co-chaperones in Figure 1.

- Does the same conclusions hold true with an HSP70 inhibitor or siRNA?

- (OPTIONAL) There are a lot of discrepancies between Hsp90 inhibitors and effective treatment concentrations. For example, NVP-AUY922 caused purinosomes to disassemble whereas STA9090 cause purinosomes to change morphology and adopt a more aggregated state. Do other Hsp90 inhibitors share the same phenotypic response as NVP-AUY922 in this study.

- The treatment time (24 h) with NVP-AUY922 is very long. Given that Hsp90 interacts with hundreds of proteins, it is hard to understand whether the effect of Hsp90 inhibition is direct or indirect. Shorter times (1 h or less) would be more insightful.

Response: To address these specific comments on the specificity of effects from HSP90 inhibitor treatment, we have conducted additional experiments using NVP-AUY922, in addition

to another HSP90 inhibitor, 17-AAG, and the HSP70 inhibitor, MKT-077, at both 24-hour and 1-hour timepoints.

Our results demonstrate that NVP-AUY922 can rescue the aggregated HPRT phenotype in WDR62 KO cells even after 1 hour of treatment (Fig. 6F, G). Similarly, 17-AAG exhibits a comparable effect, reinforcing the role of HSP90 inhibition in modulating the spatial distribution of HPRT in the cytosol (Fig. 6F, G). Additionally, we found that MKT-077, a HSP70 inhibitor, also rescues the aggregated HPRT phenotype, with the effect being most pronounced at 24 hours but still evident at 1 hour (Fig. S10A, B). We also utilized BAG2 siRNA but determined that BAG2 depletion rescued WDR62 KO effects on HPRT expression (Fig. 6L) but did not reverse the effect on HPRT spatial distribution (Fig. S10C).

2. (OPTIONAL) Does the 2.6-fold increase in BAG2 increase its association with WDR62?

Response: We observed a ~2.6-fold increase in BAG2 levels following WDR62 deletion (Fig. 6A). However, as WDR62 is not present in KO cells, it is not possible to verify whether there would be an increase association with WDR62 and we did not conduct an experiment to overexpress BAG2 in WT cells. However, we presume that increased cellular levels of BAG2 would lead to increased pulldown with WDR62 by immunoprecipitation for example.

3. Is the degradation of HPRT occurring through BAG2-mediated proteasomal degradation? Showing HPRT recovery by treating the cells with MG132 along with CHX would provide meaningful clues as to how BAG2 and HPRT might be related - Is BAG2 concentration increasing to facilitate the enhanced degradation of HPRT?

Response: We thank the reviewer for this useful suggestion. However, our initial experiments with MG132 and chloroquine to inhibit proteasomal and autophagic pathways respectively gave mixed results. Our preliminary findings suggest neither was sufficient to substantially rescue HPRT levels in WDR62 KO cells. However, this needs extensive follow up with more precise dissection of cell degradation pathways with additional inhibitor or genetic targeting of degradation machinery. Thus, we did not include these studies in the revision and will instead include this in a follow up paper once we have completed a more comprehensive investigation.

4. Does HPRT colocalize with WDR62 in cells?

Response: In response to this comment, we have demonstrated that osmotic stress induces the spatial reorganisation of endogenous HPRT into puncta that juxtapose and co-localize with WDR62 granules in a stress-dependent manner (Fig. 6H). This was further validated by examining the endogenous WDR62-HPRT interaction using PLA, which also revealed a stress-induced increase upon sorbitol treatment (Fig. 6I).

5. (OPTIONAL) It would be nice to see validation experiments of some of the hits in Figure 1D or 1E in a co-immunoprecipitation experiment conducted similar to Figure 1C.

Response: Our BioID assay, presented in Fig. 1D and E, identified WDR62 interactors, such as AURKA, JNK, CEP170 and MAPKBP1, that have been previously validated by co-IP by our

group and others. Among the chaperones identified, we focused on BAG2 in this particularly study and validated BAG2-WDR62 interactions between by colP (Fig. 2) and by proximity ligation assays (Fig. 4).

6. The authors presented the findings that suggest that BAG2 interacts differently with commonly observed WDR62 mutations in MCPH2? How do these mutations affect WDR62 condensation, colocalization with purinosomes, or alter HPRT activity? Tying back the observations to something clinical would help elevate the overall significance of the findings.

Response: We investigated the condensation of mutant WDR62. Interestingly, R438H mutant, which binds BAG2 (Fig. 2), forms granules constitutively prior to stress treatment while the 3936dupC mutant, which does not bind BAG2 (Fig. 2), did not form granules in response to sorbitol stress treatment. We also find that PFAS is colocalized with R438H granules in the absence of stress, although this requires repeated analysis and quantification. However, WDR62 deletion does not prevent PFAS or PPAAT granule formation (Fig. S6) and, given reviewer advice to focus the topic of our revised manuscript, we have not included the effects of WDR62 mutations on granule formation in our revised manuscript.

However, in response to these comments, we have conducted rescue experiments with patient-identified MCPH mutant variants of WDR62. Expression of the R438H or 3936dupC mutant in WDR62 KO cells did not rescue HPRT to the same extent as full-length WDR62 with wild-type sequence (Fig. 6B). Additionally, attempts to restore BAG2 levels in WDR62 KO cells by expressing mutant WDR62 showed no discernible difference from full-length WDR62. Thus, mutations to WDR62 associated with MCPH alters binding to BAG2 (Fig. 2, increased with R438H and decreased with 3936dupC), this was associated with dysregulated levels of BAG2 and HPRT. In our revised manuscript, we also examined the effect of HPRT depletion on neurodevelopment *in vivo* (Fig. 7) and included description of these findings at lines 417-442.

Are the text and figures clear and accurate?

1. There are many times throughout the manuscript that the wrong figure is being referenced. These mistakes caused significant confusion at many times while reviewing the manuscript. Please double check all in-text references to figures. For example, I believe that you meant to use Figure S1C instead of Figure 2E with the statement on lines 183-185. Again, I believe that correct figure reference on line 501 is Figure 7G not Figure 7E.

Response: We apologize for this oversight. We have amended the errors indicated by the reviewer. Line 544 (501 in first submission) now refers to the correct figure (Fig. 6F) and lines 204-206 (183-185 in first submission) correctly refers to Fig. S1C in addition to Fig 2E. Each of the authors have also revised the rest of the manuscript to ensure all figures are correctly referenced in the main text.

2. The figure legend on Figure S4 does not match the figure and the main text references. Please verify that the text in the figure legends correspond correctly to the figure.

Response: We apologize for these inconsistencies in the figure legend relating to Fig S4 in our original submission. In the revised manuscript, we have amended the figure legend and the main text referencing Fig. S4 to correctly correspond to order of data panels in this figure.

3. Please provide this data for the sentence on lines 399-400 in the supplemental file.

Response: As requested, we have revised the manuscript to include results on HPRT cytoplasmic localisation following osmotic stress. We show that osmotic stress induces the spatial reorganisation of HPRT into puncta that juxtapose and co-localize with WDR62 granules in a stress-dependent manner (Fig. 6H). This was further validated by examining the endogenous WDR62-HPRT interaction using PLA, which also revealed a stress-induced increase upon sorbitol treatment (Fig. 6I).

4. I believe that the authors use the phrase "cell proliferation" to describe cell viability in the main text. In the Materials and Methods section, the authors state "The XTT cell proliferation assay enables quantification of cellular redox potential, providing a colorimetric readout of cell viability." Cell proliferation, viability, and cytotoxicity are different measurements, so please revise to reflect the correct experiment that was performed.

Response: The XTT colorimetric assay can be used to determine cell proliferation or loss of cell viability depending on the specific experiment. The reviewer is correct in pointing out that our study using XTT to measure cell numbers in the context of purine-depletion (Figure 5B) is a measure of cell viability. We apologize for the misleading text in our description of the XTT methods in our original submission. In our revised manuscript, we have amended our description of the XTT assay in our methods and in the figure legend to more accurately reflect the experiment performed.

Other Minor Comments:

7. Move the sentence "In contrast, despite reduced mRNA..." (lines 387-388) to the last section when a reduction in PFAS expression was first mentioned.

Response: As requested, we have moved this line referring to PFAS protein levels in WDR62 KO cells to the previous section to when a reduction in PFAS mRNA was first mentioned.

8. Please reference the following in the manuscript:

- BioGRID database in the main text and Materials and Methods section
- The reported study showing the DNAJC7-WDR62 interaction (as curated from BioGRID)
- Fiji in the Materials and Methods section

Response: We have now included references to these in our revised manuscript. References to BioGRID database are in the main text (line 146) and in the Materials and Methods (line 765). The report of DNAJC7-WDR62 interaction (Ref #37) curated from BioGRID was added at line 157 and reference (Ref #82) to Fiji plug-in was indicated at line 690 in Materials & Methods.

9. (Line 461-463) The authors state the following: "the loss of WDR62 leads to an increase in BAG2 and vice-versa (Fig. 7A) (Fig. S9B). I am not sure that the vice-versa (i.e. loss of BAG2

increases WDR62) is true. From the data presented in Figure 7H, I do not see a significant change in WDR62 expression upon BAG2 siRNA treatment.

Response: We apologize for the incorrect use of the term “vice versa” in this context. We had meant that while WDR62 loss led to an increase in BAG2, the converse increased expression in WDR62 resulted in a decrease in BAG2 levels. The reviewer is correct that the siRNA knockdown in BAG2 did not substantially alter WDR62 levels. We have amended the text at lines 465-467 to clarify this statement.

10. For your BioID study, do you know how many or the proportion of cells that were mitotically arrested with the low dose of nocodazole (200 ng/mL)? Given the small number of unique proteins that were in the mitotic only population, it is curious to know how enriched the cells were and whether WDR62 localization is important in the context of this study.

The overnight treatment with low dose nocodazole results in an enrichment of cells arrested in late prometaphase which we estimate at 50-60% of AD293 cells compared to <5% of the population in asynchronous culture. This was determined by imaging analysis of microtubule and chromatin morphology. WDR62 is localized to microtubule spindles in mitosis while remaining predominantly cytoplasmic in interphase. Our findings indicate that some proteins interactors bind WDR62 exclusively in mitosis (eg AURKA) while other interactors such as CEP170, MAPKBP1 and BAG2, while identified in mitotic fractions (Fig. 1E), appear to be downregulated with mitotic enrichment (Fig. 1 C). For these reasons, the dynamic localization is likely an important determinant of WDR62 interacting partners and specific cell function.

11. Just to clarify, the WDR62-HA lane (third in each set) in Figure 1C is not WDR62-BirA*-HA and that it is only being used as a control.

Response: This is correct. To improve clarity, we have amended the labels on the WDR62-HA lanes in Figure 1C to say “WDR62-HA *only*”.

12. In the Discussion (lines 439-441) "We also show that WDR62 forms dynamic, phase-separated granules that co-localise with chaperones and purine metabolic enzymes, resembling purinosomes." I believe that the authors meant to say co-chaperones instead of chaperones given no microscopy data was presented showing the colocalization of HSP70/90 with WDR62 granules. Please revise.

Response: This sentence (line 473) has been revised as suggested.

****Referees cross-commenting****

I agree with the comments and recommendations by the other reviewers. Many of our shared comments are those that need to be addressed to substantiate the claims made by the authors throughout the manuscript. The proposed experiments across the reviewer comments appear feasible given that similar experiments have already been presented in this version of the manuscript. I strongly encourage the authors to consider these comments when revising their manuscript to help strengthen their claims and boost its overall significance and impact.

Full Revision

Reviewer #1 (Significance (Required)):

Describe the nature and significance of the advance (e.g. conceptual, technical, clinical) for the field. Place the work in the context of the existing literature (provide references, where appropriate).

The work presented explains a previously unknown role for WDR62 in the regulation of purine metabolism. Despite all the hard work that was performed to reach their conclusions, the use of the AD293 cell line and the lack of correlating the common WDR62 disease-promoting mutations to the observed findings throughout the entire manuscript slightly reduced my enthusiasm for this work. The presented study leverages a lot of existing literature to establish connections between WR62, co-chaperones, and purine metabolic enzymes, with an emphasis on purinosome metabolon, a condensate comprised of the enzymes in de novo purine biosynthesis.

State what audience might be interested in and influenced by the reported findings.

The audience that might be interested in the reported findings would likely be those tied to biomolecular condensates in cellular metabolism and their connection to disease. I also feel that researchers that study microcephaly might be interested in this work. In my opinion, I believe that a broader readership could happen if additional studies were performed to make stronger connections between studies presented.

Define your field of expertise with a few keywords to help the authors contextualize your point of view. Indicate if there are any parts of the paper that you do not have sufficient expertise to evaluate.

My field of expertise is tied to understanding the regulation of cellular metabolism through the use of biochemical and biophysical techniques. I am not as familiar with the in depth details of proteomic analysis such as those required for accurate reporting of data tied to protein proximity labeling (BioID) methods.

Reviewer #2 (Evidence, reproducibility and clarity (Required)):

Summary:

The authors provide evidence to reveal the novel functions of WDR62 protein in maintaining the stability and activity of purine metabolic enzymes and overall purine homeostasis. WDR62 interacted with BAG2, and they are recruited to purinosome. WDR62 loss caused accelerated degradation of purine salvage enzyme HPRT, and led to the accumulation of purine nucleotide intermediates.

While this study is compelling and significant for the field of neurodevelopmental disorders including microcephaly and purine metabolism, there are several concerns that should be addressed before publication.

Response: We thank reviewer #2 for their constructive criticisms and supportive comments noting the statement reinforcing significance of our study in the field. We have made a meaningful and concerted effort to address the reviewer comments with extensive additional experimental work and substantial revision of our manuscript.

Major comments:

Although all experiments are conducted using non-neuronal cultured cells, does this phenomenon also occur in neuronal cells?

Response: To address this comment and reviewer concerns regarding the links between WDR62 and HPRT in a neuronal context, we performed in utero-electroporation to determine the effects of HPRT depletion on formation of neocortex in mouse embryos. We electroporated embryonic day 14 (E14) mouse brains with siRNA targeting *Wdr62*, and *Hprt* and assessed neural progenitor proliferation, migration and differentiation using immunofluorescence. We find that the loss of both WDR62 and HPRT leads to a similar precocious delamination of neural progenitors from the apical ventricular surface (Fig. 7). This process is the first step in neural migration and required to generate a diversity of cells, both self-renewed (eg. outer radial glia) and differentiated neurons and glial cells in the developing neocortex (doi.org/10.1146/annurev-cellbio-101011-155801). Interestingly, we also uncovered that HPRT loss promoted the self-renewal of delaminated intermediate progenitors (IPs) which is unlike impaired the self-renewal of neural progenitors observed following WDR62 depletion (Fig. 7). Thus, brain development is sensitive to HPRT levels and the HPRT depletion phenocopies WDR62 in cell delamination which supports a neural role for WDR62-HPRT. Moreover, our findings suggest WDR62 loss has more severe neurodevelopmental defects with hints at the complex metabolic functions of WDR62 (discussed in lines 563-577).

What is the interaction between endogenous WDR62 and Bag2? This is because in overexpression systems, multiple chaperones may interact with the target protein during protein folding.

Is endogenous WDR62 also present in the purinosome in purine depleted or sorbitol condition?

Response: In response to these comments and similar concerns by reviewer #1, we examined interactions between WDR62, BAG2, HPRT, and PFAS at the endogenous level by utilising

proximity ligation assays (PLA, Fig. 4+6). We determined a robust interaction between endogenous WDR62 and BAG2 (Fig. 4K), evident by abundant PLA puncta which were nuclear excluded and localised to the cytosol, consistent with our results in overexpression systems (Fig. 4). We also confirmed endogenous WDR62 interactions with purine enzymes PFAS (Fig. 4K) and HPRT (Fig. 6I) in a similar fashion. To determine whether sorbitol stress promotes their interaction, we assessed changes in the per cell numbers of these puncta in response to sorbitol stress. We confirmed that endogenous WDR62 interaction with HPRT was dependent on BAG2 (Fig. 6J). WDR62-HPRT interactions increased with sorbitol stress (Fig. 6I).

Regarding Fig6 and Fig7, when HPRT decreases and inosine accumulates in WDR62-KO condition, did the levels of hypoxanthine, xanthine, and uric acid change?

Response: In Fig. 5G we used an untargeted metabolomics approach that relies on identification databases such as MS-DIAL and associated spectral libraries. Unlike targeted approaches, this method does not always allow for the confident identification of all metabolites of interest. As a result, hypoxanthine, xanthine, uric acid, and other purine intermediates (e.g., AICAR) were not positively identified. This is likely due to limitations in database coverage, spectral similarity to other compounds, or constraints related to our extraction method.

Does HPRT and the three microcephaly-associated WDR62 mutants also recruited in the purinosome in purine depleted or sorbitol condition?

Response: In response to this, and a similar comment by reviewer #1, we examined whether endogenous HPRT co-localised with WDR62 granules induced by sorbitol. We show that hyperosmotic stress induces the spatial reorganisation of HPRT into puncta that juxtapose and co-localize with WDR62 granules in a stress-dependent manner (Fig. 6H). This was further validated by examining the endogenous WDR62-HPRT interaction using PLA, which also revealed a stress-induced increase upon sorbitol treatment (Fig. 6I).

As to whether mutant WDR62 was recruited to purinosomes, as detailed in our response to reviewer #1 above (minor comment #6), we find that R438H mutant formed condensed granules prior to stress treatment while 3936dupC mutant did not form granules in response to stress. Therefore, MCPH mutations appear to disrupt the stress-induced formation of WDR62 granules in the cytoplasm. Since we also find that WDR62 KO did not prevent stress-induced formation of PFAS and PPAT granules, which may represent or overlap with purinosomes, we chose to not include our findings on granule localization of mutant WDR62 localization in our current revised manuscript. We instead focused on rescue of HPRT and BAG2 levels with patient-derived MCPH mutant variants of WDR62. We confirmed that, unlike WT WDR62, mutant WDR62 could not fully return HPRT or BAG2 levels in WDR62 KO cells (Fig. 6B).

In Fig7C, HPRT/tubulin ratio appears to decrease in WT from 0hr to 24h, but the graph does not show this decrease. Additionally, quantification of PFAS(FGAMS) and BAG2/tubulin should be performed.

Response: While slight variations in HPRT signals are visible from 0 h to 24 h in the representative blot, quantification across the $n = 9$ biological replicates do not support a

significant decrease, with these variations falling within the SEM shown in the graph. This representative blot was selected for its clarity and since it most clearly depicts the key trend which is the increasing difference in the HPRT/Tubulin ratio between WT and KO cells with increased duration of CHX treatment. Additionally, in response to this comment, we have now quantified PFAS and BAG2/Tubulin and have inserted these data into Fig. 6C.

Fig7D (now Fig. 6D) is problematic. HPRT in WDR62-KO cells seems to localize in the nucleus, possibly due to stronger exposure in KO conditions compared to WT. Also, the line scan is drawn in areas with low signal in WT. The comparison should be performed in areas with high perinuclear signal.

Response: We appreciate the reviewer's feedback and acknowledge their concern of an apparent differences in fluorescence intensity in WDR62 KO vs WT cells. In the original submission, slight differences in fluorescence intensity between the WT and WDR62 KO panels may have exaggerated differences in HPRT levels in the nucleus. To address this, we have replaced the representative images with those with more consistent fluorescence intensity across conditions and better represent the average population of sampled cells. Regardless, quantified the change in HPRT cytoplasmic redistribution in response to WDR62 loss across multiple independent biological replicates (n=4) and multiple cells (>12 cells per repeat) within each biological replicate to confirm a change in HPRT distribution in KO cells (Fig. 6E+G).

The localization of HPRT should be compared in WT and WDR62-KO with BAG2 siRNA. It is also possible to confirm whether the phenotypes observed in KO, such as cell proliferation and xanthosine/inosine levels, are rescued.

Response: We conducted a series of immunofluorescence experiments to assess the impact of BAG2 knockdown (siRNA) on the spatial distribution of HPRT in WT and WDR62 KO cells. BAG2 depletion had no effect on HPRT distribution and did not rescue its aggregated-like appearance in WDR62 KO cells (Fig. S10C). Thus, while abnormal HPRT localization in absence of WDR62 was due to excessive of HSP70/90 activity (Fig. 6F), this was not reversed by BAG2 siRNA. However, BAG2 siRNA reduced BAG2 levels to below wild-type cells (Fig. 6I). An imbalance of HSP and co-chaperone levels are known to be involved in aggregation of cytoplasmic proteins. (doi.org/10.1096/fj.202002645R). Therefore, while BAG2 siRNA may have returned HPRT levels, it may not have appropriately corrected the levels of HSP70/90 activity required for normal HPRT localization (lines 407-413 in revision).

We did not attempt to rescue cell proliferation and xanthosine/inosine levels with BAG2 siRNA in order to prioritize other studies requested by reviewers such as neurodevelopment function of HPRT and flux analysis of purine synthesis/salvage.

It should be considered that the induction of Bag2 in WDR62-KO might allow purinosome formation to proceed normally due to compensation. The co-localization of WDR62 and purine enzymes during purinosome formation should be compared when BAG2 expression is suppressed. Similarly, any changes in BAG2 localization in WDR62-KO should be examined. Furthermore, the purinosome formation ability should be compared in WDR62KO + Bag12 siRNA condition.

Response: To address these insightful comments and requests by reviewer #2 response, we have performed additional experiments to assess whether BAG2 facilitates WDR62 granule assembly, purinosome assembly, and the WDR62-HPRT interaction. siRNA-mediated BAG2 depletion did not prevent stress-induced assembly of WDR62 or PFAS granules (Fig. S6D+E). Thus, unlike HSP70/90 activity, purinosome assembly and WDR62 localization to purinosomes did not appear to require BAG2. Rather we demonstrated a role for WDR62-BAG2 in regulating HPRT (Fig. 6, lines 400-411).

The reduction of HPRT in WDR62-KO should be examined for potential effects of enhanced degradation via the ubiquitin-proteasome system or the autophagy-lysosome system.

Response: See our response to reviewer #1, minor comment #3. Briefly, neither MG132 blockade of proteosomal degradation nor chloroquine inhibition of autophagy was sufficient to return HPRT levels in WDR62 KO cells. However, these studies are not exhaustive and we are currently pursuing alternative and more specific inhibitors of UPS or lysosomal degradation. As this is not essential for the main findings of the current manuscript, we will include delineation of HPRT degradation pathway in a future publication.

Although it is known that HPRT-KO mice do not exhibit any effects on normal brain development except in some dopaminergic neurons, what are your thoughts on this?

Response: We thank the author for raising this interesting point. While global HPRT KO mice appear not to exhibit widespread brain development defects (doi: 10.1007/s00018-022-04326-x) this does not preclude a role for impaired HPRT to contribute to specific neurodevelopmental defects in context of WDR62 mutation or loss. In utero electroporation studies, we find that WDR62 or HPRT depletion results in precocious delamination of apical precursors which may trigger premature differentiation. However, while WDR62 depletion led to reduced proliferation of delaminated radial glia ventricular/subventricular zone, we observed increased proliferation with HPRT loss (Fig. 7). Our findings are in good concordance with the study mentioned by reviewer #2, Witteveen et al. 2022 (doi: 10.1007/s00018-022-04326-x), who similarly reported an increase in proliferation and abnormal cell migration patterns which may be attributed to apical delamination of radial glia. The increased proliferation of progenitors in the intermediate zone or outer ventricular/subventricular zone may compensate for premature differentiation of apical progenitors to explain the lack of overall reduction in brain size with HPRT deficiency alone. Thus, our findings indicate that defects in WDR62-HPRT may contribute to the premature apical delamination of radial glia but WDR62 has additional functions that are indispensable for normal brain development. This may include complex functions in regulating purine metabolism independent of HPRT. We have now included the paper by Witteveen et al. 2022 in our revised manuscript and the above was discussed in detail at lines 565-577.

Minor comments:

- Please write the full name before the abbreviation of the gene.
- There is no measurement data for Fig7C, and a measurement line is drawn only in one panel of the ROI.
- The line 488 "Fig11" looks like a typo.

Full Revision

Response: As requested by the reviewer, we have included the full name of genes before their abbreviation and corrected the typographical error (line 548 in revision). For Fig S7C (Fig. S6B in revision), we have removed the measurement line which was included in error in our original manuscript. This supplementary figure demonstrates that the stress-stimulated granule assembly of ectopically expressed PFAS and PPAT was not altered or appreciably different in WDR62 KO cells. We quantified this for sorbitol treatment (Fig S6A). We performed the purine-depletion experiment twice with identical results. Given this was a negative result we focused our efforts elsewhere.

• The table could not be found.

Response: We apologise for this oversight. The Supplementary Information file containing Tables S1-3 was excluded from the original submission has now been included in our revised submission.

• It is strange that all measurement values for WT or control in Fig2, Fig7, and FigS9 are exactly 1.0 without any variation. Please check the measurement method again.

Response: Our densitometric band measurements in western blots within the indicated figures are normalized against WT control cells as a reference condition. This removes variation in arbitrary densitometric values that changes from blot to blot even for identical samples. Thus, values are fold-change in protein levels relative to WT control conditions. Hence values for WT or control cells are 1 (no change relative to itself) as the reference points and there is no variation between replicate experiments. We apologize for not explaining this in our original submission. Our revision now describes this quantification and processing of raw data in methods and materials (lines 668-671).

• Please write the method for purine depleted medium.

Response: Our revised manuscript includes description of how we depleted cells of purines in the Materials & Methods at lines 636-640 with reference to source materials and prior studies.

****Referees cross-commenting****

I concur with the accurate point observations by the other reviewers. The authors should address the most of the comments provided, as many of the suggested experiments are feasible. If the paper aims to elucidate the one of the causes of microcephaly, specifically, the issues related to cell type and endogenous proteins experiments need to be resolved, and addressing these issues would substantially enhance its quality and impact.

Reviewer #2 (Significance (Required)):

SIGNIFICANCE

=====

Most of the roles of purinosomes in the central nervous system remain unknown. The discovery

that the WDR62/MCPH2 gene, responsible for microcephaly, is related to purinosomes will have a major impact on this field. Additionally, the ability to easily induce purinosomes through sorbitol phase separation is a significant technical advance in terms of cost and simplicity. Furthermore, many genes related to microcephaly, such as MCPH, are factors directly involved in cell division by regulating the mitotic spindle and centrosomes. This study has revealed a new role for WDR62, uncovering part of a novel molecular mechanism for microcephaly.

Reviewer #3 (Evidence, reproducibility and clarity (Required)):

Summary:

In the present work, authors describe a novel role of the microcephaly associated protein WDR62 in purine metabolism under cell stress conditions. In the proposed cellular model (AD293 WDR62 overexpression system), the WDR62 proximity binding partners are firstly identified and categorized according to their functional role in the cell (protein folding, purine metabolism, and stress granules). Among them, authors focus on BAG2 - a HSP70/90 co-chaperone involved in cellular stress responses. After the characterization of the WDR62-BAG2 physical interaction sites, suggested to be disrupted by WDR62 pathogenic mutations, their functional interaction in cellular stress responses is investigated. WDR62-associated granules are extensively characterized for their physical and dynamic properties under different conditions (i.e., hyper-osmotic stress). Further, through the evaluation of N- and C-terminally truncated form of WDR62 authors characterize the protein regions responsible for WDR62-containing granule condensation - suggesting a potential mechanism disruption in the event of pathological WDR62 mutations. Lastly, authors provide evidence that WDR62 condensation does not occur in canonical stress granules but in the so called-purinosomes, where it participates in the regulation of purine metabolic pathway stabilizing HPRT (purine salvage enzyme) via WDR62-BAG2-HSP70/90 axis.

Major comments:

Overexpression system and the employed cell line are a major limitation of the study. There is no experimental data on human neural cells and on endogenous WDR62, underestimating the potential difference in cell type-specific metabolism. In light of this consideration, the provided introduction and conclusions on neural development and microcephaly have to be reformulated. I suggest providing a more general introduction/conclusions on WDR62 role (and alterations) in cell division and cell metabolism (neurodevelopment and cancer share common patterns) since purine homeostasis is not exclusive of neural progenitor cells.

This reviewer thinks that the structure of the work is a bit convoluted (too many results in main figures that are not substantial). I suggest to re-organize and to prioritize the most relevant results. Further, it would be clinically relevant to add WDR62 mutant constructs in the functional

evaluation of purine metabolism to better dissect the physiological role of WDR62 and the impact of the mutations on cellular physiology.

Response: We are appreciative for this constructive evaluation of our manuscript and frank comments on the limitations of our study from reviewer #3. We have now included extensive new studies that provide evidence supporting endogenous mechanisms and insights into in vivo functions in neurodevelopment. We have also removed and combined several figures relating to the stress-induced purinosome assembly of WDR62 to better focus our manuscript on WDR62 interaction mechanisms and their purine metabolic function.

Fig. 1: Overexpressed WDR62 fluorescence signal might be artifactual and may hide more detailed localization pattern during interphase. Authors should also provide endogenous WDR62 immunofluorescence panel in AD293 cells. Additionally, the "cytosolic" localization of WDR62 during interphase (indicated in the introduction, lines 88-89) has been re-defined in recent works pointing out that the protein is dynamically associated with the interphasic centrosome, the Golgi apparatus, and spindle poles during mitosis.

Response: In response to this point, we have added text in the introduction (line 100-102) to clarify the dynamic association of WDR62 in cytoplasmic compartment during interphase includes the golgi apparatus. We have also added reference to the study by Dell'Amico and co-workers (doi: 10.7554/eLife.81716, Ref #24 in revision) alluded to by reviewer #3.

We utilized ectopic expression of tagged WDR62 constructs to determine redistribution to stress-responsive cytoplasmic granules and co-localization with purine enzymes. Immunofluorescence staining of endogenous WDR62 also appears to reveal granule assembly but these findings are not as clear as the primary antibodies also detect additional proteins independent of WDR62 (validated using our KO cells). We agree that protein overexpression may result in artificial localization patterns but this can be mitigated with careful controls. We find that stress-induced WDR62 granule localization is highly dynamic and reversible. We observe the same response with full-length protein using different fluorescent protein or small affinity tags at either N- or C-terminus. High expression of mutant WDR62 (e.g. 3936dupC) or a closely related family member (MAPKBP1) do not form the same purinosome-associated granules. Moreover, in response to related comments by reviewer #1 and #2, we have now included proximity ligation assays confirm interactions between WDR62, BAG2 and purine enzymes (Fig. 3 and Fig. 6).

Fig. 1C lacks quantification of BAG2/CEP170/AURKA signal. Further, how can authors exclude that is not nocodazole effect on microtubules disruption which impairs WDR62 spindle pole localization and therefore protein-protein interactions? A panel showing that "low dose" nocodazole do not impinge endogenous and exogenous WDR62 localization in mitotic cells is needed.

Response: WDR62-BirA specific biotinylation and affinity isolation of BAG2, CEP170 and AURKA, compared to BirA or WDR62-HA only controls, was very clear in Fig. 1C. We did not quantify the extent that mitotic synchronization increased or decreased binding to WDR62 as the mitosis specific context was not a focus in our subsequent figures. Rather we focused on and quantified in detail WDR62-BAG2-HPRT mechanisms in response to cell stress.

We are also very confident that low dose nocodazole treatment does not prevent spindle pole localization. This treatment impinges on microtubule dynamics to trigger spindle checkpoints, arresting cells in mitosis. The bipolar organization of spindles is lost but spindle microtubules and minus-end microtubule directed localization of WDR62 at spindle asters are retained under these conditions and is specific to mitotic cells. The robust WDR62-BirA biotinylation of AURKA, which is spindle pole-associated, specifically in mitotic arrested cells further confirms WDR62 is retained at the spindle. We demonstrated this in our previous papers (Ref. 5+6). Others have also shown that both endogenous (doi: 10.7554/elife.81716) and exogenous WDR62 (doi: 10.1083/jcb.202007167, doi: 10.1242/jcs.157537) retain spindle pole localisation under similar conditions.

Fig. 3 H-J: The fluorescence signals are saturated (also evident in the intensity profile plot) and thus not applicable for any analysis. Further, how these linear ROIs are chosen? The signal pattern is not homogeneously distributed in those images. Please provide a more consistent fluorescence analysis.

Response: We acknowledge reviewer #3 concerns but while some granules, particularly those expressing G3BP-EGFP, exhibit saturated fluorescence signals, this does not impact or prevent our analysis. Our aim was not to quantify subtle fluorescence intensity changes within individual granules, but rather to compare fluorescence signal between granules across different channels to identify overlap. The linear ROIs were selected at random to illustrate that WDR62 and G3BP signals do not overlap between WDR62 and G3BP-positive granules.

Minor comments:

Abstract, line 49: How can these WDR62 mutations can result in a complete loss of the protein ("In cells lacking WDR62") if authors report co-IP experiments (Fig. 2) with clear mutant WDR62 bands? Rephrase accordingly.

Response: The statement in our original abstract referenced by reviewer #3 referred to results presented in Fig 7 (now Fig. 6 in our revision) comparing WDR62 KO with WT cells and not co-IP experiments with mutant WDR62 in Fig 2. We have revised our abstract substantially to incorporate additional experimental work and to ensure clarity in our statements related to KO cells lacking WDR62 and cells expressing WDR62 mutants.

Result referred to Fig. 2D reports that "BAG2 co-immunoprecipitated with WDR62(N)-EGFP but not WDR62(C)-EGFP". The blot and the relative quantification in figure 2D instead show BAG2 signal in the WDR62(C)-EGFP - even if significantly lower. Please rephrase accordingly.

Response: We have revised line 192 of the main text to more accurately state the reduced interaction between WDR62(C)-EGFP and BAG2.

Lines 186-187: authors declare that the C-terminal tail comprising the helix-loop-helix domain is required for BAG2 to bind full-length WDR62. There are no such data in support of this. The C-terminal fragment includes both the disordered region and the dimerization domain. How can authors conclude that the dimerization domain alone is sufficient to bind BAG2?

Response: In Fig. 2, we show that the co-IP of BAG2 was significantly impaired in cells expressing WDR62(3936dupC), which lacks the C-terminal helix-loop-helix (HLH) domain. Additionally, we demonstrate that the C-terminal half of WDR62, which includes the HLH domain, does not bind BAG2. Based on these findings, we conclude that while the HLH domain is necessary for BAG2 binding to full-length WDR62, it is alone not sufficient. To ensure clarity, we have revised the main text (lines 207-209) to state "...the C-terminal helix-loop-helix domain—required for WDR62 dimerisation—is necessary but not sufficient for BAG2 to bind full-length WDR62."

Lines 189-190: results in AD293 cell line are not directly applicable in demonstrating that poor WDR62-BAG2 interaction can lead to alterations in brain development. Please rephrase.

Response: We established that WDR62 interacts with BAG2 co-chaperone and MCPH mutations in WDR62 disrupt this interaction. Although our results were performed in AD293 cells, it seemed reasonable to speculate that WDR62 interactions with chaperones might contribute to brain development given well established WDR62 functions in this context. However, we acknowledge that this speculation may not be appropriate at this point of the manuscript, so we have removed this text (line 210) in our revised manuscript.

Line 196: Indicate here, as the first mention, stress granules as "SGs" and use the abbreviation consistently throughout the manuscript.

Response: We have abbreviated stress granules as suggested (first mentioned at line 102) and utilized this abbreviation consistently throughout the manuscript.

Line 255: are human neural progenitor cells enough sensitive to sorbitol? If not, the proposed experimental design is a bit artificial and the results/conclusions cannot be related to neural development alterations. I suggest applying more "physiological" stressors and frame the results in meaningful neurodevelopmental/tumorigenic environment. Please add this point to the discussion.

Response: Neural progenitors are likely sensitive to sorbitol, as hyperosmotic stress has been used to induce phase separation of a wide variety of proteins in neural contexts (doi: doi.org/10.1038/s41598-023-39090-w, doi.org/10.1016/j.celrep.2018.06.094). In this study, we leveraged sorbitol-induced hyperosmotic stress as a controlled and reproducible means of triggering WDR62 phase separation, enabling us to examine its downstream interactions with BAG2, HPRT, and other purine enzymes. We further extend these observations to metabolic cell stress with purine-depletion.

We found that WDR62 phase separation occurs rapidly at low sorbitol concentrations (~50 mM) (Fig. 3B), suggesting that even milder osmotic stress, particularly under prolonged exposure, could similarly drive WDR62 condensation in physiological settings. As requested by the reviewer, we have added a small section to the discussion (lines 472-480) to discuss the physiological implications of sorbitol stress on WDR62 granule assembly.

Line 240: WDR62 granules association with microtubules and especially mitochondria is not

convincing (Fig. S5). This data seems to be a bit qualitative, please provide more detailed quantification of this parameter.

Response: The association of WDR62 granules with microtubules and mitochondria is quantitatively assessed using two methods, as shown in the graphs to the right of the images. One graph presents the proportion of WDR62 granules overlapping with CytC/Tubulin, providing a binary measure of colocalization. We also examined the degree of signal correlation across the entire ROI by calculating Pearson's correlation coefficient. In response to sorbitol, we showed a higher association of WDR62 with Tubulin and CytC compared to randomised controls. We have updated the Materials and Methods to include a detailed description of this analysis (lines 708-720).

Fig. 4 is convoluted. I suggest moving some data to supplementary to improve the clarity of the figure.

Response: In addressing this comment and related comments from other reviewers to focus our manuscript, we have removed our data on fluorescence recovery and post-stress disassembly of WDR62 granules from what was Fig. 4 in our original submission and combined remaining components with Fig. 3 to centre on stress-induced assembly of WDR62 granules for our revised manuscript.

Line 273: "Liquid-like protein condensates also exchange their contents with the bulk cytosol [52]". Reference 52 reviews the existing literature referred to biomolecular condensates that exert nuclear function (e.g., genome organization, gene expression, and DNA repair). No mention on events involving cytoplasm. Please add a more relevant reference.

Response: We thank the reviewer for highlighting this inconsistency. However, this reference is no longer required and has been removed from our revised manuscript as the section of the main text has been deleted in alignment with the above response where figure panels relating to WDR62 phase separation were removed for focus and clarity.

Lines 290-291: have authors considered the effect of sorbitol on microtubules dynamic that might reflect in granules dynamic changes?

Response: We thank the authors for this insightful comment. Hyperosmotic stressors such as sorbitol are known to reduce microtubule dynamicity (doi.org/10.1016/j.devcel.2022.02.001), likely due to increased cytoplasmic viscosity and crowding effects. While we have not directly assessed microtubule dynamics in our study, it is certainly possible that these changes could influence WDR62 granule dynamics, given their association with microtubules (Fig. S6). While we have reduced emphasis on the dynamic nature of WDR62 granules in our revision, a useful direction for future studies to explore how alterations in microtubule dynamics induced by physiological stressors facilitate changes in WDR62 granule assembly or dynamics (e.g., fission, fusion).

Line 295: I suggest moving the prediction of the disordered region of WDR62 when first mentioned (e.g., Supplement referred to Fig. 2)

Response: This text is no longer required as we have removed this dataset from our revised manuscript to address reviewer consensus feedback to enhance cohesiveness and clarity.

Fig. S6C-E, I: Unclear which is the criterion by which a cell is marked as "with" or "without" granules.

Response: This text is no longer required as we have removed this dataset from our revised manuscript to address reviewer consensus feedback to enhance cohesiveness and clarity.

Fig. S8: Unclear, also from the micrograph showed in the figure, how authors have counted/considered the microtubules/mitochondria associated purinosomes. Seems very qualitative and observer dependent. Please provide a more reliable analysis.

Response: We apologise for omitting a description of the methodology used in the analysis of the images in Fig. S8 (now Fig. S6 in revision). We have now provided a detailed description in the Materials and Methods section (lines 709-721) of how microtubule- and mitochondria-associated purinosomes were identified and quantified.

Fig. 6A: The same blot of WDR62 KO is shown in Fig. S7. Please remove one.

Response: As requested, we have removed a set of blots demonstrating WDR62 protein deletion in KO cells from Fig. S7 (Fig. S6 in revision).

Fig. 6C, D: Method for cell proliferation measure is indirect and "rounded cells" as indicator of cell death is sub-optimal. Analysis with specific markers would be preferable in both cases.

Response: We used an XTT assay to measure cell viability as a function of cell number. In revised text, and also detailed in our response to reviewer #1 (point 4 under Text and Figures), we clarified that this was a measure of cell viability in response to purine-depletion as oppose to a direct measure of cell proliferation. Our amended text attributes the results in Fig 6C (now Fig. 5B in revision) to changes in cell viability rather than proliferation.

With regards to additional measure of cell death, we had also performed LDH release assays to quantify cell death in addition to our measurement of cell rounding. The LDH assay is widely used and accepted measure of cell death or cytotoxicity and is indicated in Figure 5D in the revision.

Fig.7B: Why the transfection control vector "EGFP only" significantly increases/decreases the BAG2/HPRT expression with respect to the negative control?

Response: The reviewer comment here on Fig. 7B (now Fig. 6B in revision) refers to the control vector (EGFP only) transfected into WDR62 KO cells, as opposed to WT cells. Therefore, the difference in protein expression in this condition does not match the WT cells in the first lane as BAG2 and HPRT are increased and decreased respectively in KO cells compared to WT. This aligns with results presented in Fig. 6A.

Paragraph from line 410 to 434: very confusing, the reported results are not well conveyed and therefore not convincing. To be reformulated.

Full Revision

Response: We thank the reviewer for the direct and constructive feedback. The revised section (lines 378–416) addresses whether WDR62-BAG2 regulates HPRT levels. It has been substantially updated to include new experimental data and to reflect our latest findings and conclusions. We believe these revisions have significantly improved the logical flow and clarity of the discussion.

Lines 524-526: the author's conclusion that "...the loss of purine metabolic enzymes, including HPRT, disrupts neurogenesis, resulting in microcephaly, cell cycle defects, ciliopathies, and abnormalities in proliferation and neural progenitor fate decisions, mirroring the loss of WDR62." is not supported by the cited literature [29] and by the results presented in this work. Please provide additional references or remove the statement.

Response: As requested by the reviewer, we have removed the statement and substantially revised this section of the discussion (lines 563-677) to incorporate findings from our additional studies such as *in utero* electroporation.

Lines 527-529: if authors state that "...other WD repeat-containing and microcephaly-associated proteins interact with purine enzymes..." have to provide additional references in addition to the NWD1 one. Otherwise, these lines should be rephrased as "another WD repeat containing and microcephaly-associated protein...".

Response: We have amended this statement (line 589-592 in revision) as requested.

Reference 62 is not well indexed in the Reference section. Please adjust.

Response: We thank the reviewer for pointing out this error. The reference to Rauch et al. (2014) [Ref. 60 in the revised manuscript] has been corrected and now includes the complete bibliographic details.

****Referees cross-commenting****

This reviewer thinks that the points raised by reviewer #1 and #2 are very accurate and significant. Some of them are also shared between our three review reports and in general are referred to: clarity of the manuscript improvement, little consistency between the results displayed in the figures and the text/conclusions in some points, concerns about the reliability of some measurements/result and the employed cellular model, and the lack of endogenous protein data.

Reviewer #3 (Significance (Required)):

General assessment:

The here described new role of WDR62 in purine metabolism and the proposed pathway are novel and relevant to shed light on pathophysiological cellular and molecular mechanisms that potentially underlie neurodevelopmental defects and carcinogenesis - processes in which WDR62 is implicated. The experimental design is extended and generally well-conceived even though quite dispersive in some points.

Full Revision

The strength of the work resides in its versatility - making these findings potentially applicable to different cell types and different contexts (e.g., from neural development to malignancies) - and in the protein-protein interactions characterization under several conditions.

Similarly, the major weakness is the generalist trait of the findings that describes WDR62 cellular behavior mostly in an over-expression system in an immortalized cell line, underestimating the intrinsic metabolic and protein expression-level differences among cell types.

Advance:

WDR62 is a scaffold protein with pleiotropic functions and a plethora of molecular interactors. Literature reports many molecular pathways involving WDR62 mainly in cell cycle progression, primary cilia biogenesis and centrosomal functions in a neurodevelopmental context. In the present work, authors describe mechanistic insights of a never reported WDR62-BAG2-HSP70/90 molecular pathway shedding new light on the role of this protein in cellular metabolism thus providing a new perspective on WDR62 pathophysiological functions.

Audience:

Basic research audience will be interested in this research work. The described molecular pathway involving WDR62 in purine metabolism might be relevant to other research on how WDR62 cellular and molecular dynamics are impactful on neural development and malignancies.

Expertise:

Human neural development and alterations, iPSCs, neural stem cells, CRISPR-Cas9

Attendee panel closed

Dear Dr. Ng,

Thank you again for submitting your revised manuscript EMBOJ-2025-121757-T, which had initially been reviewed at Review Commons, for consideration by The EMBO Journal, and for your patience during peer review. Your revised manuscript has now been seen by the three experts who had previously assessed the first version of your study at Review Commons, and we have received their detailed and constructive reports, which you can find below.

I am pleased to say that, as you will see, all three referees find the revised manuscript substantially improved and strengthened by the addition of new data and reorganization of the text, which has enhanced its clarity. On the other hand, all three referees raise several remaining technical issues -many of which are related to fluorescence intensity, used markers, consistency of quantification, and clarity- that must be addressed before we can publish the manuscript at The EMBO Journal.

Given the referees' input, I would like to invite you to submit another revised version of your manuscript fully addressing all remaining technical concerns of the referees. Please submit along with your revised manuscript a detailed point-by-point response addressing all referees' comments. Please note that we can allow only a single additional round of experimental revision, and acceptance of your manuscript will therefore depend on the completeness of your responses in this revised version. Please let me know if you have any questions or comments that you would like to discuss with me. If there are any major points you do not agree with or cannot address during your revision, I would encourage you to share them with me as early as possible to discuss how to proceed further in the most efficient way.

We generally allow three months as standard revision time (November 3, 2025). As a matter of policy, competing manuscripts published during this period will not negatively impact our assessment of the conceptual advance presented by your study. However, we request that you contact us as soon as possible upon publication of any related work, to discuss how to proceed. Should you foresee a problem in meeting this three-month deadline, please let us know in advance and we will be able to grant an extension.

Thank you for the opportunity to consider your work for publication in The EMBO Journal. I look forward to your revision.

Best regards,

Ioannis

Instructions for preparing your revised manuscript

1. When you are ready to submit the revision, please upload:

- A Word file of the manuscript text (including legends of main Figures, EV Figures and Tables). Please make sure that changes are highlighted (or "tracked") to be clearly visible.

- Individual production-quality figure files (one file per figure). When assembling your figures, please refer to our figure preparation guidelines in order to ensure proper formatting and readability in print as well as on screen:

If the data shown in a figure are obtained from n {less than or equal to} 2, please use scatter plots showing the individual data points.

- i. the name of the statistical test used to generate error bars and P values
- ii. the number (n) of independent experiments (please specify technical or biological replicates) underlying each data point (discussion of statistical methodology can be reported in the Materials and Methods section, but figure legends should contain a basic description of n , P , and the test applied)
- iii. the nature of the bars and error bars (s.d., s.e.m.).

- A point-by-point response to the referees' comments, with a detailed description of the changes made (as a word file). All referees' concerns must be fully addressed and their suggestions taken on board. When preparing your letter of response to the referees' comments, please bear in mind that this will form part of the Review Process File and will therefore be available online to the community. Please note that you have the possibility to opt out of the transparent process at any stage prior to publication by letting the editorial office know (contact@embojournal.org); if you do opt out, the Review Process File link will point to the following statement: "No Review Process File is available with this article, as the authors have chosen not to make the review process public in this case.". For more details on our Transparent Editorial Process, please visit our website: <https://www.embopress.org/page/journal/14602075/authorguide#transparentprocess>

- Expanded View (EV) files (replacing Supplementary Information) that are collapsible/expandable online. A maximum of 5 EV Figures can be typeset. EV Figures should be cited as "Figure EV1, Figure EV2" etc. in the text, and their respective legends should be included in the manuscript file after the legends of regular figures. See detailed instructions regarding Expanded View files here: <https://www.embopress.org/page/journal/14602075/authorguide#expandedview>

- For the figures that you do NOT wish to display as Expanded View figures, they should be bundled together with their legends in a single PDF file called "Appendix", which should start with a short Table of Contents (including page numbers). Appendix figures should be referred to in the main text as: "Appendix Figure S1, Appendix Figure S2" etc. Please see detailed instructions here: <https://www.embopress.org/page/journal/14602075/authorguide#expandedview>

- A complete author checklist, which you can download from our author guidelines (<https://www.embopress.org/page/journal/14602075/authorguide>). Please note that the checklist will also be part of the Review Process File.

2. Please note that no statistics should be calculated and shown in Figures if $n=2$. Please also note that each p value should be reported as an exact value.

3. Before submitting your revision, primary datasets (and computer code, where appropriate) produced in this study need to be deposited in appropriate public databases (see <https://www.embopress.org/page/journal/14602075/authorguide#dataavailability>). In particular, we kindly request you to submit all mass spectrometry data produced in your study to an appropriate repository. The accession numbers, database, and the specific URLs (links) should be listed in a formal "Data availability" section (placed after Methods), following the example below:

"The RNA-seq datasets produced in this study are available in the following database:
Gene Expression Omnibus GSE46843 (<https://www.ncbi.nlm.nih.gov/geo/query/acc.cgi?acc=GSE46843>)"

*** All links should resolve to a page where the data can be accessed. ***

*** Please remember to provide in the Data availability section of your revised manuscript reviewer passwords if the datasets are not yet public. ***

*** The Data Availability Section is restricted to new primary data that are part of this study. In case you have no data that require deposition in a public database, please state so instead of referring to the database: "Our study includes no data deposited in public repositories." under the heading "Data availability". ***

4. The materials and methods need to be described in the manuscript using our structured methods format, which is now required for all research articles. According to this format, the Methods section includes a single "Reagents and Tools Table" - listing key reagents, experimental models, software and relevant equipment including their sources and relevant identifiers - followed by a "Methods and Protocols" section describing the methods. Please download and fill our Reagents and Tools Table template (.docx), which you can find in our author guide:

<https://www.embopress.org/page/journal/14602075/authorguide#structuredmethods>. When submitting your revised manuscript, please do not include the Reagents and Tools Table in the Methods section of the manuscript but instead upload it as a separate file choosing the file type "Reagent Table".

5. Please check that the title and the abstract of the manuscript are brief, yet explicit, even to non-specialists. The length of the title should not exceed 100 characters, and the abstract should be a single paragraph not exceeding 175 words.

6. Please also note our reference format: <https://www.embopress.org/page/journal/14602075/authorguide#referencesformat>.

8. Please remember: digital image enhancement is acceptable practice, as long as it accurately represents the original data and conforms to community standards. If a figure has been subjected to significant electronic manipulation, this must be noted in the

figure legend or in the "Materials and Methods" section. The editors reserve the right to request original versions of figures and the original images that were used to assemble the figure.

9. Our journal encourages inclusion of data citations in the reference list to directly cite datasets that were obtained from public databases. Data citations in the article text are distinct from normal bibliographical citations and should directly link to the database records from which the data can be accessed. In the main text, data citations are formatted as follows: "Data ref: Smith et al, 2001" or "Data ref: NCBI Sequence Read Archive PRJNA342805, 2017". In the Reference list, data citations must be labeled with "[DATASET]". A data reference must provide the database name, accession number/identifiers, and a resolvable link to the landing page from which the data can be accessed at the end of the reference. Further instructions are available at: <https://www.embopress.org/page/journal/14602075/authorguide#referencesformat>.

10. We request authors to consider both actual and perceived competing interests. Please review our policy (<https://www.embopress.org/page/journal/14602075/authorguide#conflictsofinterest>) and update your competing interests statement if necessary. Please name this section 'Disclosure and competing interests statement' and place it after the Acknowledgements section.

11. Please note that all corresponding authors are required to provide an ORCID ID upon submission of a revised manuscript (<https://orcid.org/>). Please find instructions on how to link your ORCID ID to your account in our manuscript tracking system in our Author guidelines (<https://www.embopress.org/page/journal/14602075/authorguide#authorshipguidelines>).

12. We use CRediT to specify the contributions of each author in the journal submission system. CRediT replaces the author contribution section, which should be removed from the manuscript. Please use the free text box to provide more detailed descriptions. See also guide to authors: <https://www.embopress.org/page/journal/14602075/authorguide#authorshipguidelines>.

14. We would also welcome the submission of cover suggestions or motifs to be used by our Graphics Illustrator in designing a cover.

15. Please use the link below to submit your revision:

Referee #1:

Compared to the previous version, substantial revisions and data additions have been made. I agree with most of the authors' responses; however, further clarification and revision are still necessary regarding the *in vivo* experiments.

Major concerns

- Figure 7C: In the HPRT si group, the thickness of the VZ/SVZ appears to be greatly reduced, while the CP is abnormally thick. This suggests that the images may not represent the same cortical region as in the control. The authors should replace the images and quantify using correctly matched sections.

- The delamination phenotype observed in Figure 7B is not present in the HPRT si group of Figure 7C or the Wdr62, HPRT si group in Figure 7D, raising concerns about data consistency.

- Figure 7D: Tbr2 is typically used as a marker for intermediate progenitor cells in the SVZ; however, in the control group, the Tbr2 signal appears to extend into the apical surface of VZ, suggesting high background staining.

- When quantifying neural progenitor cells after *in utero* electroporation, it is standard to stain for both Pax6 and Tbr2. Pax6⁺Tbr2⁻ radial glia and Pax6⁻Tbr2⁺ intermediate progenitors should be separately quantified in the VZ and SVZ. With only the number of Tbr2-positive intermediate progenitors presented, it is difficult to interpret whether the increase reflects expansion of the progenitor pool or premature differentiation from radial glia.

- Investigating the endogenous expression pattern of WDR62 with cell type markers during cortical development by IHC would further support the interpretation of the observed *in vivo* phenotypes.

- Similarly, Ki67 and pH3 should also be quantified separately in the VZ and SVZ.

Referee #2:

In the revised version of the manuscript, the authors have substantially addressed many of the concerns previously raised by this reviewer. One of the main issues—the overly general nature of the study—has been notably improved by the inclusion of the in utero electroporation knockdown experiment and the subsequent evaluation of its effects on delaminating neural progenitors. These additions significantly strengthen the case for the role of WDR62 in purinosomes and their connection to brain development. Furthermore, the proximity ligation assay provides additional support for the proposed WDR62-BAG2-HPRT axis. The reorganization of the manuscript has also enhanced its clarity and improved the overall delivery of the main message. Taken together, this reviewer now considers the manuscript suitable for publication. The only remaining concern pertains to the following:

- Previous concern: "Fig. 3 H-J: The fluorescence signals are saturated (also evident in the intensity profile plot) and thus not applicable for any analysis. Further, how these linear ROIs are chosen? The signal pattern is not homogeneously distributed in those images. Please provide a more consistent fluorescence analysis."
- Response from the authors: We acknowledge reviewer #3 concerns but while some granules, particularly those expressing G3BP-EGFP, exhibit saturated fluorescence signals, this does not impact or prevent our analysis. Our aim was not to quantify subtle fluorescence intensity changes within individual granules, but rather to compare fluorescence signal between granules across different channels to identify overlap. The linear ROIs were selected at random to illustrate that WDR62 and G3BP signals do not overlap between WDR62 and G3BP-positive granules.
- Although the authors did not aim to quantify subtle changes in fluorescence intensity, overexposed and oversaturated pixels can introduce artifacts, compromising the quality of both the analysis and the overall work. As a result, accurate signal quantification is not feasible, and it becomes difficult to distinguish true signal from potential precipitates or nonspecific dots. I recommend selecting representative and reliable fields for analysis and providing justification for the image processing pipelines used.

Referee #3:

In the revised manuscript, the authors significantly improved its readability by fixing some of the inconsistencies between the main text and figures. Many of the bold claims that were not well substantiated were also addressed. While they softened several claims connecting WDR62 to the purinosome, they did not affect its overall significance and impact to the scientific community. I commend the authors on this revised manuscript. With that being said, I have a few points that remain.

Comments:

1. (Line 148, Minor) The authors state "Overall, we identified 42 interactors, with 8 of these matching entries in the BioGRID database." In Figure 1D, I still only see 5 overlapping proteins. The rebuttal letter states that there are only 5 overlapping as well. Please revise to properly clear up this inconsistency between the main text and Figure 1D.
2. (Lines 156-162, Figure 1I, Minor) The authors mention the network "identified members within... several HSP40 family members (e.g., BAG2, BAG5, DNAJA2, and DNAJC7)..." When I look at the network in Figure 1I, I only see BAG2. Why are the others mentioned but not in the figure? These proteins are in Figure 1D (bold faced). Therefore, this reviewer still is not sure why proteins identified in the BioID screen are also not present in the network image despite being referenced in the main text.
3. (Figure 1I, Major) In the rebuttal, the authors state "As the reviewer correctly points out, purine metabolic enzymes were not direct interactors of WDR62. Purine enzymes are linked to the molecular chaperones which, in turn, associated with WDR62 from the BioGRID database." There are several protein classes, many more notable than purine metabolic enzymes such as kinases, that are known to interact with HSP70 and their co-chaperones (HSP40s), but it is still not clear how the authors narrowed in on purine metabolism from data generated by these two independent sources at this point in the manuscript. I would have assumed a lot more nodes from the protein folding/chaperone cluster. It appears that there was some bias introduced into this network to link WDR62 with purine metabolism to substantiate their hypothesis. Given that purine enzymes are not mentioned until much later in the manuscript, maybe Figure 1I should just be only inclusive of the results generated from Figure 1D. The connection then can be made when introducing their hypothesis of a link between WDR62 and purine metabolism later. Otherwise, this is confusing to the reader as to why these are included in this figure.
4. (Methods, Minor) For the protein mass spectrometry, please provide the same level of detail of sample preparation, data acquisition and analysis as was provided for the metabolite tracing experiments. In addition, it would be helpful to include the number of technical replicates that were performed with each condition.

5. (Overall, Major) Given that the connection between WDR62 and purine metabolism is through HSP70/90 chaperones, I would highly recommend that the authors perform a co-IP (at least) to demonstrate that there are these associations between WDR62 and the chaperones. The question remains whether WDR62 is a client of these chaperones or not, and this would be helpful in proper interpretation and rationale for using HSP90 and HSP70 inhibitors. Is it possible that WDR62 only interacts with BAG2-HSP70 and not HSP90? HSP70 is known to have plenty of functions independent of HSP90. Further, HSP90 inhibitors are not the most specific. Therefore, the conclusions could be strengthened with co-IPs.

6. (Overall, Major) I am still not sure from the paper I understand how WDR62-BAG2 impacts HPRT. The authors lead you to think that it might be through changes in regulation of chaperones. So, is it that WDR62 interacts with HPRT, or is it that WDR62 alters chaperone activity/associations that can impact the stability of HPRT (i.e. acts as a co-chaperone)? If the latter is true, then I would recommend that HPRT be established as a client of HSP70/90 through co-IP assays +/- HSP70 and 90 inhibitors.

Rev_Com_number: RC-2024-02582

New_manu_number: EMBOJ-2025-121757-T

Corr_author: Ng

Title: The microcephaly protein WDR62 regulates purine metabolism through chaperone interactions.

*Please note that line number refer to lines on merged PDF of our manuscript and main figures. The line numbers on our original word document may differ. Our revisions on the word document are indicated by track changes.

Referee #1:

Compared to the previous version, substantial revisions and data additions have been made. I agree with most of the authors' responses; however, further clarification and revision are still necessary regarding the in vivo experiments.

Major concerns

- Figure 7C: In the HPRT si group, the thickness of the VZ/SVZ appears to be greatly reduced, while the CP is abnormally thick. This suggests that the images may not represent the same cortical region as in the control. The authors should replace the images and quantify using correctly matched sections.

Response: The reviewer has made a keen observation. However, we believe this is likely normal rostrocaudal/mediolateral variation in these developmental zones. Slight but inevitable variability in the location of the electroporated cell field necessitates sampling cortical sections and insets from slightly varying regions along the rostrocaudal/mediolateral axes. To provide reassurance of this, we have quantified the VZ/SVZ and CP thickness for all brain slices to allow direct comparison between groups and provided this in the supplementary material (Fig. S13A). Whilst there was a very slight increase in CP thickness in *Hprt* si group, this was not statistically significant and there were no differences between other groups. We modified the following sentence in our revised manuscript to address this concern.

We first assessed the spatial distribution of GFP-positive (GFP+) cells in the neocortex and noted that the overall thickness of ventricular and subventricular zones (VZ/SVZ) and cortical plate (CP) were not substantially different between groups (Fig. S13A). (Line 444 – 447)

In addition, we have also updated the representative image of brain sections electroporated with *Hprt* si in Figure 7C with brain layer thickness this is more representative of the average proportions that we found between conditions.

- The delamination phenotype observed in Figure 7B is not present in the HPRT si group of Figure 7C or the Wdr62, HPRT si group in Figure 7D, raising concerns about data consistency.

Response: We appreciate the reviewer's careful observation. We agree that the delamination in the representative image for the *Hprt* si condition in Figure 7C was not as striking compared to Figure 7B. However, the extent of delamination falls within the range of phenotypic variation we observed between animals in each treatment group. The differences in representative panels between Figures 7B and C

reflects this variability rather than an inconsistency in the phenotype. We have included an additional graph in supplementary material illustrating the distribution of GFP-positive cells, with error bars to depict variability between animals (Fig. S13B) and added the following sentence to revised manuscript.

While there was a degree of variability in the extent of movement of GFP+ cells from the ventricular surface between animals within the same siRNA group (Fig. S13B), our findings indicate with WDR62 siRNA, indicating precocious delamination and differentiation of progenitors from the ventricular surface following WDR62 or HPRT depletion compared to control animals (Fig. 7B). (Line 450 - 453)

We also replaced the representative image for the *Hprt* si group in Figure 7C to one that is more representative of the average quantified data and which also addresses the previous point regarding layer thickness.

Moreover, in addition to the delamination of apical progenitors, the precocious movement of GFP-labelled cells into the upper VZ/SVZ and IZ and away from the ventricular surface was observed consistently in Figure 7B, 7C and 7F in both *Wdr62* si and *Hprt* si animals, supporting finding of premature differentiation. Our revised manuscript text places greater emphasis on premature differentiation rather than delamination specifically.

Together, these findings suggest that while both WDR62 and HPRT regulate progenitor differentiation, their mechanisms in cortical development diverge such that HPRT loss favours the differentiation of apical progenitors into proliferative IPs, whereas WDR62 loss impairs the self-renewal of neural progenitors, likely via delamination and precocious differentiation into neurons. (Line 465 – 469)

• Figure 7D: *Tbr2* is typically used as a marker for intermediate progenitor cells in the SVZ; however, in the control group, the *Tbr2* signal appears to extend into the apical surface of VZ, suggesting high background staining.

Response: Reviewer #1 is correct in that *Tbr2* immunolabelling is often used to delineate intermediate progenitors in the SVZ, and the VZ is relatively negative for *Tbr2* during early cortical developmental stages (e.g. E12 or E13). However, at later stages, like the ones in these figures (E16), a pattern of overlap with *Tbr2* immunolabelling extending into the (e.g. *Pax6*^{+ve}) VZ is expected, likely due to the relatively high number of intermediate progenitors at this stage compared to earlier stages, the relatively smaller VZ, and the *en masse* terminal differentiation and delamination of progenitors at this concluding stage of neurogenesis. This has been shown by other independent groups through *Tbr2* immunohistochemistry (e.g. Guo et al. 2019 Cerebral Cortex DOI: 10.1093/cercor/bhz018, Kerimoglu et al. 2021. Science Advances DOI: 10.1126/sciadv.abc6792) and at the mRNA expression level (e.g. Allen brain atlas *eomes* expression shown below).

Reviewer Figure 1. Eomes/Tbr2 expression at E15.5 (Allen Brain Atlas)

Our extensive experience with Tbr2 expression and labelling across a broad range of ages and species (e.g. Paolino et al., 2023, Nat Comms. DOI: 10.1038/s41467-023-41652-5) makes us confident that the labelling in what was Figure 7D but now Figure 7F in the revised manuscript is bona fide and specific for Tbr2.

- When quantifying neural progenitor cells after in utero electroporation, it is standard to stain for both Pax6 and Tbr2. Pax6⁺Tbr2⁻ radial glia and Pax6⁻Tbr2⁺ intermediate progenitors should be separately quantified in the VZ and SVZ. With only the number of Tbr2-positive intermediate progenitors presented, it is difficult to interpret whether the increase reflects expansion of the progenitor pool or premature differentiation from radial glia.

Response: We thank the reviewer for emphasising the importance of distinguishing Pax6⁺/Tbr2⁻ apical radial glia and Pax6⁻/Tbr2⁺ intermediate progenitors in our electroporated cortices. However, previous studies have shown that electroporated cells from ventricular zone origin that are Tbr2 negative (Tbr2⁻) are basically Pax6⁺ apical radial glia and not other dividing progenitors with more committed fate (doi:10.1523/JNEUROSCI.2899-04.2005, doi:10.1038/s41586-021-03670-5). At E16.5, Tbr2⁺ cells label basal progenitors, intermediate progenitors and small number of differentiating neurons but not radial glia (doi:10.1523/JNEUROSCI.2899-04.2005). Moreover, progenitor cells in the SVZ at E16.5 express either Pax6 or Tbr2 but not both (doi:10.1523/JNEUROSCI.2899-04.2005).

Due the small brains limiting the number of antibody stains we were able to do with appropriately positioned electroporate cell fields, we made use of existing data to address the reviewer's excellent point. We re-analysed Tbr2⁺/GFP⁺ data to quantify Tbr2⁻/GFP⁺ cells within the progenitor zones as a proxy for Pax6⁺ radial glia (Fig. 7F). In HPRT-depleted cortices, there was a significant reduction in Tbr2⁻/GFP⁺ cells

in the VZ/SVZ (Fig. 7F). Combined with our previous finding of an increase in Tbr2+/GFP+ intermediate progenitors, this pattern strongly supports premature differentiation of apical radial glia rather than progenitor pool expansion. Our revised manuscript text incorporates these additional analyses, and we thank this reviewer as we think these additional analyses improve the specificity of our conclusions.

Furthermore, HPRT depletion significantly increased Tbr2+ cells in the IZ and reduced Tbr2- cells in VZ/SVZ with HPRT loss (Fig. 7F). Given previous reports that GFP-labelled cells that are Tbr2- in the VZ/SVZ are an accurate indicator of apical progenitors [52, 53], our findings indicate depletion of HPRT enhanced transition of apical to basal progenitors and differentiation into IPs without a loss in proliferative capacity. (Line 460-464).

- Investigating the endogenous expression pattern of WDR62 with cell type markers during cortical development by IHC would further support the interpretation of the observed *in vivo* phenotypes.

Response: We appreciate this point raised by Reviewer 1 and realise that we did not emphasize the well establish expression patterns of WDR62 that have been determined in prior studies. For instance, human transcriptomic datasets show that WDR62 expression peaks only in proliferative zones (VZ/SVZ) between ~15-21 post-conception weeks in humans (corresponding to ~E13-E17 in mice) and declines as neurogenesis proceeds (doi:10.3389/fcell.2021.640753). This pattern is further supported by *in situ* hybridisation and immunostaining studies in mice by Gunel (doi:10.1038/nature09327) and Xu labs (doi:10.1016/j.celrep.2013.12.016), including at E16.5 where Wdr62 is expressed in progenitors within the VZ/SVZ and IZ and in migrating neurons (doi:10.1016/j.celrep.2013.12.016).

Importantly, our current findings align with these expression profiles, demonstrating that WDR62 is functionally required in the embryonic cortex during developmental stages (E14.5–E16.5), consistent with our previous reports (doi:10.1242/jcs.107326) and with the expression of Wdr62 reported in prior studies (doi:10.1038/nature09327, doi:10.1016/j.celrep.2013.12.016).

We have now added a section to our introduction clarifying that WDR62 has indeed been previously demonstrated to be expressed in the cortex at the ages and in the cell types analysed in our study.

...WD40 repeat-containing protein 62 (WDR62), a protein that is highly expressed in the proliferative zones of the developing neocortex and has vital non-redundant functions in embryonic brain growth [2-4]. [Line 76 – 78].

- Similarly, Ki67 and pH3 should also be quantified separately in the VZ and SVZ.

Response: As indicated in our response above regarding the specificity of Tbr2 staining in Fig. 7D, the extension of Tbr2 staining into the VZ makes delineating the VZ/SVZ boundary challenging at this embryonic stage. However, to address reviewer

concerns here, we have split the VZ/SVZ region in half based on distance from the ventricular surface (**upper** and **lower** VZ/SVZ) and quantified the percentage of Ki67+GFP+ and PH3+GFP+ cells that fall into each region (Fig. 7D+E). Our segregation of the VZ/SVZ into upper and lower regions would broadly delineate between apical and delaminated progenitors to determine their proliferative capacity. We find significantly increased proliferation of HPRT depleted cells the in upper region of the VZ/SVZ (Fig. 7D) as progenitors transition to the IZ, consistent with the distribution of Tb2+ intermediate progenitors in Fig. 7F. In contrast, we did not find significant differences in the proportion of pH3+/GFP+ cells in either sub-region between treatment groups (Fig. 7E), consistent with our original analysis. There did appear to be a modest increase in the proportion of PH3+/GFP+ cells in the HPRT depleted condition within the lower VZ/SVZ (Fig. 7E). While this was not statistically different to controls, it suggests that HPRT loss may also increase mitotic activity in early neural progenitors. The mismatch in Ki67 and pH3 statistical differences may also reflect altered cell cycle length that often accompany changes to proliferation/differentiation (doi.org/10.1159/000521678).

We incorporated description of these new analyses in the results as follows:

In contrast, HPRT depletion led to increased Ki67 levels in GFP+ cells (Fig. 7C+D). Specifically, there was a significant increase in Ki67+ cells in the IZ and the upper region of VZ/SVZ of HPRT depleted cortices (Fig. 7C+D). We also observed a modest but non-significant reduction in PH3 levels in WDR62 depleted progenitors and apparent increase in PH3 with HPRT loss within VZ/SVZ regions (Fig. 7C+E). (Line 455-460)

Referee #2:

In the revised version of the manuscript, the authors have substantially addressed many of the concerns previously raised by this reviewer. One of the main issues-the overly general nature of the study-has been notably improved by the inclusion of the in utero electroporation knockdown experiment and the subsequent evaluation of its effects on delaminating neural progenitors. These additions significantly strengthen the case for the role of WDR62 in purinosomes and their connection to brain development. Furthermore, the proximity ligation assay provides additional support for the proposed WDR62-BAG2-HPRT axis. The reorganization of the manuscript has also enhanced its clarity and improved the overall delivery of the main message.

Taken together, this reviewer now considers the manuscript suitable for publication.

The only remaining concern pertains to the following:

- Previous concern: "Fig. 3 H-J: The fluorescence signals are saturated (also evident in the intensity profile plot) and thus not applicable for any analysis. Further, how these linear ROIs are chosen? The signal pattern is not homogeneously distributed in those images. Please provide a more consistent fluorescence analysis."

- Response from the authors: We acknowledge reviewer #3 concerns but while some granules, particularly those expressing G3BP-EGFP, exhibit saturated fluorescence signals, this does not impact or prevent our analysis. Our aim was not to quantify subtle fluorescence intensity changes within individual granules, but rather to compare fluorescence signal between granules across different channels to identify overlap. The linear ROIs were selected at random to illustrate that WDR62 and G3BP signals do not overlap between WDR62 and G3BP-positive granules.

- Although the authors did not aim to quantify subtle changes in fluorescence intensity, overexposed and oversaturated pixels can introduce artifacts, compromising the quality of both the analysis and the overall work. As a result, accurate signal quantification is not feasible, and it becomes difficult to distinguish true signal from potential precipitates or nonspecific dots. I recommend selecting representative and reliable fields for analysis and providing justification for the image processing pipelines used.

Response: We appreciate the concerns raised by Reviewer #2 regarding saturated fluorescence signals and accurate determination of *bona fide* protein granules. The images utilized for Fig. 3J have now been replaced with those with fluorescence signal that is not overexposed or oversaturated.

Referee #3:

In the revised manuscript, the authors significantly improved its readability by fixing some of the inconsistencies between the main text and figures. Many of the bold claims that were not well substantiated were also addressed. While they softened several claims connecting WDR62 to the purinosome, they did not affect its overall significance and impact to the scientific community. I commend the authors on this revised manuscript. With that being said, I have a few points that remain.

Comments:

1. (Line 148, Minor) The authors state "Overall, we identified 42 interactors, with 8 of these matching entries in the BioGRID database." In Figure 1D, I still only see 5 overlapping proteins. The rebuttal letter states that there are only 5 overlapping as well. Please revise to properly clear up this inconsistency between the main text and Figure 1D.

Response: We apologise for this error. The revised text now indicates 5 overlapping interactors.

Overall, we identified 42 interactors, with 5 of these matching entries in the BioGRID database. (Line 147)

2. (Lines 156-162, Figure 1I, Minor) The authors mention the network "identified members within... several HSP40 family members (e.g., BAG2, BAG5, DNAJA2, and DNAJC7)..." When I look at the network in Figure 1I, I only see BAG2. Why are the others mentioned but not in the figure? These proteins are in Figure 1D (bold faced). Therefore, this reviewer still is not sure why proteins identified in the BioID screen are also not present in the network image despite being referenced in the main text.

Response: Figure 1I was mis-characterized in the text description of the results. We apologize for this error. Figure 1I depicts WDR62 interactors identified in our BirA proximity labelling study while BAG5, DNAJA2 and DNAJC7 are WDR62 interactors from BioGRID database. We chose not to add the BioGRID interactors in the STRING network in Fig. 1I to aid visual presentation of data (less cluttered). In our revision, we made minor changes to the text in the 'Results' to more accurately describe to the findings depicted in Fig. 1D and Fig 1I. We have revised the text to the following:

This identified members of the chaperonin-containing TCP1 complex (CCT2-CCT8, TCP1), the stress-responsive molecular chaperones HSP60 and HSP70, as well as several HSP40 family members (e.g., BAG2, BAG5, DNAJA2, DNAJC7) that function as HSC70/90 and HSP70/90 co-chaperones under basal and stress-activated conditions (Fig. 1D) [36, 37]. (Line 150-154)

We next used STRING to assemble visualize a network of WDR62-associated proteins identified in our proximity labelling experiments. This reinforced a large subset of interactors with functions in protein folding in addition to centrosomal and cytoskeletal functions (Fig. 1I). (Line 162-165)

3. (Figure 1I, Major) In the rebuttal, the authors state "As the reviewer correctly points out, purine metabolic enzymes were not direct interactors of WDR62. Purine enzymes are linked to the molecular chaperones which, in turn, associated with WDR62 from the BioGRID database." There are several protein classes, many more notable than purine metabolic enzymes such as kinases, that are known to interact with HSP70 and their co-chaperones (HSP40s), but it is still not clear how the authors narrowed in on purine metabolism from data generated by these two independent sources at this point in the manuscript. I would have assumed a lot more nodes from the protein folding/chaperone cluster. It appears that there was some bias introduced into this network to link WDR62 with purine metabolism to substantiate their hypothesis. Given that purine enzymes are not mentioned until much later in the manuscript, maybe Figure 1I should just be only inclusive of the results generated from Figure 1D. The connection then can be made when

introducing their hypothesis of a link between WDR62 and purine metabolism later. Otherwise, this is confusing to the reader as to why these are included in this figure.

Response: We appreciate this valuable feedback from reviewer #3. As suggested, we removed the purine enzyme links with chaperones from the STRING analysis in Fig. 1I. The interactions in our revised Fig. 1I now depict direct interacting partners of WDR62 identified in our BioID analysis. As purine enzymes are removed from Fig. 1I, we have also removed the following reference to the figure in our results section.

~~Further, we incorporated a separate cluster of purine metabolic enzymes into our STRING network analysis, revealing significant interactions with chaperone machinery that were associated with WDR62 (Fig. 1I).~~

4. (Methods, Minor) For the protein mass spectrometry, please provide the same level of detail of sample preparation, data acquisition and analysis as was provided for the metabolite tracing experiments. In addition, it would be helpful to include the number of technical replicates that were performed with each condition.

Response: We have now added additional requested information on how protein fractions from streptavidin bead pulldowns were processed for LC-MS/MS in our Methods section (Line 806-815). We performed multiple pulldowns (n=3) and WDR62-interacting proteins identified in 2 separate experiments were considered.

5. (Overall, Major) Given that the connection between WDR62 and purine metabolism is through HSP70/90 chaperones, I would highly recommend that the authors perform a co-IP (at least) to demonstrate that there are these associations between WDR62 and the chaperones. The question remains whether WDR62 is a client of these chaperones or not, and this would be helpful in proper interpretation and rationale for using HSP90 and HSP70 inhibitors. Is it possible that WDR62 only interacts with BAG2-HSP70 and not HSP90? HSP70 is known to have plenty of functions independent of HSP90. Further, HSP90 inhibitors are not the most specific. Therefore, the conclusions could be strengthened with co-IPs.

Response: We have performed the requested experiments and showed that affinity-tagged WDR62 co-immunoprecipitated with HSP70 but not HSP90. Moreover, in proximity ligation assays, we showed proximity interactions between endogenous WDR62 with HSP70 and HSP90. These additional findings are consistent with WDR62 forming a complex with BAG2/HSP70. While not detected by co-IP, WDR62 complexes may also include HSP90 as indicated by PLA. We also showed that these interactions, detected by co-IP and PLA, were not altered by inhibitors of HSP70/90. Thus, WDR62 does not appear to be a client of HSP70/90.

These additional finding are presented as Supplementary Figure 11. We have also added the following text to results and discussion sections of our revised manuscript.

In addition to interactions with BAG2 (Fig 2), we showed that exogenously expressed WDR62 co-immunoprecipitated with HSP70 but not HSP90 (Suppl Fig. 11A-C). Interestingly, we found endogenous WDR62 interacted with HSP70 and HSP90 in PLAs (Fig. S11D+E). These interactions were not altered with HSP70 or HSP90 activity inhibitor (Fig. S11B, D+E). Thus, WDR62 forms a complex with heat shock protein chaperones and this does not appear to be due to chaperone-client protein interaction. (Line 390-395)

We identified endogenous interactions between WDR62, HPRT, and BAG2, and associated chaperones. WDR62 is not a client protein as the inhibition of protein folding did not disrupt WDR62 interactions with BAG2/HSP70. (Line 561-563)

6. (Overall, Major) I am still not sure from the paper I understand how WDR62-BAG2 impacts HPRT. The authors lead you to think that it might be through changes in regulation of chaperones. So, is it that WDR62 interacts with HPRT, or is it that WDR62 alters chaperone activity/associations that can impact the stability of HPRT (i.e. acts as a co-chaperone)? If the latter is true, then I would recommend that HPRT be established as a client of HSP70/90 through co-IP assays +/- HSP70 and 90 inhibitors.

Response: We appreciate this insightful comment from Reviewer 3. We previously showed a clear interaction between WDR62 and HPRT that is dependent on BAG2 (Fig. 6I+J). We have investigated this further as suggested by the reviewer and showed that WDR62-HPRT interactions were not disrupted by HSP70 or HSP90 inhibitors (Fig. S12C). We also included new PLA experiments that showed significant spatial association between HSP70 and HPRT in the cytosol (Fig. S12D), evidenced by abundant cytosolic PLA puncta, consistent with an interaction or complex formation. We also tested the effect of HSP70 inhibition and observed a modest, non-significant increase in HSP70-HPRT interaction (Fig. S12D). Unfortunately, we were unable to perform PLA with HSP90 due to antibody limitations.

Whilst our data does not definitively establish HPRT as a client of HSP70, this remains a possibility. We can however conclude from our data that WDR62, BAG2 and HSP70 form a multi-subunit complex to mediate turnover of HPRT. Dissecting the contribution of each component and precise mechanism of how this happens will be challenging and probably beyond the scope to the current study.

We have included these additional findings and discussion points in our revised manuscript.

WDR62-HPRT interactions were not altered by HSP70/90 inhibition but HSP70 binding to HPRT trended higher with HSP70 activity inhibition but did not reach statistical significance (Fig. S12C+D). Thus, WDR62-HPRT complex formation does

not require protein folding activity but HPRT may represent a client of HSP70. (Line 415-419)

The interaction between HSP70-HPRT trended higher with inhibition of HSP70 activity but did not reach statistical significance (Fig. S12D). Thus, our data does not definitively establish HPRT as a client of HSP70 but this remains a possibility. The precise mechanisms behind how multi-unit complex comprising WDR62, BAG2 and HSP70 regulates HPRT turnover will require further testing. (Line 570-574)

Dear Dr. Ng,

Thank you for the submission of your revised manuscript (EMBOJ-2025-121757R) to The EMBO Journal for our consideration, and for your patience during peer review. The three original referees, who had also assessed the previous version of your manuscript, have now seen your revision, and we have received their comments, which are appended below.

I am very pleased to say that all three referees are satisfied with the revision, find the previously raised concerns adequately addressed, and recommend publication of the manuscript. In light of this input, I am glad to inform you that your manuscript has been accepted in principle for publication in The EMBO Journal. Congratulations on an excellent work.

As you will see, there are only two minor comments from referee #1 regarding the presentation of data in Figure 7. I would like to ask you to address these remaining comments in a final version of your manuscript, as well as in a brief point-by-point response detailing any changes to the manuscript and/or its Figures.

From the editorial side, there are also a few changes we need you to make in the final version of your manuscript, before we can move forward with its formal acceptance and publication in The EMBO Journal:

- All funding information should be included in the Acknowledgements section of the revised manuscript and match the information provided in our online manuscript handling system; currently, information on the "Australian Government Research Training Program (RTP) scholarship" is missing from the online system, and information on the following grant is missing from the manuscript: " grant number RSP-079-2024 for Tour de Cure PhD Support Scholarship".
- Please check our reference format in our author guidelines and adjust the formatting of your references list accordingly (the list should be alphabetical, including the names of the first 10 co-authors, followed by "et al."): <https://link.springer.com/journal/44318/submission-guidelines#cms-Reference-guidelines>.
- All mass spectrometry datasets produced in your study (BioID pulldown and metabolomics experiments) must be deposited in appropriate publicly available repositories, and the information must be provided in the Data availability section of the revised manuscript. Please make sure that the deposited data will be publicly available at the time of publication. For each dataset, the database, the unique dataset identifier (ID), and the specific and permanent URL should be included in the Data availability statement.
- Please change heading "Conflicts of interest" to "Disclosure and competing interests statement".
- Heading "Materials and Methods" should be renamed to "Methods".
- The author contributions statement should be removed from the manuscript file. Instead, we use CRediT to specify the contributions of each author in the journal submission system. Please feel free to use the free text box to provide more detailed descriptions during submission. See also our editorial policies webpage for more information: <https://link.springer.com/partners/embo-press/editorial-policies>.
- The callouts for Suppl. Fig. 11A-C should be renamed to "Appendix Fig. S11A-C"; the words "Supplemental/Supplementary" should not be used for any Figures.
- We also noticed that callouts for Fig. 5G are missing, while there is one for the missing Figure panel 5H.
- The title page of the Appendix PDF file should contain the heading "Appendix for:", followed by the manuscript's title and a Table of Contents including page numbers for all listed items. The nomenclature throughout the Appendix file and the main manuscript file should be "Appendix Figure Sx" and "Appendix Table Sx"; the legend of each Appendix Figure should be placed below the respective Figure in the single Appendix PDF file.
- Thank you for providing the Source Data for your Figures. Please note that the following points must be resolved before we can accept the manuscript for publication:
 1. all Excel files with numerical data need to be clearly labeled with the names/numbers of the corresponding Figure panels
 2. Source Data for the following Figure panels appear to be missing: Figures 3GHI, 4J, 5A
 3. The "Figure 3 Source Data" folder also contains Source Data for Figure 4 - they must be separated.
- Please note that EMBO press papers are accompanied online by:
 - A) a short (2 sentences) summary of the findings and their significance,
 - B) 2-5 short bullet points highlighting the key results, and
 - C) a synopsis image in .jpg or .png format that is exactly 550 pixels wide and 300-600 pixels high (the height is variable). Please note that all text needs to be legible at the final size.

Please upload this information along with your revised manuscript (the text for A and B should be provided in a separate Word file).

- During our standard pre-publication Figure integrity checks, we detected possible reuse of cells between some of your Figure panels, as well as a Figure in your Appendix file that appears to be repeated. Please check carefully the following Figure panels, revise them if necessary, or explicitly state reuse in the respective Figure legends if it is justified and intentional. Please also clarify and detail any changes in your Figure set in your cover letter or your point-by-point response:

1. Possible reuse of cells between Figure 3A and Figure 4F.
2. Possible reuse of cells between Figure 4D and Figure 4I.
3. Possible reuse of cells between Supp Fig4A and Supp Fig 11D.
4. Pages 18 and 19 of the Appendix file. Is this the same figure repeated?

- Our data editors have checked your Figures and their legends, and raised the following queries. Please address all points below completely in your revised manuscript (all changes should be highlighted):

1. Please define the annotated p-values ****/****/**/* and provide the exact p-values for the same in the legends of Figures 6A, D, F as appropriate.
2. Please provide the exact p-values in the legends of Figures 2D, E; 3E, K, L; 4I, 5B, C, D, E, F; 6A, B, C, E, G, H, I, J, K, L.
3. Please indicate the statistical test used for data analysis in the legends of Figures 1F-H; 4I, 6A, D, E, F.
4. Please note that information related to "n" (i.e., number and nature of replicates) is missing in the legends of Figures 6D, E, F.
5. Please note that the error bars are not defined in the legends of Figures 5B-G; 6A, B, E, G, H, I, J, K, L; 7D, E, F.
6. Please note that the scale bar needs to be defined for Figure 1B.
7. Please note that the scale bar and its definition are missing for Figure 5C.

- Please note that the legend of Movie S1 has not been removed from the Appendix file, while the Movie appears to be a sequence of GIF images. Please provide the file in a Movie format, and make sure that its legend is only provided in the same ZIP folder.

- The manuscript sections need to be named and ordered as follows: Title page - Abstract - Keywords - Introduction - Results - Discussion - Methods - Data Availability - Acknowledgements - Disclosure and Competing Interests Statement - References - Figure Legends - main Tables (if there are any) - Expanded View Figure Legends.

- Please also note that as part of the EMBO Press transparent editorial process, The EMBO Journal publishes online a Peer Review File along with each accepted manuscript. This File will be published in conjunction with your paper and will include the referee reports, your point-by-point responses and all pertinent correspondence relating to the manuscript. Please note that your Author's Checklist will also be published at the end of the Peer Review File. Please let us know in case you want to remove any data or figures from your point-by-point responses before they are published as part of the Peer Review File. Retaining unpublished data in the Peer Review File means that these count as published and that the Peer Review File would need to be referenced in future publications. Please let the editorial office know in case you want to remove any data from this file (contact@embojournal.org).

We look forward to seeing a final version of your manuscript as soon as possible. Please let us know if you have any questions.

Best regards,

Ioannis

Referee #1:

In the revised version, the authors have addressed the comments within the limitations of the experimental materials and have provided more solid evidence. I believe the manuscript is suitable for publication.

As a minor comment, I would like to mention the GFP and TBR2 images in Fig. 7F.

The combination of orange for TBR2 and green for GFP has low visibility and is difficult to see. Therefore, I suggest changing the TBR2 color to magenta.

In addition, the background level of the TBR2 image in the control group appears to be much higher than that in the Wdr62 and Hprt si groups (possibly due to subtle differences in fixation, exposure, or contrast), so adjustment seems necessary.

Referee #2:

In this newly revised version of the manuscript, the authors have addressed the remaining concern raised by this reviewer in the last round. The manuscript is suitable for publication to this reviewer.

Referee #3:

All prior critiques have been adequately addressed, and I have no further comments.

Rev_Com_number: RC-2024-02582

New_manu_number: EMBOJ-2025-121757R

Corr_author: Ng

Title: The microcephaly protein WDR62 regulates purine metabolism through chaperone interactions.

Referee #1:

In the revised version, the authors have addressed the comments within the limitations of the experimental materials and have provided more solid evidence. I believe the manuscript is suitable for publication.

As a minor comment, I would like to mention the GFP and TBR2 images in Fig. 7F. The combination of orange for TBR2 and green for GFP has low visibility and is difficult to see. Therefore, I suggest changing the TBR2 color to magenta.

Response: We have made this change to the images in Fig. 7F as suggested by reviewer #1.

In addition, the background level of the TBR2 image in the control group appears to be much higher than that in the Wdr62 and Hprt si groups (possibly due to subtle differences in fixation, exposure, or contrast), so adjustment seems necessary.

Response: We have made slight adjustment to reduce the background level of TBR2 in the representative image in the control group to address this comment from reviewer #1.

Referee #2:

In this newly revised version of the manuscript, the authors have addressed the remaining concern raised by this reviewer in the last round. The manuscript is suitable for publication to this reviewer.

Referee #3:

All prior critiques have been adequately addressed, and I have no further comments.

Dear Dr. Ng,

Congratulations on an excellent manuscript! Please excuse the delay in the processing of your manuscript, but we have now checked all remaining issues and I am pleased to inform you that your manuscript has been accepted for publication in The EMBO Journal. Thank you for comprehensively addressing the initially raised referee concerns and our editorial requests for changes and corrections.

Before we can move forward with the publication of your manuscript, we would kindly request that you address the following remaining points:

1. We noticed that the previously identified Figure-related issues have been addressed by either correcting or replacing the Figures, but we have not received the requested explanation/description of the changes. Please send to me a detailed explanation of all changes in the Figure set and of the reasons that necessitated the changes.
2. We kindly request that your mass spectrometry data be deposited in a publicly accessible dedicated repository for proteomics data (we would recommend PRIDE: PRoteomics IDentification Database). Please send me the database, accession IDs of all deposited datasets, and the specific and permanent URLs, which I will then add to the "Data availability" section of your manuscript.
3. Please note that no page numbers appear on most of your Appendix file pages. Please send me a corrected copy of this PDF file with page numbers included on all pages.
4. Please note that it is The EMBO Journal policy for the transcript of the editorial process (containing referee reports and your response letters) to be published as an online supplement to each paper. If you should prefer removal of any referee-only figures included in the point-by-point response(s), e.g. because they may still be used for future publication or because they have been reproduced from published work by others, please do let me know via response email. More information is available here: <https://link.springer.com/partners/embo-press/editorial-policies#Peer%20review>.

Once we have received this information from you, your manuscript will be processed for publication by EMBO Press. It will be copy edited and you will receive page proofs prior to publication. Please note that you will be contacted by Springer Nature Author Services to complete licensing and payment information.

You may qualify for financial assistance for your publication charges - either via a Springer Nature fully open access agreement or an EMBO initiative. Check your eligibility: <https://link.springer.com/journal/44318/how-to-publish-with-us>

If you have any questions, please do not hesitate to contact the Editorial Office. Thank you for your contribution to The EMBO Journal. Working with you has been a pleasure.

Best regards,

Ioannis

Rev_Com_number: RC-2024-02582

New_manu_number: EMBOJ-2025-121757R1

Corr_author: Ng

Title: The microcephaly protein WDR62 regulates purine metabolism through chaperone interactions.